# Robust and differentially private mean estimation

**Xiyang Liu, Weihao Kong, Sham Kakade, Sewoong Oh**
Paul G. Allen School of Computer Science and Engineering,
University of Washington
{xiyangl,whkong,sham,sewoong}@cs.washington.edu

## Abstract

In statistical learning and analysis from shared data, which is increasingly widely adopted in platforms such as federated learning and meta-learning, there are two major concerns: privacy and robustness. Each participating individual should be able to contribute without the fear of leaking one's sensitive information. At the same time, the system should be robust in the presence of malicious participants inserting corrupted data. Recent algorithmic advances in learning from shared data focus on either one of these threats, leaving the system vulnerable to the other. We bridge this gap for the canonical problem of estimating the mean from i.i.d. samples. We introduce PRIME, which is the first efficient algorithm that achieves both privacy and robustness for a wide range of distributions. We further complement this result with a novel exponential time algorithm that improves the sample complexity of PRIME, achieving a near-optimal guarantee and matching a known lower bound for (non-robust) private mean estimation. This proves that there is no extra statistical cost to simultaneously guaranteeing privacy and robustness.

## 1  Introduction

When releasing database statistics on a collection of entries from individuals, we would ideally like to make it impossible to reverse-engineer each individual's potentially sensitive information. Privacy-preserving techniques add just enough randomness tailored to the statistical task to guarantee protection. At the same time, it is becoming increasingly common to apply such techniques to databases collected from multiple sources, not all of which can be trusted. Emerging data access frameworks, such as federated analyses across users' devices or data silos [50], make it easier to temper with such collected datasets, leaving private statistical analyses vulnerable to a malicious corruption of a fraction of the data.

Differential privacy has emerged as a widely accepted de facto measure of privacy, which is now a standard in releasing the statistics of the U.S. Census data [2] statistics and also deployed in real-world commercial systems [74, 40, 41]. A statistical analysis is said to be *differentially private* (DP) if the likelihood of the (randomized) outcome does not change significantly when a single arbitrary entry is added/removed (formally defined in §1.2). This provides a strong privacy guarantee: even a powerful adversary who knows all the other entries in the database cannot confidently identify whether a particular individual is participating in the database based on the outcome of the analysis. This ensures *plausible deniability*, central to protecting an individual's privacy.

In this paper, we focus on one of the most canonical problems in statistics: estimating the mean of a distribution from i.i.d. samples. For distributions with unbounded support, such as sub-Gaussian and heavy-tailed distributions, fundamental trade-offs between accuracy, sample size, and privacy have only recently been identified [58, 52, 54, 3] and efficient private estimators proposed. However, these approaches are brittle when a fraction of the data is corrupted, posing a real threat, referred to as *data poisoning* attacks [19, 79]. In defense of such attacks, robust (but not necessarily private) statistics has emerged as a popular setting of recent algorithmic and mathematical breakthroughs [73, 30].

35th Conference on Neural Information Processing Systems (NeurIPS 2021).

One might be misled into thinking that privacy ensures robustness since DP guarantees that a single outlier cannot change the estimation too much. This intuition is true only in a low dimension; each sample has to be an obvious outlier to significantly change the mean. However, in a high dimension, each corrupted data point can look perfectly uncorrupted but still shift the mean significant when colluding together (e.g., see Fig. 1). Focusing on the canonical problem of mean estimation, we introduce novel algorithms that achieve robustness and privacy simultaneously even when a fraction of data is corrupted arbitrarily. For such algorithms, there is a fundamental question of interest: do we need more samples to make private mean estimation also robust against adversarial corruption?

**Sub-Gaussian distributions.** If we can afford exponential run-time in the dimension, robustness can be achieved without extra cost in sample complexity. We introduce a novel estimator that $(i)$ satisfies $(\varepsilon, \delta)$-DP, $(ii)$ achieves near-optimal robustness under $\alpha$-fraction of corrupted data, achieving accuracy of $O(\alpha\sqrt{\log(1/\alpha)})$ nearly matching the fundamental lower bound of $\Omega(\alpha)$ that holds even for a (non-private) robust mean estimation with *infinite* samples, and $(iii)$ achieves near-optimal sample complexity matching that of a fundamental lower bound for a (non-robust) private mean estimation as shown in Table 1.

**Theorem 1** (Informal Theorem 7, exponential time). *Algorithm 2 is $(\varepsilon, \delta)$-DP. When $\alpha$ fraction of the data is arbitrarily corrupted from $n$ samples from a $d$-dimensional sub-Gaussian distribution with mean $\mu$ and an identity sub-Gaussian parameter, if $n = \widetilde{\Omega}(d/\alpha^2 + (d + d^{1/2}\log(1/\delta))/(\alpha\varepsilon))$ then Algorithm 2 achieves $\|\hat{\mu} - \mu\|_2 = O(\alpha\sqrt{\log(1/\alpha)})$ w.h.p.*

We introduce PRIME (PRIvate and robust Mean Estimation) in §2.3 with details in Algorithm 9 in Appendix E.1, to achieve computational efficiency. It requires a run-time of only $\widetilde{O}(d^3 + nd^2)$, but at the cost of requiring extra $d^{1/2}$ factor larger number of samples. This cannot be improved upon with current techniques since efficient robust estimators rely on the top PCA directions of the covariance matrix to detect outliers. [78] showed that $\widetilde{\Omega}(d^{3/2})$ samples are necessary to compute PCA directions while preserving $(\varepsilon, \delta)$-DP when $\|x_i\|_2 = O(\sqrt{d})$. It remains an open question if this $\widetilde{\Omega}(d^{3/2}/(\alpha\varepsilon))$ bottleneck is fundamental; no matching lower bound is currently known.

**Theorem 2** (Informal Theorem 6, polynomial time). *PRIME is $(\varepsilon, \delta)$-DP and under the assumption of Thm.1, if $n = \widetilde{\Omega}(d/\alpha^2 + (d^{3/2}\log(1/\delta))/(\alpha\varepsilon))$, achieves $\|\hat{\mu} - \mu\|_2 = O(\alpha\sqrt{\log(1/\alpha)})$ w.h.p.*

|  | Upper bound (poly-time) | Upper bound (exp-time) | Lower bound |
|---|---|---|---|
| $(\varepsilon, \delta)$-DP [52] | $\widetilde{O}(\frac{d}{\alpha^2} + \frac{d\log^{1/2}(1/\delta)}{\alpha\varepsilon})$ | $\widetilde{O}(\frac{d}{\alpha^2} + \frac{d}{\alpha\varepsilon})^{\clubsuit}$ | $\widetilde{\Omega}(\frac{d}{\alpha^2} + \frac{d}{\alpha\varepsilon})^{\spadesuit}$ |
| $\alpha$-corruption [36] | $\widetilde{O}(\frac{d}{\alpha^2})$ | $O(\frac{d}{\alpha^2})$ | $\Omega(\frac{d}{\alpha^2})$ |
| $\alpha$-corruption and $(\varepsilon, \delta)$-DP (this paper) | $\widetilde{O}(\frac{d}{\alpha^2} + \frac{d^{3/2}\log(1/\delta)}{\alpha\varepsilon})$ [Theorem 6] | $\widetilde{O}(\frac{d}{\alpha^2} + \frac{d + d^{1/2}\log(1/\delta)}{\alpha\varepsilon})$ [Theorem 7] | $\widetilde{\Omega}(\frac{d}{\alpha^2} + \frac{d}{\alpha\varepsilon})^{\spadesuit}$ [52] |

Table 1: For estimating the mean $\mu \in \mathbb{R}^d$ of a *sub-Gaussian* distribution with a known covariance, we list the sufficient or necessary conditions on the sample sizes to achieve an error $\|\hat{\mu} - \mu\|_2 = \widetilde{O}(\alpha)$ under $(\varepsilon, \delta)$-DP, corruption of an $\alpha$-fraction of samples, and both. $\clubsuit$ requires the distribution to be a Gaussian [14] and $\spadesuit$ requires $\delta \leq \sqrt{d}/n$.

**Heavy-tailed distributions.** When samples are drawn from a distribution with a bounded covariance, parameters of Algorithm 2 can be modified to nearly match the optimal sample complexity of (non-robust) private mean estimation in Table 2. This algorithm also matches the fundamental limit on the accuracy of (non-private) robust estimation, which in this case is $\Omega(\alpha^{1/2})$.

**Theorem 3** (Informal Theorem 8, exponential time). *From a distribution with mean $\mu \in \mathbb{R}^d$ and covariance $\Sigma \preceq \mathbf{I}$, $n$ samples are drawn and $\alpha$-fraction is corrupted. Algorithm 2 is $(\varepsilon, \delta)$-DP and if $n = \widetilde{\Omega}((d + d^{1/2}\log(1/\delta))/(\alpha\varepsilon) + d^{1/2}\log^{3/2}(1/\delta)/\varepsilon)$ achieves $\|\hat{\mu} - \mu\|_2 = O(\alpha^{1/2})$ w.h.p.*

The proposed PRIME-HT for covariance bounded distributions achieve computational efficiency at the cost of an extra factor of $d^{1/2}$ in sample size. This bottleneck is also due to DP PCA, and it remains open whether this gap can be closed by an efficient estimator.

**Theorem 4** (Informal Theorem 9, polynomial time). *PRIME-HT is $(\varepsilon, \delta)$-DP and if $n = \widetilde{\Omega}((d^{3/2}\log(1/\delta))/(\alpha\varepsilon))$ achieves $\|\hat{\mu} - \mu\|_2 = O(\alpha^{1/2})$ w.h.p. under the assumptions of Thm. 3.*

| | Upper bound (poly-time) | Upper bound (exp-time) | Lower bound |
|---|---|---|---|
| $(\varepsilon,\delta)$-DP [54] | $\widetilde{O}(\frac{d\log^{1/2}(1/\delta)}{\alpha\varepsilon})$ | $\widetilde{O}(\frac{d\log^{1/2}(1/\delta)}{\alpha\varepsilon})$ | $\Omega(\frac{d}{\alpha\varepsilon})$ |
| $\alpha$-corruption [36] | $\widetilde{O}(\frac{d}{\alpha})$ | $\widetilde{O}(\frac{d}{\alpha})$ | $\Omega(\frac{d}{\alpha})$ |
| $\alpha$-corruption and $(\varepsilon,\delta)$-DP (this paper) | $\widetilde{O}(\frac{d^{3/2}\log(1/\delta)}{\alpha\varepsilon})$ [Theorem 9] | $\widetilde{O}(\frac{d+d^{1/2}\log^{3/2}(1/\delta)}{\alpha\varepsilon})$ [Theorem 8] | $\Omega(\frac{d}{\alpha\varepsilon})$ ([54]) |

Table 2: For estimating the mean $\mu \in \mathbb{R}^d$ of a covariance bounded distribution, we list the sufficient or necessary conditions on the sample size to achieve an error $\|\hat{\mu} - \mu\|_2 = O(\alpha^{1/2})$ under $(\varepsilon,\delta)$-DP, corruption of an $\alpha$-fraction of samples, and both.

## 1.1 Technical contributions

We introduce PRIME which simultaneously achieves $(\varepsilon,\delta)$-DP and robustness against $\alpha$-fraction of corruption. A major challenge in making a standard filter-based robust estimation algorithm (e.g., [30]) private is the high sensitivity of the filtered set that we pass from one iteration to the next. We propose a new framework which makes private only the statistics of the set, hence significantly reducing the sensitivity. Our major innovation is a tight analysis of the end-to-end sensitivity of this multiple interactive accesses to the database. This is critical in achieving robustness while preserving privacy and is also of independent interest in making general iterative filtering algorithms private.

The classical filter approach (see, e.g. [30]) needs to access the database $O(d)$ times, which brings an extra $O(\sqrt{d})$ factor in the sample complexity due to DP composition. In order to reduce the iteration complexity, following the approach in [36], we propose filtering multiple directions simultaneously using a new score based on the matrix multiplicative weights (MMW). In order to privatize the MMW filter, our major innovation is a novel adaptive filtering algorithm DPTHRESHOLD($\cdot$) that outputs a *single private threshold* which guarantees sufficient progress at every iteration. This brings the number of database accesses from $O(d)$ to $O((\log d)^2)$.

One downside of PRIME is that it requires an extra $d^{1/2}$ factor in the sample complexity, compared to known lower bounds for (non-robust) DP mean estimation. To investigate whether this is also necessary, we propose a *sample optimal* exponential time robust mean estimation algorithm in §4 and prove that there is no extra statistical cost to jointly requiring privacy and robustness. Our major technical innovations is in using *resilience property of the dataset* to not only find robust mean (which is the typical use case of resilience) but also bound sensitivity of that robust mean.

## 1.2 Preliminary on differential privacy (DP)

DP is a formal metric for measuring privacy leakage when a dataset is accessed with a query [37].

**Definition 1.1.** *Given two datasets $S = \{x_i\}_{i=1}^n$ and $S' = \{x_i'\}_{i=1}^{n'}$, we say $S$ and $S'$ are* neighboring *if $d_\triangle(S,S') \leq 1$ where $d_\triangle(S,S') \triangleq \max\{|S \setminus S'|, |S' \setminus S|\}$, which is denoted by $S \sim S'$. For an output of a stochastic query $q$ on a database, we say $q$ satisfies $(\varepsilon,\delta)$-differential privacy for some $\varepsilon > 0$ and $\delta \in (0,1)$ if $\mathbb{P}(q(S) \in A) \leq e^\varepsilon \mathbb{P}(q(S') \in A) + \delta$ for all $S \sim S'$ and all subset $A$.*

Let $z \sim \mathrm{Lap}(b)$ be a random vector with entries i.i.d. sampled from Laplace distribution with pdf $(1/2b)e^{-|z|/b}$. Let $z \sim \mathcal{N}(\mu,\Sigma)$ denote a Gaussian random vector with mean $\mu$ and covariance $\Sigma$.

**Definition 1.2.** *The* sensitivity *of a query $f(S) \in \mathbb{R}^k$ is defined as $\Delta_p = \sup_{S \sim S'} \|f(S) - f(S')\|_p$ for a norm $\|x\|_p = (\sum_{i \in [k]} |x_i|^p)^{1/p}$. For $p = 1$, the Laplace mechanism outputs $f(S) + \mathrm{Lap}(\Delta_1/\varepsilon)$ and achieves $(\varepsilon,0)$-DP [37]. For $p = 2$, the Gaussian mechanism outputs $f(S) + \mathcal{N}(0, (\Delta_2(\sqrt{2\log(1.25/\delta)})/\varepsilon)^2 \mathbf{I})$ and achieves $(\varepsilon,\delta)$-DP [38].*

We use these output perturbation mechanisms along with the exponential mechanism [69] as building blocks. Appendix A provides detailed survey of privacy and robust estimation.

## 1.3 Problem formulation

We are given $n$ samples from a sub-Gaussian distribution with a known covariance but unknown mean, and $\alpha$ fraction of the samples are corrupted by an adversary. Our goal is to estimate the

unknown mean. We follow the standard definition of adversary in [30], which can adaptively choose which samples to corrupt and arbitrarily replace them with any points.

**Assumption 1.** *An uncorrupted dataset $S_{\text{good}}$ consists of $n$ i.i.d. samples from a $d$-dimensional sub-Gaussian distribution with mean $\mu \in \mathbb{R}^d$ and covariance $\mathbb{E}[xx^\top] = \mathbf{I}_d$, which is 1-sub-Gaussian, i.e., $\mathbb{E}[\exp(v^\top x)] \leq \exp(\|v\|_2^2/2)$ for all $v \in \mathbb{R}^d$. For some $\alpha \in (0, 1/2)$, we are given a corrupted dataset $S = \{x_i \in \mathbb{R}^d\}_{i=1}^n$ where an adversary adaptively inspects all the samples in $S_{\text{good}}$, removes $\alpha n$ of them, and replaces them with $S_{\text{bad}}$ which are $\alpha n$ arbitrary points in $\mathbb{R}^d$.*

Similarly, we consider the same problem for heavy-tailed distributions with a bounded covariance. We present the assumption and main results for covariance bounded distributions in Appendix B.

**Notations.** Let $[n] = \{1, 2, \ldots, n\}$. For $x \in \mathbb{R}^d$, we use $\|x\|_2 = (\sum_{i \in [d]} (x_i)^2)^{1/2}$ to denote the Euclidean norm. For $X \in \mathbb{R}^{d \times d}$, we use $\|X\|_2 = \max_{\|v\|_2=1} \|Xv\|_2$ to denote the spectral norm. The $d \times d$ identity matrix is $\mathbf{I}_{d \times d}$. Whenever it is clear from context, we use $S$ to denote both a set of data points and also the set of indices of those data points. $\widetilde{O}$ and $\widetilde{\Omega}$ hide poly-logarithmic factors in $d, n, 1/\alpha$, and the failure probability.

**Outline.** We present PRIME for sub-Gaussian distribution in §2, and present theoretical analysis in §3. We then introduce an exponential time algorithm with near optimal guarantee in §4. Due to space constraints, analogous results for heavy-tailed distributions are presented in Appendix B.

## 2 PRIME: efficient algorithm for robust and DP mean estimation

In order to describe the proposed algorithm PRIME, we need to first describe a standard (non-private) iterative filtering algorithm for robust mean estimation.

### 2.1 Background on (non-private) iterative filtering for robust mean estimation

*Non-private* robust mean estimation approaches recursively apply the following *filter*, whose framework is first proposed in [28]. Given a dataset $S = \{x_i\}_{i=1}^n$, the current set $S_0 \subseteq [n]$ of data points is updated starting with $S_1 = [n]$. At each step, the following filter (Algorithm 1 in [63]) attempts to detect the corrupted data points and remove them.

1. Compute the top eigenvector $v_t \leftarrow \arg \max_{v:\|v\|_2=1} v^\top \mathrm{Cov}(S_{t-1}) v$ of the covariance of the current data set $\{x_i\}_{i \in S_{t-1}}$ ;

2. Compute scores for all data points $j \in S_{t-1}$: $\tau_j \leftarrow \left( v_t^\top (x_j - \mathrm{Mean}(S_{t-1})) \right)^2$ ;

3. Draw a random threshold: $Z_t \leftarrow \mathrm{Unif}([0, 1])$ ;

4. Remove outliers from $S_{t-1}$ defined as $\{i \in S_{t-1} : \tau_i \text{ is in the largest } 2\alpha\text{-tail of } \{\tau_j\}_{j \in S_{t-1}}$ and $\tau_i \geq Z_t \tau_{\max}\}$, where $\tau_{\max} = \max_{j \in S_{t-1}} \tau_j$

This is repeated until the empirical covariance is sufficiently small and the empirical mean $\hat{\mu}$ is output. At a high level, the correctness of this algorithm relies on the key observation that the $\alpha$-fraction of adversarial corruption can not significantly change the mean of the dataset without introducing large eigenvalues in the empirical covariance. Therefore, the algorithm finds top eigenvector of the empirical covariance in step 1, and tries to correct the empirical covariance by removing corrupted data points. Each data point is assigned a score in step 2 which indicates the "badness" of the data points, and a threshold $Z_t$ in step 3 is carefully designed such that step 4 guarantees to remove more corrupted data points than good data points (in expectation). This guarantees the following bound achieving the near-optimal sample complexity shown in the second row of Table 1. A formal description of this algorithm is in Algorithm 4 in Appendix C.

**Proposition 2.1** (Corollary of [63, Theorem 2.1]). *Under assumption 1, the above filtering algorithm achieves accuracy $\|\hat{\mu} - \mu\|_2 \leq O(\alpha \sqrt{\log(1/\alpha)})$ w.p. 0.9 if $n \geq \widetilde{\Omega}(d/\alpha^2)$.*

**Challenges in making robust mean estimation private.** To get a DP and robust mean, a naive attempt is to apply a standard output perturbation mechanism to $\hat{\mu}$. However, this is obviously challenging since the end-to-end sensitivity is intractable. The standard recipe to circumvent this is to make the current "state" $S_t$ private at every iteration. Once $S_{t-1}$ is private (hence, public knowledge), making the next "state" $S_t$ private is simpler. We only need to analyze the sensitivity of a single step

and apply some output perturbation mechanism with $(\varepsilon_t, \delta_t)$. End-to-end privacy is guaranteed by accounting for all these $(\varepsilon_t, \delta_t)$'s using the advanced composition [51]. This recipe has been quite successful, for example, in training neural networks with (stochastic) gradient descent [1], where the current state can be the optimization variable $\mathbf{x}_t$. However, for the above (non-private) filtering algorithm, this standard recipe fails, since the state $S_t$ is a set and has large sensitivity. Changing a single data point in $S_t$ can significantly alter which (and how many) samples are filtered out.

## 2.2  A new framework for *private* iterative filtering

Instead of making the (highly sensitive) $S_t$ itself private, we propose a new framework which makes private only the statistics of $S_t$: the mean $\mu_t$ and the top principal direction $v_t$. There are two versions of this algorithm, which output the exactly same $\hat{\mu}$ with the exactly same privacy guarantees, but are written from two different perspectives. We present here the *interactive* version from the perspective of an analyst accessing the dataset via DP queries ($q_{\mathrm{range}}, q_{\mathrm{size}}, q_{\mathrm{mean}}, q_{\mathrm{norm}}$ and $q_{\mathrm{PCA}}$), because this version makes clear the inner operations of each private mechanisms, hence making $(i)$ the sensitivity analysis transparent, $(ii)$ checking the correctness of privacy guarantees easy, and $(iii)$ tracking privacy accountant simple. In practice, one should implement the *centralized* version (Algorithm 7 in Appendix D), which is significantly more efficient.

---

**Algorithm 1:** Private iterative filtering (interactive version)

**Input:**  $S = \{x_i\}_{i \in [n]}$, $\alpha \in (0, 1/2)$, probability $\zeta \in (0, 1)$, # of iterations $T = \Theta(d)$, $(\varepsilon, \delta)$

1  $(\bar{x}, B) \leftarrow q_{\mathrm{range}}(S, 0.01\varepsilon, 0.01\delta)$

2  $\varepsilon_1 \leftarrow \min\{0.99\varepsilon, 0.9\}/(4\sqrt{2T \log(2/\delta)})$, $\delta_1 \leftarrow 0.99\delta/(8T)$

3  **if** $n < (4/\varepsilon_1) \log(1/(2\delta_1))$ **then Output:** $\emptyset$

4  **for** $t = 1, \ldots, T$ **do**

5  $\quad$ $n_t \leftarrow q_{\mathrm{size}}(\{(\mu_\ell, v_\ell, Z_\ell)\}_{\ell \in [t-1]}, \varepsilon_1, \bar{x}, B)$, **if** $n_t < 3n/4$ **then Output:** $\emptyset$

6  $\quad$ $\mu_t \leftarrow q_{\mathrm{mean}}(\{(\mu_\ell, v_\ell, Z_\ell)\}_{\ell \in [t-1]}, \varepsilon_1, \bar{x}, B)$

7  $\quad$ $\lambda_t \leftarrow q_{\mathrm{norm}}(\{(\mu_\ell, v_\ell, Z_\ell)\}_{\ell \in [t-1]}, \mu_t, \varepsilon_1, \bar{x}, B)$

8  $\quad$ **if** $\lambda_t \leq (C - 0.01)\alpha \log 1/\alpha$ **then Output:** $\mu_t$

9  $\quad$ $v_t \leftarrow q_{\mathrm{PCA}}(\{(\mu_\ell, v_\ell, Z_\ell)\}_{\ell \in [t-1]}, \mu_t, \varepsilon_1, \delta_1, \bar{x}, B))$

10  $\quad$ $Z_t \leftarrow \mathrm{Unif}([0, 1])$

**Output:** $\mu_t$

---

We give a high-level explanation of each step of Algorithm 1 here and give the formal definitions of all the queries in Appendix D. First, $q_{\mathrm{range}}$ returns (the parameters of) a hypercube $\bar{x} + [-B/2, B/2]^d$ that is guaranteed to include all uncorrupted samples while preserving privacy. This is achieved by running $d$ coordinate-wise private histograms and selecting $\bar{x}_j$ as the center of the largest bin for the $j$-th coordinate. Since covariance is $\mathbf{I}$, $q_{\mathrm{range}}$ returns a fixed $B = 8\sigma\sqrt{\log(dn/\zeta)}$. Such an adaptive estimate of the support is critical in tightly bounding the sensitivity of all subsequent queries, which operate on the clipped dataset; all data points are projected as $\mathcal{P}_{\bar{x}+[-B/2,B/2]^d}(x) = \arg\min_{y \in \bar{x}+[-B/2,B/2]^d} \|y - x\|_2$ in all the queries that follow. With clipping, a single data point can now change at most by $B\sqrt{d}$.

The subsequent steps perform the non-private filtering algorithm of §2.1, but with private statistics $\mu_t$ and $v_t$. As the set $S_t$ changes over time, we lower bound its size (which we choose to be $|S_t| > n/2$) to upper bound the sensitivity of other queries $q_{\mathrm{mean}}, q_{\mathrm{norm}}$ and $q_{\mathrm{PCA}}$.

At the $t$-th iterations, every time a query is called the data curator $(i)$ uses $(\bar{x}, B)$ to clip the data, $(ii)$ computes $S_t$ by running $t - 1$ steps of the non-private filtering algorithm of §2.1 but with a given *fixed* set of parameters $\{(\mu_\ell, v_\ell)\}_{\ell \in [t-1]}$ (and the given randomness $\{Z_\ell\}_{\ell \in [t-1]}$), and $(iii)$ computes the queried private statistics of $S_t$. If the private spectral norm of the covariance of $S_t$ (i.e., $\lambda_t$) is sufficiently small, we output the private and robust mean $\hat{\mu} = \mu_t$ (line 8). Otherwise, we compute the private top PCA direction $v_t$ and draw an randomness $Z_t$ to be used in the next step of filtering, as in the non-private filtering algorithm. We emphasize that $\{S_\ell\}$ are not private, and hence never returned to the analyst. We also note that this interactive version is redundant as every query is re-computing $S_t$. In our setting, the analyst has the dataset and there is no need to separate them. This leads to a

*centralized* version we provide in Algorithm 7 in the appendix, which avoids redundant computations and hence is significantly more efficient.

The main challenge in this framework is the privacy analysis. Because $\{S_\ell\}_{\ell \in [t-1]}$ is not private, each query runs $t-1$ steps of filtering whose end-to-end sensitivity could blow-up. Algorithmically, $(i)$ we start with a specific choice of a non-private iterative filtering algorithm (among several variations that are equivalent in non-private setting but widely differ in its sensitivity), and $(ii)$ make appropriate changes in the private queries (Algorithm 1) to keep the sensitivity small. Analytically, the following key technical lemma allows a sharp analysis of the end-to-end sensitivity of iterative filtering.

**Lemma 2.2.** *Let $S_t(\mathcal{S})$ denote the resulting subset of samples after $t$ iterations of the filtering in the queries ($q_{\text{size}}$, $q_{\text{mean}}$, $q_{\text{norm}}$, and $q_{\text{PCA}}$) are applied to a dataset $\mathcal{S}$ using* fixed *parameters $\{(\mu_\ell, v_\ell, Z_\ell)\}_{\ell=1}^t$. Then, we have $d_\triangle(S_t(\mathcal{S}), S_t(\mathcal{S}')) \leq d_\triangle(\mathcal{S}, \mathcal{S}')$, where $d_\triangle(\mathcal{S}, \mathcal{S}') \triangleq \max\{|\mathcal{S} \setminus \mathcal{S}'|, |\mathcal{S}' \setminus \mathcal{S}|\}$.*

Recall that two datasets are neighboring, i.e., $\mathcal{S} \sim \mathcal{S}'$, iff $d_\triangle(\mathcal{S}, \mathcal{S}') \leq 1$. This lemma implies that if two datasets are neighboring, then they are still neighboring after filtering with the same parameters, no matter how many times we filter them. Hence, this lemma allows us to use the standard output-perturbation mechanisms with $(\varepsilon_1, \delta_1)$-DP. Advanced composition ensures that end-to-end guarantee of $4T$ such queries is $(0.99\varepsilon, 0.99\delta)$-DP. Together with $(0.01\varepsilon, 0.01\delta)$-DP budget used in $q_{\text{range}}$, this satisfied the target privacy. Analyzing the utility of this algorithm, we get the following guarantee.

**Theorem 5.** *Algorithm 1 is $(\varepsilon, \delta)$-DP. Under Assumption 1, there exists a universal constant $c \in (0, 0.1)$ such that if $\alpha \leq c$ and $n = \widetilde{\Omega}\left((d/\alpha^2) + d^2(\log(1/\delta))^{3/2}/(\varepsilon\alpha)\right)$ then Algorithm 1 achieves $\|\hat{\mu} - \mu\|_2 \leq O(\alpha\sqrt{\log(1/\alpha)})$ with probability $0.9$.*

The first term $O(d/\alpha^2)$ in the sample complexity is optimal (cf. Table 1), but there is a factor of $d$ gap in the second term. This is due to the fact that we need to run $O(d)$ iterations in the worst-case. Such numerous accesses to the database result in large noise to be added at each iteration, requiring large sample size to combat that extra noise. We introduce PRIME to reduce the number of iterations to $O((\log d)^2)$ and significantly reduce the sample complexity.

## 2.3 PRIME: novel robust and private mean estimator

Algorithm 1 (specifically Filter($\cdot$) in Algorithm 1) accesses the database $O(d)$ times. This is necessary for two reasons. First, the filter checks only one direction $v_t$ at each iteration. In the worst case, the corrupted samples can be scattered in $\Omega(d)$ orthogonal directions such that the filter needs to be repeated $O(d)$ times. Secondly, even if the corrupted samples are clustered together in one direction, the filter still needs to be repeated $O(d)$ times. This is because we had to use a large (random) threshold of $dB^2 Z_t = O(d)$ to make the threshold data-independent so that we can keep the sensitivity of Filter($\cdot$) low, which results in slow progress. We propose filtering multiple directions simultaneously using a new score $\{\tau_i\}$ based on the matrix multiplicative weights. Central to this approach is a novel adaptive filtering algorithm DPTHRESHOLD($\cdot$) that guarantees sufficient decrease in the total score at every iteration.

### 2.3.1 Matrix Multiplicative Weight (MMW) scoring

The MMW-based approach, pioneered in [36] for non-private robust mean estimation, filters out multiple directions simultaneously. It runs over $O(\log d)$ epochs and every epoch consists of $O(\log d)$ iterations. At every epoch $s$ and iteration $t$, step 2 of the iterative filtering in §2.1 is replaced by a new score $\tau_i = (x_i - \text{Mean}(S_t^{(s)}))^T U_t^{(s)}(x_i - \text{Mean}(S_t^{(s)}))$ where $U_t^{(s)}$ now accounts for all directions in $\mathbb{R}^d$ but appropriately weighted. Precisely, it is defined via the matrix multiplicative update:

$$U_t^{(s)} = \frac{\exp\left(\alpha^{(s)} \sum_{r \in [t]}(\text{Cov}(S_r^{(s)}) - \mathbf{I})\right)}{\text{Tr}\left(\exp(\alpha^{(s)} \sum_{r \in [t]}(\text{Cov}(S_r^{(s)}) - \mathbf{I}))\right)},$$

for some choice of $\alpha^{(s)} > 0$. If we set the number of iterations to one, a choice of $\alpha^{(s)} = \infty$ recovers the previous score that relied on the top singular vector from §2.1 and a choice of $\alpha^{(s)} = 0$ gives a simple norm based score $\tau_i = \|x_i\|_2^2$. An appropriate choice of $\alpha^{(s)}$ smoothly interpolates between these two extremes, which ensures that $O(\log d)$ iterations are sufficient for the spectral norm of the

covariance to decrease strictly by a constant factor. This guarantees that after $O(\log d)$ epochs, we sufficiently decrease the covariance to ensure that the empirical mean is accurate enough. Critical in achieving this gain is our carefully designed filtering algorithm DPTHRESHOLD that uses the privately computed MMW-based scores using Gaussian mechanism on the covariance matrices as shown in Algorithm 11 in Appendix E.

### 2.3.2 Adaptive filtering with DPTHRESHOLD

**Novelty.** The corresponding non-private filtering of [36, Algorithm 9] for robust mean estimation takes advantage of an *adaptive threshold*, but filters out each sample independently resulting in a prohibitively large sensitivity; the coupling between each sample and the randomness used to filter it can change widely between two neighboring datasets. On the other hand, Algorithm 1 (i.e., Filter(·) in Algorithm 6) takes advantage of jointly filtering all points above a *single threshold* $B^2 dZ_t$ with a single randomness $Z_t \sim \text{Unif}[0, 1]$, but the non-adaptive (and hence large) choice of the range $B^2 d$ results in a large number of iterations because each filtering only decrease the score by little. To sufficiently reduce the total score while maintaining a small sensitivity, we introduce a filter with a single and adaptive threshold.

**Algorithm.** Our goal here is to privately find a single scalar $\rho$ such that when a randomized filter is applied on the scores $\{\tau_i\}$ with a (random) threshold $\rho Z$ (with $Z$ drawn uniform in $[0, 1]$), we filter out enough samples to make progress in each iteration while ensuring that we do not remove too many uncorrupted samples. This is a slight generalization of the non-private algorithm in Section 2.1, which simply set $\rho = \max_{j \in S_t} \tau_j$. While this guarantees the filter removes more corrupted samples than good samples, it does not make sufficient progress in reducing the total score of the samples.

Ideally, we want the thresholding to decrease the total score by a constant multiplicative factor, which will in the end allow the algorithm to terminate within logarithmic iterations. To this end, we propose a new scheme of using the largest $\rho$ such that the following inequality holds:

$$\sum_{\tau_i > \rho} (\tau_i - \rho) \geq 0.31 \sum_{\tau_i \in S_t} (\tau_i - 1) . \tag{1}$$

We use a private histogram of the scores to approximate this threshold. Similar to [55, 58], we use geometrically increasing bin sizes such that we use only $O(\log B^2 d)$ bins while achieving a preferred *multiplicative* error in our quantization. At each epoch $s$ and iteration $t$, we run DPTHRESHOLD sketched in the following to approximate $\rho$ followed by a random filter. Step 3 replaces the non-private condition in Eq. (1). A complete description is provided in Algorithm 11.

1. Privately compute scores for all data points $i \in S_t^{(s)}$ : $\tau_i \leftarrow (x_i - \mu_t)^\top U_t^{(s)} (x_i - \mu_t)$ ;
2. Compute a private histogram $\{\tilde{h}_j\}_{j=1}^{2+\log(B^2 d)}$ of the scores over geometrically sized bins $I_1 = [1/4, 1/2), I_2 = [1/2, 1), \ldots, I_{2+\log(B^2 d)} = [2^{\log(B^2 d)-1}, 2^{\log(B^2 d)}]$ ;
3. Privately find the largest $\ell$ satisfying $\sum_{j \geq \ell} (2^j - 2^\ell) \tilde{h}_j \geq 0.31 \sum_{i \in S_t^{(s)}} (\tau_i - 1)$ ;
4. Output $\rho = 2^\ell$ .

## 3 Analyses of PRIME

Building on the framework of Algorithm 1, PRIME (Algorithm 9) replaces the score with the MMW-based score presented in §2.3.1 and the filter with the adaptive DPTHRESHOLD. This reduces the number of iterations to $T = O((\log d)^2)$ achieving the following bound.

**Theorem 6.** *PRIME is $(\varepsilon, \delta)$-differentially private. Under Assumption 1 there exists a universal constant $c \in (0, 0.1)$ such that if $\alpha \leq c$ and $n = \widetilde{\Omega}((d/\alpha^2) + (d^{3/2}/(\varepsilon\alpha)) \log(1/\delta))$, then PRIME achieves $\|\hat{\mu} - \mu\|_2 = O(\alpha\sqrt{\log(1/\alpha)})$ with probability $0.9$.*

A proof is provided in Appendix F. The notation $\widetilde{\Omega}(\cdot)$ hides logarithmic terms in $d$, $R$, and $1/\alpha$. To achieve an error of $O(\alpha\sqrt{\log(1/\alpha)})$, the first term $\widetilde{\Omega}(d/\alpha^2 \log(1/\alpha))$ is necessary even if there is no corruption. The accuracy of $O(\alpha\sqrt{\log(1/\alpha)})$ matches the lower bound shown in [33] for any polynomial time statistical query algorithm, and it nearly matches the information theoretical lower bound on robust estimation of $\Omega(\alpha)$. On the other hand, the second term of $\widetilde{\Omega}(d^{3/2}/(\varepsilon\alpha \log(1/\alpha)))$

has an extra factor of $d^{1/2}$ compared to the optimal one achieved by exponential time Algorithm 2. It is an open question if this gap can be closed by a polynomial time algorithm.

The bottleneck is the private matrix multiplicative weights. Such spectral analyses are crucial in filter-based robust estimators. Even for a special case of privately computing the top principal component, the best polynomial time algorithm requires $O(d^{3/2})$ samples [39, 18, 78], and this sample complexity is also necessary as shown in [39, Corollary 25].

To boost the success probability to $1 - \zeta$ for some small $\zeta > 0$, we need an extra $\log(1/\zeta)$ factor in the sample complexity to make sure the dataset satisfies the regularity condition with probability $\zeta/2$. Then we can run PRIME $\log(1/\zeta)$ times and choose the output of a run that satisfies $n^{(s)} > n(1 - 10\alpha)$ and $\lambda^{(s)} \leq C\alpha \log(1/\alpha)$ at termination.

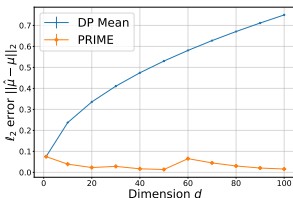 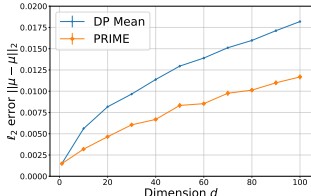 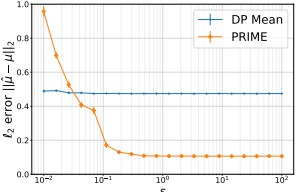

Figure 1: Private mean estimators (e.g., DP mean [52]) are vulnerable to adversarial corruption especially in high dimensions, while the proposed PRIME achieves robustness (and privacy) regardless of the dimension of the samples.

Numerical experiments support our theoretical claims. The left figure with $(\alpha, \varepsilon, \delta, n) = (0.05, 20, 0.01, 10^6)$ is in the large $\alpha$ regime where the DP Mean error is dominates by $\alpha\sqrt{d}$ and PRIME error by $\alpha\sqrt{\log(1/\alpha)}$. Hence, PRIME error is constant whereas DP Mean error increases with the dimension $d$. The second figure with $(\alpha, \varepsilon, \delta, n) = (0.001, 20, 0.01, 10^6)$ is in the small $\alpha$ regime when DP Mean error consists of $\alpha\sqrt{d} + \sqrt{d/n}$ and PRIME is dominated by $\sqrt{d/n}$. Both increase with the dimension $d$, and the gap can be made large by increasing $\alpha$. The right figure with $(\alpha, \delta, d, n) = (0.1, 0.01, 10, 10^6)$ is when DP Mean error is dominated by $\alpha\sqrt{d}$ and PRIME by $\alpha\sqrt{\log(1/\alpha)}$ when $\varepsilon > cd^{1.5}/(\alpha n)$. Below this threshold, which happens in this example around $\varepsilon = 0.05$, the added noise in the private mechanism starts to dominate with decreasing $\varepsilon$. Both algorithms have respective thresholds below which the error increases with decreasing $\varepsilon$. This threshold is larger for PRIME because it uses the privacy budget to perform multiple operations and hence the noise added to the final output is larger compared to DP Mean. Below this threshold, which can be easily determined based on the known parameters $(\varepsilon, \delta, n, \alpha)$, we should either collect more data (which will decrease the threshold) or give up filtering and spend all privacy budget on $q_{\mathrm{range}}$ and the empirical mean (which will reduce the error). Details of the experiments are in Appendix L.

## 4 Exponential time algorithm with near-optimal sample complexity

**Novelty.** An existing exponential time algorithm for robust and private mean estimation in [14] strictly requires the uncorrupted samples to be drawn from a Gaussian distribution. We also provide a similar algorithm based on private Tukey median in Appendix I and its analysis in Appendix J. In this section, we introduce a novel estimator that achieves near-optimal guarantees for more general sub-Gaussian distributions (and also covariance bounded distributions) but takes an *exponential* run-time. Its innovation is in leveraging on the *resilience* property of well-behaved distributions not only to estimate the mean robustly (which is the standard use of the property) but also to adaptively bound the sensitivity of the estimator, thus achieving optimal privacy-accuracy tradeoff.

**Definition 4.1** (Resilience from Definition 1 in [73]). *A set of points $\{x_i\}_{i \in S}$ lying in $\mathbb{R}^d$ is $(\sigma, \alpha)$-resilient around a point $\mu$ if $\|(1/|T|) \sum_{i \in T} (x_i - \mu)\|_2 \leq \sigma$ for all subsets $T \subset S$ of size $(1 - \alpha)|S|$.*

**Algorithm.** As data is corrupted, we define $R(S)$ as a surrogate for resilience of the uncorrupted part of the set. If $S$ indeed consists of a $1 - \alpha$ fraction of independent samples from the promised class of distributions, the goodness score $R(S)$ will be close to the resilience property of the good data.

**Definition 4.2** (Goodness of a set). *For $\mu(S) = (1/|S|) \sum_{i \in S} x_i$, let us define*

$$R(S) \quad \triangleq \quad \min_{S' \subset S, |S'|=(1-2\alpha)|S|.} \quad \max_{T \subset S', |T|=(1-\alpha)|S'|.} \quad \|\mu(T) - \mu(S')\|_2 \ .$$

Algorithm 2 first checks if the resilience matches that of the promised distribution. The data is pre-processed with $q_{\text{range}}$ to ensure we can check $R(S)$ privately. Once resilience is cleared, we can safely use the exponential mechanism based on the score function $d(\hat{\mu}, S)$ in Definition 4.3 to select an approximate robust mean $\hat{\mu}$ privately. The choice of the sensitivity critically relies on the fact that resilient datasets have small sensitivity of $O((1/n)\sqrt{\log(1/\alpha)})$. Without the resilience check, the sensitivity is $O(d^{1/2}/n)$ resulting in an extra factor of $\sqrt{d}$ in the sample complexity.

---

**Algorithm 2:** Exponential-time private and robust mean estimation

---

**Input:** $S = \{x_i\}_{i \in [n]}, \alpha \in (0, 1/2), (\varepsilon, \delta)$
1 **if** $n < cd^{1/2}\log(1/\delta)/(\varepsilon\alpha\sqrt{\log(1/\alpha)})$ **then Output:** $\emptyset$ [ $cd^{1/2}\log(1/\delta)/(\varepsilon\alpha)$ for hevay-tail]
2 $(\bar{x}, B) \leftarrow q_{\text{range}}(S, (1/3)\varepsilon, (1/3)\delta)$ [ $q_{\text{range-ht}}(\cdot)$ for hevay-tail]
3 Project the data points onto the ball: $x_i \leftarrow \mathcal{P}_{\mathcal{B}_{\sqrt{d}B/2}(\bar{x})}(x_i)$, for all $i \in [n]$
4 $\widehat{R}(S) \leftarrow R(S) + \text{Lap}(3Bd^{1/2}/(n\varepsilon))$
5 **if** $\widehat{R}(S) > 2\alpha\sqrt{\log(1/\alpha)}$ **then Output:** $\emptyset$ [$\widehat{R}(S) > 2c_\zeta\sqrt{\alpha}$ for hevay-tail]
6 **else Output:** a randomly drawn point $\hat{\mu} \in \mathcal{B}_{\sqrt{d}B/2}(\bar{x})$ sampled from a density

7 $\quad r(\hat{\mu}) \propto e^{-(1/(24\sqrt{\log(1/\alpha)}))\varepsilon\, n\, d(\hat{\mu},S)}$ [$e^{-(\varepsilon n\sqrt{\alpha}/(24c_\zeta))d(\hat{\mu},S)}$ for heavy-tail]

---

We propose the score function $d(\hat{\mu}, S)$ in the following definition, which is a robust estimator of the distance between the mean and the candidate $\hat{\mu}$.

**Definition 4.3.** *For a set of data $\{x_i\}_{i \in S}$ lying in $\mathbb{R}^d$, for any $v \in \mathbb{S}^{d-1}$, define $\mathcal{T}^v$ to be the $3\alpha|S|$ points with the largest $v^\top x_i$ value, $\mathcal{B}^v$ to be the $3\alpha|S|$ points with the smallest $v^\top x_i$ value, and $\mathcal{M}^v = S \setminus (\mathcal{T}^v \cup \mathcal{B}^v)$. Define $d(\hat{\mu}, S) \triangleq \max_{v \in \mathbb{S}^{d-1}} |v^\top (\mu(\mathcal{M}^v) - \hat{\mu})|$ .*

**Analysis.** For any direction $v$, the truncated mean estimator $\mu(\mathcal{M}^v)$ provides a robust estimation of the true mean along the direction $v$, thus the distance can be simply defined by taking the maximum over all directions $v$. We show the sensitivity of this simple estimator is bounded by the resilience property $\sigma$ divided by $n$, which is $O((1/n)\sqrt{\log(1/\alpha)})$ once the resilience check is passed. This leads to the following near-optimal sample complexity. We provide a proof in Appendix H.2.

**Theorem 7** (Exponential time algorithm for sub-Gaussian distributions). *Algorithm 2 is $(\varepsilon, \delta)$-DP. Under Assumption 1, this algorithm achieves $\|\hat{\mu} - \mu\|_2 = O(\alpha\sqrt{\log(1/\alpha)})$ with probability $1 - \zeta$ if*

$$n = \widetilde{\Omega}\left( \frac{d + \log\frac{1}{\zeta}}{\alpha^2 \log\frac{1}{\alpha}} + \frac{d\log\left(d\sqrt{\log(dn/\zeta)}/\alpha\right) + d^{1/2}\log\frac{1}{\delta} + \log\frac{1}{\zeta}}{\varepsilon\alpha} + \frac{\sqrt{d\log\frac{1}{\delta}}\log\frac{d}{\zeta\delta}}{\varepsilon} \right) .$$

**Run-time.** Computing $R(S)$ exactly can take $O(de^{\Theta(n)})$ operations. The exponential mechanism implemented with $\alpha$-covering for $\hat{\mu}$ and a constant covering for $v$ can take $O(nd(\sqrt{\log(dn/\zeta)}/\alpha)^d)$ operations.

## 5 Conclusion

Differentially private mean estimation is brittle against a small fraction of the samples being corrupted by an adversary. We show that robustness can be achieved without any increase in the sample complexity by introducing a novel DP mean estimator, which requires run-time exponential in the dimension of the samples. The technical contribution is in leveraging the resilience property of well-behaved distributions in an innovative way to not only find robust mean (which is the typical use case of resilience) but also bound sensitivity for optimal privacy guarantee. To cope with the computational challenge, we propose an efficient algorithm, which we call PRIME, that achieves the optimal target accuracy at the cost of an increased sample complexity. The technical contributions are

($i$) a novel framework for private iterative filtering and its tight analysis of the end-to-end sensitivity and ($ii$) novel filtering algorithm of DPTHRESHOLD which is critical in privately running matrix multiplicative weights and hence significantly reducing the number of accesses to the database. With appropriately chosen parameters, we show that our exponential time approach achieves near-optimal guarantees for both sub-Gaussian and covariance bounded distributions and PRIME achieves the same accuracy efficiently but at the cost of an increased sample complexity by a $d^{1/2}$ factor.

There are several directions for improving our results further and applying the framework to solve other problems. PRIME provides a new design principle for private and robust estimation. This can be more broadly applied to fundamental statistical analyses such as robust covariance estimation [28, 30, 64] robust PCA [60, 48], and robust linear regression [59, 35].

PRIME could be improved in a few directions. First, the sample complexity of $\widetilde{\Omega}((d/(\alpha^2 \log(1/\alpha)))+ (d^{3/2}/(\varepsilon\alpha\log(1/\alpha)))\log(1/\delta))$ in Theorem 6 is suboptimal in the second term. Improving the $d^{3/2}$ factor requires bypassing differentially private singular value decomposition, which seems to be a challenging task. However, it might be possible to separate the $\log(1/\delta)$ factor from the rest of the terms and get an additive error of the form $\widetilde{\Omega}((d/(\alpha^2 \log(1/\alpha))) + (d^{3/2}/(\varepsilon\alpha\log(1/\alpha))) + (1/\varepsilon)\log(1/\delta))$. This requires using Laplace mechanism in private MMW (line 16 Algortihm 10). Secondly, the time complexity of PRIME is dominated by computation time of the matrix exponential in (line 16 Algortihm 10). Total number of operations scale as $\widetilde{O}(d^3 + nd^2)$. One might hope to achieve $\widetilde{O}(nd)$ time complexity using approximate computations of $\tau_j$'s using techniques from [36]. This does not improve the sample complexity, as the number of times the dataset is accessed remains the same. Finally, for (non-robust) private mean estimation, COINPRESS provides a practical improvement in the small sample regime by progressively refining the search space [12]. The same principle could be applied to PRIME to design a robust version of COINPRESS. One important question remains open; how are differential privacy and robust statistics fundamentally related? We believe our exponential time algorithm hints on a fundamental connection between robust statistics of a data projected onto one-dimensional subspace and sensitivity of resulting score function for the exponential mechanism. It is an interesting direction to pursue this connection further to design novel algorithms that bridge privacy and robustness.

## Acknowledgement

Sham Kakade acknowledges funding from the National Science Foundation under award CCF-1703574. Sewoong Oh acknowledges funding from Google faculty research award, NSF grants IIS-1929955, CCF-1705007, CNS-2002664, CCF 2019844 as a part of Institute for Foundation of Machine Learning, and CNS-2112471 as a part of Institute for Future Edge Networks and Distributed Intelligence.

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
