# Appendix

## Table of Contents

# A   Related work

**Private statistical analysis.** Traditional private data analyses require bounded support of the samples to leverage the resulting bounded sensitivity. For example, each entry is constrained to have finite $\ell_2$ norm in standard private principal component analysis [18], which does not apply to Gaussian samples. Fundamentally departing from these approaches, [58] first established an optimal mean estimation of Gaussian samples with *unbounded* support. The breakthrough is in first adaptively estimating the range of the data using a private histogram, thus bounding the support and the resulting sensitivity. This spurred the design of private algorithms for high-dimensional mean and covariance estimation [52, 12], heavy-tailed mean estimation [54], learning mixture of Gaussian [53], learning Markov random fields [80], and statistical testing [16]. Under the Gaussian distribution with no adversary, [3] achieves an accuracy of $\|\hat{\mu} - \mu\|_2 \leq \tilde{\alpha}$ with the best known sample complexity of $n = \widetilde{O}((d/\tilde{\alpha}^2) + (d/\tilde{\alpha}\varepsilon) + (1/\varepsilon)\log(1/\delta))$ while guaranteeing $(\varepsilon, \delta)$-differential privacy. This nearly matches the known lower bounds of $\Omega(d/\tilde{\alpha}^2)$ for non-private finite sample complexity, $\widetilde{\Omega}((1/\varepsilon)\log(1/\delta))$ for privately learning one-dimensional unit variance Gaussian [58], and $\widetilde{\Omega}(d/\tilde{\alpha}\varepsilon)$ for multi-dimensional Gaussian estimation [52]. However, this does not generalize to sub-Gaussian distributions and [3] does not provide a tractable algorithm. A polynomial time algorithm is proposed in [52] that achieves a slightly worse sample complexity of $\widetilde{O}((d/\tilde{\alpha}^2) + (d\log^{1/2}(1/\delta)/\tilde{\alpha}\varepsilon))$, which can also seamlessly generalized to sub-Gaussian distributions.

[15] takes a different approach of deviating from standard definition of sub-Gaussianity to provide a larger lower bound on the sample complexity scaling as $n = \Omega(d\sqrt{\log(1/\delta)}/(\alpha\varepsilon))$ for mean estimation with a known covariance. Concretely, they consider distributions satisfying $\mathbb{E}_{x \sim P}[e^{\lambda\langle x-\mu, e_k\rangle}] \leq e^{\lambda^2\sigma^2}$ for all $k \in [d]$ where $e_k$ is the $k$-th standard basis vector. Notice that this condition only requires sub-Gaussianity when projected onto standard bases. Standard definition of high-dimensional sub-Gaussianity (which is assumed in this paper) requires sub-Gaussianity in all directions. Therefore, their lower bound is not comparable with our achievable upper bounds. Further, the example they construct to show the lower bound does not satisfy our sub-Gaussianity assumptions.

In an attempt to design efficient algorithms for robust and private mean estimation, [26] proposed an algorithm with a mis-calculated sensitivity, which can result in violating the privacy guarantee. This can be corrected by pre-processing with our approach of checking the resilience (as in Algorithm 2), but this requires a run-time exponential in the dimension.

For estimating the mean of a *covariance bounded* distributions up to an error of $\|\hat{\mu} - \mu\|_2 = O(\tilde{\alpha}^{1/2})$, [54] shows that $\Omega(d/(\tilde{\alpha}\varepsilon))$ samples are necessary and provides an efficient algorithm matching this up to a factor of $\log^{1/2}(1/\delta)$. For a more general family of distributions with bounded $k$-moment, [54] shows that an error of $\|\hat{\mu} - \mu\|_2 = O(\tilde{\alpha}^{(k-1)/k})$ can be achieved with $n = \widetilde{O}((d/\tilde{\alpha}^{2(k-1)/k}) + (d\log^{1/2}(1/\delta)/(\varepsilon\tilde{\alpha})))$ samples.

However, under $\alpha$-corruption, [46] shows that achieving an error better than $O(\alpha^{1/2})$ under $k$-th moment bound is as computationally hard as the small-set expansion problem, even without requiring DP. Hence, under the assumption of $P \neq NP$, no polynomial-time algorithm exists that can outperform our PRIME-HT even if we have stronger assumptions of $k$-th moment bound. On the other hand, there exists an exponential time algorithm for non-private robust mean estimation that achieves $\|\mu - \hat{\mu}\|_2 = O(\alpha^{(k-1)/k})$ [81]. Combining it with the bound of [46], an interesting open question is whether there is an (exponential time) algorithm that achieves $\|\mu - \hat{\mu}\|_2 = O(\alpha^{(k-1)/k})$ with sample complexity $n = \widetilde{O}((d/\alpha^{2(k-1)/k}) + (d\log^{1/2}(1/\delta)/(\varepsilon\alpha)))$ under $\alpha$-corruption and $(\varepsilon, \delta)$-DP.

**Robust estimation.** Designing robust estimators under the presence of outliers has been considered by statistics community since 1960s [76, 6, 47]. Recently, [28, 62] give the first polynomial time algorithm for mean and covariance estimation with no (or very weak) dependency on the dimensionality in the estimation error. Since then, there has been a flurry of research on robust estimation problems, including mean estimation [30, 36, 43, 44, 31], covariance estimation [21, 64], linear regression and sparse regression [11, 10, 8, 42, 71, 59, 29, 65, 57, 23, 70, 35, 56], principal component analysis [60, 48], mixture models [27, 49, 61, 45] and list-decodable learning [34, 72, 17, 7, 22]. See [32] for a survey of recent work.

One line of work that is particularly related to our algorithm PRIME is [20, 36, 24, 21, 22], which leverage the ideas from matrix multiplicative weight and fast SDP solver to achieve faster, sometimes nearly linear time, algorithms for mean and covariance estimation. In PRIME, we use a matrix multiplicative weight approach similar to [36] to reduce the iteration complexity to logarithmic, which enables us to achieve the $d^{3/2}$ dependency in the sample complexity.

The concept of *resilience* is introduced in [73] as a sufficient condition such that learning in the presence of adversarial corruption is information-theoretically possible. The idea of resilience is later generalized in [81] for a wider range of adversarial corruption models. While there exists simple exponential time robust estimation algorithm under resilience condition, it is challenging to achieve differential privacy due to high sensitivity. We propose a novel approach to leverage the resilience property in our exponential time algorithm for sub-gaussian and heavy-tailed distributions.

## B  Main results under heavy-tailed distributions

We consider distributions with bounded covariance as defined as follows.

**Assumption 2.** *An uncorrupted dataset $S_{\text{good}}$ consists of $n$ i.i.d. samples from a distribution with mean $\mu \in \mathbb{R}^d$ and covariance $\Sigma \preceq \mathbf{I}$. For some $\alpha \in (0, 1/2)$, we are given a corrupted dataset $S = \{x_i\}_{i=1}^n$ where an adversary adaptively inspects all samples in $S_{\text{good}}$, removes $\alpha n$ of them and replaces them with $S_{\text{bad}}$ that are $\alpha n$ arbitrary points in $\mathbb{R}^d$.*

Under these assumptions, Algorithm 2 achieves near optimal guarantees but takes exponential time. The dominant term in the sample complexity $\widetilde{\Omega}(d/(\varepsilon\alpha))$ cannot be improved as it matches that of the optimal non-robust private estimation [54]. The accuracy $O(\sqrt{\alpha})$ cannot be improved as it matches that of the optimal non-private robust estimation [36]. We provide a proof in Appendix H.1.

**Theorem 8** (Exponential time algorithm for covariance bounded distributions). *Algorithm 2 is $(\varepsilon, \delta)$-differentially private. Under Assumption 2, if*

$$n = \Omega\Big(\frac{d\log(d/\alpha^{1.5}) + d^{1/2}\log(1/\delta)}{\varepsilon\alpha} + \frac{d^{1/2}\log^{3/2}(1/\delta)log(d/\delta)}{\varepsilon}\Big),$$

*this algorithm achieves $\|\hat{\mu} - \mu\|_2 = O(\sqrt{\alpha})$ with probability $0.9$.*

We propose an efficient algorithm PRIME-HT and show that it achieves the same optimal accuracy but at the cost of increased sample complexity of $O(d^{3/2}\log(1/\delta)/(\varepsilon\alpha))$. In the first step, we need increase the radius of the ball to $O(\sqrt{d/\alpha})$ to include a $1 - \alpha$ fraction of the clean samples, where $q_{\text{range-ht}}$ returns $B = O(1/\sqrt{\alpha})$ and $\mathcal{B}_{\sqrt{d}B/2}(\bar{x})$ is a $\ell_2$-ball of radius $\sqrt{d}B/2$ centered at $\bar{x}$. This is followed by a matrix multiplicative weight filter similar to DPMMWFILTERR but the parameter choices are tailored for covariance bounded distributions. We provide a proof in Appendix K.2.

**Theorem 9** (Efficient algorithm for covariance bounded distributions). *PRIME-HT is $(\varepsilon, \delta)$-differentially private. Under Assumption 2 there exists a universal constant $c \in (0, 0.1)$ such that if $\alpha \leq c$, and $n = \widetilde{\Omega}((d^{3/2}/(\varepsilon\alpha))\log(1/\delta))$, then PRIME-HT achieves $\|\hat{\mu} - \mu\|_2 = O(\alpha^{1/2})$ with probability $0.9$. The notation $\widetilde{\Omega}(\cdot)$ hides logarithmic terms in $d$, and $1/\alpha$.*

*Remark 1.* To boost the success probability to $1 - \zeta$ for some small $\zeta > 0$, we will randomly split the data into $O(\log(1/\zeta))$ subsets of equal sizes, and run Algorithm 3 to obtain a mean estimation from each of the subset. Then we can apply multivariate "mean-of-means" type estimator [68] to get $\|\hat{\mu} - \mu\|_2 = O(\alpha^{1/2})$ with probability $1 - \zeta$. This is efficient as we only have $O(\log 1/\zeta)$ trials and run-time of mean-of-means is dominated by the time it takes to find all pairwise distances, which is only $O(d(\log(1/\zeta))^2)$. There are $(\log(1/\zeta))^2$ pairs, and for each pair we compute the distance between means in $d$ operations.

---

**Algorithm 3:** PRIvate and robust Mean Estimation for covariance bounded distributions (PRIME-HT)

---

**Input:** $S = \{x_i \in \mathbb{R}^d\}_{i=1}^n$, adversarial fraction $\alpha \in (0, 1/2)$, number of iterations
$\quad\quad T_1 = O(\log(d/\alpha)), T_2 = O(\log d)$, target privacy $(\varepsilon, \delta)$

1 $(\bar{x}, B) \leftarrow q_{\text{range-ht}}(S, 0.01\varepsilon, 0.01\delta)$        [Algorithm 14 in Appendix K]
2 Project the data onto the ball: $\tilde{x}_i \leftarrow \mathcal{P}_{\mathcal{B}_{\sqrt{d}B/2}(\bar{x})}(x_i)$, for all $i \in [n]$

3 $\hat{\mu} \leftarrow \text{DPMMWFILTER-HT}(\{\tilde{x}_i\}_{i=1}^n, \alpha, T_1, T_2, 0.99\varepsilon, 0.99\delta)$    [Algorithm 15 in Appendix K]
**Output:** $\hat{\mu}$

---

## C    Background on (non-private) robust mean estimation

The following tie-breaking rule is not essential for robust estimation, but is critical for proving differential privacy, as shown later in Appendix F.1.

**Definition C.1** (Subset of the largest $\alpha$ fraction)**.** *Given a set of scalar values $\{\tau_i = \langle V, (x_i - \mu)(x_i - \mu)^\top \rangle\}_{i \in S'}$ for a subset $S' \subseteq [n]$, define the sorted list $\pi$ of $S'$ such that $\tau_{\pi(i)} \geq \tau_{\pi(i+1)}$ for all $i \in [|S'| - 1]$. When there is a tie such that $\tau_i = \tau_j$, it is broken by $\pi^{-1}(i) \leq \pi^{-1}(j) \Leftrightarrow x_{i,1} \geq x_{j,1}$. Further ties are broken by comparing the remaining entries of $x_i$ and $x_j$, in an increasing order of the coordinate. If $x_i = x_j$, then the tie is broken arbitrarily. We define $\mathcal{T}_\alpha = \{\pi(1), \ldots, \pi(\lceil n\alpha \rceil)\}$ to be the set of largest $\lceil n\alpha \rceil$ valued samples.*

With this definition of $\alpha$-tail, we can now provide a complete description of the robust mean estimation that achieves the guarantee provided in Proposition 2.1.

---

**Algorithm 4:**   Non-private robust mean estimation [63]

---

**Input:** $S = \{x_i\}_{i=1}^n$, $\alpha \in (0, 1)$, $S_0 = [n]$

1 **for** $t = 1, \ldots$ **do**
2     **if** $\|\sum_{i \in S_{t-1}}(x_i - \mu_{t-1})(x_i - \mu_{t-1})^\top - \mathbf{I}\|_2 < C\alpha \log(1/\alpha)$ **then**
       $\lfloor$ **Output:** $\hat{\mu} = \sum_{i \in S_{t-1}} x_i$
3     **else**
4        $\mu_t \leftarrow (1/|S_{t-1}|)\sum_{i \in S_{t-1}} x_i$
5        $v_t \leftarrow$ 1st principal direction of $(\{(x_i - \mu_t)\}_{i \in S_{t-1}})$
6        $Z_t \leftarrow \text{Unif}([0, 1])$
7        $S_t \leftarrow S_{t-1} \setminus \{i \,|\, i \in \mathcal{T}_{2\alpha} \text{ for } \{\tau_j = (v_t^\top(x_j - \mu_t))^2\}_{j \in S_{t-1}}$ and
          $\tau_i \geq Z_t \max_{j \in S_{t-1}}(v_t^\top(x_j - \mu_t))^2\}$, where $\mathcal{T}_{2\alpha}$ is defined in Definition C.1.

---

## D    A new framework for *private* iterative filtering

We provide complete descriptions of all algorithms used in private iterative filtering. We present the *interactive* version first, followed by the *centralized* version.

### D.1   Interactive version of the algorithm

Adaptive estimation of the range of the dataset is essential in computing private statistics of data. We use the following algorithm proposed in [58]. It computes a private histogram of a set of 1-dimensional points and select the largest bin as the one potentially containing the mean of the data. Note that $B$ does not need not be chosen adaptively to include all the uncorrupted data with a high probability.

The following guarantee (and the algorithm description) is used in the analysis (and the implementation) of the query $q_{\text{range}}$.

**Lemma D.1** (Histogram Learner, Lemma 2.3 in [58])**.** *For every $K \in \mathbb{N} \cup \infty$, domain $\Omega$, for every collection of disjoint bins $B_1, \ldots, B_K$ defined on $\Omega$, $n \in \mathbb{N}$, $\varepsilon, \delta \in (0, 1/n)$, $\beta > 0$ and $\alpha \in (0, 1)$*

**Algorithm 5:** Differentially private range estimation ($q_{\text{range}}$) [58, Algorithm 1]

**Input:** $\mathcal{D}_n = \{x_i\}_{i=1}^n$, $\varepsilon$, $\delta$, $\sigma = 1$

1 **for** $j \leftarrow 1$ **to** $d$ **do**

2 $\quad R_{\max}^{(j)} \leftarrow \max_{i \in [n]} x_i^{(j)}$ and $R_{\min}^{(j)} \leftarrow \min_{i \in [n]} x_i^{(j)}$ where $x_i^{(j)}$ is the $j$-th coordinate of $x_i$

3 $\quad$ Run the histogram learner of Lemma D.1 with privacy parameters

$\quad \left( \min\{\varepsilon, 0.9\}/2\sqrt{2d \log(2/\delta)}, \delta/(2d) \right)$ and bins $B_l = (2\sigma \ell, 2\sigma(\ell+1)]$ for all

$\quad \ell \in \{\lceil R_{\min}^{(j)}/2\sigma \rceil - 1, \ldots, \lceil R_{\max}^{(j)}/2\sigma \rceil\}$ on input $\mathcal{D}_n$ to obtain noisy estimates

$\quad \{\tilde{h}_{j,l}\}_{l = \lceil R_{\min}^{(j)}/2\sigma \rceil - 1}^{\lceil R_{\max}^{(j)}/2\sigma \rceil}$

4 $\quad \bar{x}_j \leftarrow 2\sigma \cdot \arg\max_{\ell \in \{\lceil R_{\min}^{(j)}/2\sigma \rceil - 1, \ldots, \lceil R_{\max}^{(j)}/2\sigma \rceil\}} \tilde{h}_{j,\ell}$

**Output:** $(\bar{x}, B = 8\sigma\sqrt{\log(dn/\zeta)})$

---

*there exists an $(\varepsilon, \delta)$-differentially private algorithm $M : \Omega^n \to \mathbb{R}^K$ such that for any set of data $X_1, \ldots, X_n \in \Omega^n$*

1. $\hat{p}_k = \frac{1}{n} \sum_{X_i \in B_k} 1$

2. $(\tilde{p}_1, \ldots, \tilde{p}_K) \leftarrow M(X_1, \ldots, X_n)$, *and*

3.

$$n \geq \min \left\{ \frac{8}{\varepsilon\beta} \log(2K/\alpha), \frac{8}{\varepsilon\beta} \log(4/\alpha\delta) \right\}$$

*then,*

$$\mathbb{P}(|\tilde{p}_k - \hat{p}_k| \leq \beta) \geq 1 - \alpha$$

*Proof.* This is an intermediate result in the proof of Lemma 2.3 in [58]. Note that, conceptually, we are applying the private histogram algorithm to an infinite number of bins in the intervals $\{\cdots, (-4\sigma, -2\sigma], (-2\sigma, 0], (0, 2\sigma], (2\sigma, 4\sigma], (4\sigma, 6\sigma] \cdots\}$ each of length $2\sigma$. This is possible because the algorithm only changes the bins that are occupied by at least on sample. Practically, we only need to add noise to those bins that are occupied, and hence we limit the range from $R_{\min}^{(j)}$ to $R_{\max}^{(j)}$ without loss of generality and without any changes to the privacy guarantee of the algorithm. $\qquad \square$

---

The rest of the queries ($q_{\text{size}}$, $q_{\text{mean}}$, $q_{\text{PCA}}$, and $q_{\text{norm}}$) are provided below. The most innovative part is the repeated application of filtering that is run every time one of the queries is called. In the Filter query below, because we choose $(i)$ to use the sampling version of robust mean estimation as opposed to weighting version which assigned a weight on each sample between zero and one measuring how good (i.e., score one) or bad (i.e., score zero) each sample point is, and $(ii)$ we switched the threshold to be $dB^2 Z_\ell$, we can show that this filtering with fixed parameters $\{\mu_\ell, v_\ell, Z_\ell\}_{\ell \in [t-1]}$ preserves sensitivity in Lemma 2.2. This justifies the choice of noise in each output perturbation mechanism, satisfying the desired level of $(\varepsilon, \delta)$-DP. We provide the complete privacy analysis in Appendix D.3 and also the analysis of the utility of the algorithm as measure by the accuracy.

### D.2 Centralized version of the algorithm

In practice, one should run the centralized version of the private iterative filtering, in order to avoid multiple redundant computations of the interactive version. The main difference is that the redundant

---

**Algorithm 6:** Interactive private queries used in Algorithm 1

---

1 Filter $(\{(\mu_\ell, v_\ell, Z_\ell)\}_{\ell \in [t-1]}, \bar{x}, B)$**:**
2     $S_0 \leftarrow [n]$
3     Clip the data points: $x_i \leftarrow \mathcal{P}_{\bar{x}+[-B/2, B/2]^d}(x_i)$, for all $i \in [n]$
4     **for** $\ell = 1, \ldots, t-1$ **do**
5        $S_\ell \leftarrow S_{\ell-1} \setminus \{i \in S_{\ell-1} : i \in \mathcal{T}_{2\alpha} \text{ for } \{\tau_j = (v_\ell^\top (x_j - \mu_\ell))^2\}_{j \in S_{\ell-1}} \text{ and } \tau_i \geq d\,B^2\,Z_\ell\}$

6 $q_{\text{mean}}(\{(\mu_\ell, v_\ell, Z_\ell)\}_{\ell \in [t-1]}, \varepsilon, \bar{x}, B)$**:**
7     Filter$(\{(\mu_\ell, v_\ell, Z_\ell)\}_{\ell \in [t-1]}, \bar{x}, B)$
8     **return** $\mu_t \leftarrow (1/|S_{t-1}|)\left(\sum_{i \in S_{t-1}} x_i\right) + \text{Lap}(2B/(n\varepsilon))$

9 $q_{\text{PCA}}(\{(\mu_\ell, v_\ell, Z_\ell)\}_{\ell \in [t-1]}, \mu_t, \varepsilon, \delta, \bar{x}, B)$**:**
10     Filter$(\{(\mu_\ell, v_\ell, Z_\ell)\}_{\ell \in [t-1]}, \bar{x}, B)$
11     **return** $v_t \leftarrow$ top singular vector of $\Sigma_{t-1} =$
12     $(1/n) \sum_{i \in S_{t-1}} (x_i - \mu_t)(x_i - \mu_t)^\top + \mathcal{N}(0, (B^2 d \sqrt{2\log(1.25/\delta)}/(n\varepsilon))^2 \mathbf{I}_{d^2 \times d^2})$

13 $q_{\text{norm}}(\{(\mu_\ell, v_\ell, Z_\ell)\}_{\ell \in [t-1]}, \mu_t, \varepsilon, \bar{x}, B)$**:**
14     Filter$(\{(\mu_\ell, v_\ell, Z_\ell)\}_{\ell \in [t-1]}, \bar{x}, B)$
15     **return** $\lambda_t \leftarrow \|(1/n) \sum_{i \in S_{t-1}} (x_i - \mu_t)(x_i - \mu_t)^\top\|_2 + \text{Lap}(2B^2 d/(n\varepsilon))$

16 $q_{\text{size}}(\{(\mu_\ell, v_\ell, Z_\ell)\}_{\ell \in [t-1]}, \varepsilon, \bar{x}, B)$**:**
17     Filter$(\{(\mu_\ell, v_\ell, Z_\ell)\}_{\ell \in [t-1]}, \bar{x}, B)$
18     **return** $n_t \leftarrow |S_{t-1}| + \text{Lap}(1/\varepsilon)$

---

filtering repeated every time a query is called in the interactive version is now merged into a single run. The resulting estimation and the privacy loss are exactly the same.

---

**Algorithm 7:** Private iterative filtering (centralized version)

---

**Input:** $S = \{x_i \in \mathbb{R}^d\}_{i=1}^n$, adversarial fraction $\alpha \in (0,1)$, target probability $\eta \in (0,1)$, number
       of iterations $T = \widetilde{\Theta}(d)$, target privacy $(\varepsilon, \delta)$
1 $(\bar{x}, B) \leftarrow q_{\text{range}}(S, 0.01\varepsilon, 0.01\delta)$                               [Algorithm 5]
2 Clip the data points: $\tilde{x}_i \leftarrow \mathcal{P}_{\bar{x}+[-B/2, B/2]^d}(x_i)$, for all $i \in [n]$
3 $\hat{\mu} \leftarrow \text{DPFILTER}(\{\tilde{x}_i\}_{i=1}^n, \alpha, T, 0.99\varepsilon, 0.99\delta)$                     [Algorithm 8]
**Output:** $\hat{\mu}$

---

First, $q_{\text{range}}$ introduced in [58], returns a hypercube $\bar{x} + [-B, B]^d$ that is guaranteed to include all uncorrupted samples, while preserving privacy. It is followed by a private filtering DPFILTER in Algorithm 8.

## D.3    The analysis of private iterative filtering (Algorithms 1 and 7) and a proof of Theorem 5

$q_{\text{range}}$, introduced in [58], returns a hypercube $\bar{x} + [-B, B]^d$ that is guaranteed to include all uncorrupted samples, while preserving privacy. In the following lemma, we show that $q_{\text{range}}$ is also *robust* to adversarial corruption. Such adaptive bounding of the support is critical in privacy analysis of the subsequent steps. We clip all data points by projecting all the points with $\mathcal{P}_{\bar{x}+[-B/2, B/2]^d}(x) = \arg\min_{y \in \bar{x}+[-B/2, B/2]^d} \|y - x\|_2$ to lie inside the hypercube and pass them to DPFILTER for filtering. The algorithm and a proof are provided in §D.3.1.

**Lemma D.2.** $q_{\text{range}}(S, \varepsilon, \delta)$ *(Algorithm 5) is $(\varepsilon, \delta)$-differentially private. Under Assumption 1, $q_{\text{range}}(S, \varepsilon, \delta)$ returns $(\bar{x}, B)$ such that if $n = \Omega\left((\sqrt{d \log(1/\delta)} \log(d/(\zeta\delta))/\varepsilon)\right)$ and $\alpha < 0.1$, then all uncorrupted samples in $S$ are in $\bar{x} + [-B, B]^d$ with probability $1 - \zeta$.*

In DPFILTER, we make only the mean $\mu_t$ and the top principal direction $v_t$ private to decrease sensitivity. The analysis is now more challenging since $(\mu_t, v_t)$ depends on all past iterates $\{(\mu_j, v_j)\}_{j=1}^{t-1}$ and internal randomness $\{Z_j\}_{j=1}^{t-1}$. To decrease the sensitivity, we modify the filter in line 12 to

---

**Algorithm 8:** Differentially private filtering (DPFILTER)

---

**Input:** $S = \{x_i \in \bar{x} + [-B/2, B/2]^d\}_{i=1}^n, \alpha \in (0, 1/2),$
$\qquad T = \widetilde{O}(dB^2 \log(dB^2/(\alpha \log(1/\alpha)))), (\varepsilon, \delta)$

1   $S_0 \leftarrow [n], \varepsilon_1 \leftarrow \min\{\varepsilon, 0.9\}/(4\sqrt{2T \log(2/\delta)}), \delta_1 \leftarrow \delta/(8T)$
2   **if** $n < (4/\varepsilon_1) \log(1/(2\delta_1))$ **then Output:** $\emptyset$
3   **for** $t = 1, \ldots, T$ **do**
4      $n_t \leftarrow |S_{t-1}| + \text{Lap}(1/\varepsilon_1)$
5      **if** $n_t < 3n/4$ **then**
6        **Output:** $\emptyset$
7      $\mu_t \leftarrow (1/|S_{t-1}|) \sum_{i \in S_{t-1}} x_i + \text{Lap}(2B/(n\varepsilon_1))$
8      $\lambda_t \leftarrow \|(1/n) \sum_{i \in S_{t-1}} (x_i - \mu_t)(x_i - \mu_t)^\top - \mathbf{I}\|_2 + \text{Lap}(2B^2 d/(n\varepsilon_1))$
9      **if** $\lambda_t \leq (C - 0.01)\alpha \log(1/\alpha)$ **then**
       **Output:** $\mu_t$
10     $v_t \leftarrow$ top singular vector of $\Sigma_{t-1} \triangleq$
       $\frac{1}{n} \sum_{i \in S_{t-1}} (x_i - \mu_t)(x_i - \mu_t)^\top + \mathcal{N}(0, (B^2 d\sqrt{2\log(1.25/\delta)}/(n\varepsilon_1))^2 \mathbf{I}_{d^2 \times d^2})$
11     $Z_t \leftarrow \text{Unif}([0, 1])$
12     $S_t \leftarrow S_{t-1} \setminus \{i \mid i \in \mathcal{T}_{2\alpha} \text{ for } \{\tau_j = (v_t^\top(x_j - \mu_t))^2\}_{j \in S_{t-1}} \text{ and } \tau_i \geq d\, B^2\, Z_t\}$, where $\mathcal{T}_{2\alpha}$
       is defined in Definition C.1.

---

use the maximum support $dB^2$ (which is data independent) instead of the maximum contribution $\max_i (v_t^\top (x_i - \mu_t))^2$ (which is data dependent and sensitive). While one data point can significantly change $\max_i (v_t^\top (x_i - \mu_t))^2$ and the output of one step of the filter in Algorithm 4, the sensitivity of the proposed filter is bounded conditioned on all past $\{(\mu_j, v_j)\}_{j=1}^{t-1}$, as we show in the following lemma. This follows from the fact that conditioned on $(\mu_j, v_j)$, the proposed filter is a contraction. We provide a proof in Appendix D.3.3 and Appendix D.3.4. Putting together Lemmas D.2 and D.3, we get the desired result in Theorem 5.

**Lemma D.3.** DPFILTER$(S, \alpha, T, \varepsilon, \delta)$ *is* $(\varepsilon, \delta)$-*differentially private. Under the hypotheses of Theorem 5,* DPFILTER$(S, \alpha, T = \widetilde{\Theta}(B^2 d), \varepsilon, \delta)$ *achieves* $\|\hat{\mu} - \mu\|_2 = O(\alpha\sqrt{\log(1/\alpha)})$ *with probability* $0.9$, *if* $n = \widetilde{\Omega}(d/\alpha^2 + B^3 d^2 \log(1/\delta)/(\varepsilon\alpha))$ *and* $B$ *is large enough such that the original uncorrupted samples are inside the hypercube* $\bar{x} + [-B/2, B/2]^d$.

**Differential privacy guarantee.** To achieve $(\varepsilon_0, \delta_0)$ end-to-end target privacy guarantee, Algorithm 7 separates the privacy budget into two. The $(0.01\varepsilon_0, 0.01\delta_0)$-DP guarantee of $q_{\text{range}}$ follows from Lemma D.2. The $(0.99\varepsilon_0, 0.99\delta_0)$-DP guarantee of DPFILTER follows from Lemma D.3.

**Accuracy.** From Lemma D.2 $q_{\text{range}}$ is guaranteed to return a hypercube that includes all clean data in the dataset. It follows from Lemma D.3 that when $n = \widetilde{\Omega}(d/\alpha^2 + d^2 \log(1/\delta)/(\varepsilon\alpha))$, we have $\|\mu - \hat{\mu}\|_2 = O(\alpha\sqrt{\log(1/\alpha)})$.

### D.3.1   Proof of Lemma D.2 and the analysis of $q_{\text{range}}$ in Algorithm 5

Assuming the distribution is $\sigma^2$ sub-Gaussian, we use $\mathcal{P}$ to denote the sub-Gaussian distribution. Denote $I_l = [2\sigma l, 2\sigma(l+1)]$ as the interval of the $l$'th bin. Denote the population probability in the $l$'th bin $h_{j,l} = \mathbb{P}_{x \sim \mathcal{P}}[x_j \in I_l]$, empirical probability in the $l$'th bin $\tilde{h}_{j,l} = \frac{1}{n} \sum_{x_i \in \mathcal{D}} \mathbf{1}\{x_{i,j} \in I_l\}$, and the noisy version $\hat{h}_{j,l}$ computed by the histogram learner of Lemma D.1. Notice that Lemma D.1 with $d$ compositions (Lemma G.13) immediately implies that our algorithm is $(\varepsilon, \delta)$-differentially private.

For the utility of the algorithm, we will first show that for all dimension $j \in [d]$, the output $|\bar{x}_j - \mu_j| = O(\sigma)$. Note that by the definition of $\sigma^2$-subgaussian, it holds that for all $i \in [d]$, $\mathbb{P}[|x_i - \mu_i| \geq z] \leq 2\exp(-z^2/\sigma^2)$ where $x$ is drawn from distribution $\mathcal{P}$. This implies that $\mathbb{P}[|x_i - \mu_i| \geq 2\sigma] \leq 2\exp(-4) \leq 0.04$. Suppose the $k$'th bin contains $\mu_j$, namely $\mu_j \in I_k$. Then

it is clear that $[\mu_j - 2\sigma, \mu_j + 2\sigma] \subset (I_{k-1} \cup I_k \cup I_{k+1})$. This implies $h_{j,k-1} + h_{j,k} + h_{j,k+1} \geq 1 - 0.04 = 0.96$, hence $\min(h_{j,k-1}, h_{j,k}, h_{j,k+1}) \geq 0.32$.

Recall that $\mathcal{G}$ is the set of clean data drawn from distribution $P$. By Dvoretzky-Kiefer-Wolfowitz inequality and an union bound over $j \in [d]$, we have that with probability $1 - \zeta$, $\max_{j,l}(|h_{j,l} - \frac{1}{n}\sum_{x \in G} x_j|) \leq \sqrt{\frac{\log(d/\zeta)}{n}}$. The deviation due to corruption is at most $\alpha$ on each bin, hence we have $\max_{j,l}(|h_{j,l} - \hat{h}_{j,l}|) \leq \sqrt{\frac{\log(d/\zeta)}{n}} + \alpha$. Lemma D.1 and a union bound over $j \in [d]$ implies that with probability $1 - \zeta$, $\max_{j,l}(|\tilde{h}_{j,l} - \hat{h}_{j,l}|) \leq \beta$ when $n \geq \Omega\left(\frac{\sqrt{d\log(1/\delta)}}{\varepsilon\beta}\log(d/\zeta\delta)\right)$.

Assuming that $n = \Omega\left(\frac{\sqrt{d\log(1/\delta)}}{\varepsilon\beta}\log(d/\zeta\delta)\right)$, we have that with probability $1 - \zeta$, $\max_{j,l}(|h_{j,l} - \hat{h}_{j,l}|) \leq 0.01 + \alpha$. Using the assumption that $\alpha \leq 0.1$, since $\min(h_{j,k-1}, h_{j,k}, h_{j,k+1}) - 0.11 \geq 0.31 \geq 0.04 + 0.11 \geq \max_{l \neq k-1,k,k+1} h_{j,l} + 0.11$. This implies that with probability $1 - \zeta$, the algorithm choose the bin from $k-1, k, k+1$, which means the estimate $|\bar{x}_j - \mu| \leq 4\sigma$. By the tail bound of sub-Gaussian distribution and a union bound over $n, d$, we have that with probability $1 - \zeta$, for all $x_i \in \mathcal{D}$ and $j \in [d]$, $x_{i,j} \in [\bar{x}_j - 8\sigma\sqrt{\log(nd/\zeta)}, \bar{x}_j + 8\sigma\sqrt{\log(nd/\zeta)}]$.

### D.3.2 Proofs of the sensitivity of the filtering in Lemma 2.2 and Lemma F.1

**Proof of Lemma 2.2.** We only need to show that one step of the proposed filter is a contraction. To this end, we only need to show contraction for two datasets at distance 1, i.e., $d_\triangle(\mathcal{D}, \mathcal{D}') = 1$. For fixed $(\mu, v)$ and $Z$, we apply filter to set of scalars $(v^\top(\mathcal{D} - \mu))^2$ and $(v^\top(\mathcal{D}' - \mu))^2$, whose distance is also one. If the entries that are different (say $a \in \mathcal{D}$ and $a' \in \mathcal{D}'$) are both below the subset of the top $2n\alpha$ points (as in Definition C.1), then the same set of points will be removed for both and the distance is preserved $d_\triangle(S(\mathcal{D}), S(\mathcal{D}')) = 1$. If they are both above the top $2n\alpha$ subset, then either both are removed, one of them is removed, or both remain. The rest of the points that are removed coincide in both sets. Hence, $d_\triangle(S(\mathcal{D}), S(\mathcal{D}')) \leq 1$. If $a$ is below and $a'$ is above the top $2n\alpha$ subset of respective datasets, then either $a'$ is not removed (in which case $d_\triangle(S(\mathcal{D}), S(\mathcal{D}')) = 1$) or $a'$ is removed (in which case $S(\mathcal{D}) = S(\mathcal{D}') \cup \{a\}$ and the distance remains one).

Note that when there are ties, it is critical to resolve them in a consistent manner in both datasets $\mathcal{D}$ and $\mathcal{D}'$. The tie breaking rule of Definition C.1 is critical in sorting those samples with the same score $\tau_i$'s in a consistent manner.

**Proof of Lemma F.1.** The analysis of contraction of the filtering step in DPMMWFILTER is analogous to that of private iterative filtering in Lemma 2.2.

### D.3.3 Proof of part 1 of Lemma D.3 on differential privacy of DPFILTER

We explicitly write out how many times we access the database and how much privacy is lost each time in an interactive version of DPFILTER in Algorithm 1, which performs the same operations as DPFILTER. In order to apply Lemma G.13, we cap $\varepsilon$ at 0.9 in initializing $\varepsilon_1$. We call $q_{\text{mean}}, q_{\text{PCA}}, q_{\text{norm}}$ and $q_{\text{size}}$ $T$ times, each with $(\varepsilon_1, \delta_1)$ guarantee. In total this accounts for $(\varepsilon, \delta)$ privacy loss, using Lemma G.13 and our choice of $\varepsilon_1$ and $\delta_1$.

This proof is analogous to the proof of DP for DPMMWFILTER in Appendix F.1, and we omit the details here. We will assume for now that $|S_r| \geq n/2$ for all $r \in [t]$ and prove privacy. This happens with probability larger than $1 - \delta_1$, hence ensuring the privacy guarantee. In all sub-routines, we run Filter($\cdot$) in Algorithm 1 to simulate the filtering process so far and get the current set of samples $S_t$. Lemma 2.2 allows us to prove privacy of all interactive mechanisms. This shows that the two data datasets $S_t$ and $S_t'$ are neighboring, if they are resulting from the identical filtering but starting from two neighboring datasets $\mathcal{D}_n$ and $\mathcal{D}_n'$. As all four sub-routines are output perturbation mechanisms with appropriately chosen sensitivities, they satisfy the desired $(\varepsilon_1, \delta_1)$-DP guarantees. Further, the probability that $n_t > 3/4n$ and $|S_t| \leq n/2$ is less than $\delta_1$ for $n = \tilde{\Omega}((1/\varepsilon_1)\log(1/\delta_1))$.

### D.3.4 Proof of part 2 of Lemma D.3 on accuracy of DPFILTER

The following theorem analyzing DPFILTER implies the desired Lemma D.3 when the good set is $\alpha$-subgaussian good, which follows from G.3 and the assumption that $n = \widetilde{\Omega}(d/\alpha^2)$.

**Theorem 10** (Anlaysis of DPFILTER). *Let $S$ be an $\alpha$-corrupted sub-Gaussian dataset under Assumption 1, where $\alpha \leq c$ for some universal constant $c \in (0, 1/2)$. Let $S_{\text{good}}$ be $\alpha$-subgaussian good with respect to $\mu \in \mathbb{R}^d$. Suppose $\mathcal{D} = \{x_i \in \bar{x} + [-B/2, B/2]^d\}_{i=1}^n$ be the projected dataset where all of the uncorrupted samples are contained in $\bar{x} + [-B/2, B/2]^d$. If $n = \widetilde{\Omega}\left(d^2 B^3 \log(1/\delta)/(\varepsilon\alpha)\right)$, then DPFILTER terminates after at most $O\left(dB^2\right)$ iterations and outputs $S_t$ such that with probability $0.9$, we have $|S_t \cap S_{\text{good}}| \geq (1 - 10\alpha)n$ and*

$$\|\mu(S_t) - \mu\|_2 \lesssim \alpha\sqrt{\log 1/\alpha} \ .$$

To prove this theorem, we use the following lemma to first show that we do not remove too many uncorrupted samples. The upper bound on the accuracy follows immediately from Lemma G.7 and the stopping criteria of the algorithm.

**Lemma D.4.** *If $n \gtrsim \frac{B^2 d^{3/2}}{\varepsilon_1 \alpha \log 1/\alpha} \log(1/\delta)$, $\lambda_t \geq (C - 0.01) \cdot \alpha \log 1/\alpha$ and $|S_t \cap S_{\text{good}}| \geq (1 - 10\alpha)n$, then there exists constant $C > 0$ such that for each iteration $t$, with probability $1 - O(1/d)$, we have Eq. (4) holds. If this condition holds, we have*

$$\mathbb{E}\left|(S_t \setminus S_{t+1}) \cap S_{\text{good}}\right| \leq \mathbb{E}\left|S_t \setminus S_{t+1} \cap S_{\text{bad}}\right| \ .$$

We measure the progress by by summing the number of clean samples removed up to iteration $t$ and the number of remaining corrupted samples, defined as $d_t \triangleq |(S_{\text{good}} \cap S) \setminus S_t| + |S_t \setminus (S_{\text{good}} \cap S)|$. Note that $d_1 = \alpha n$, and $d_t \geq 0$. At each iteration, we have

$$\mathbb{E}[d_{t+1} - d_t | d_1, d_2, \cdots, d_t] \quad = \quad \mathbb{E}\left[|S_{\text{good}} \cap (S_t \setminus S_{t+1})| - |S_{\text{bad}} \cap (S_t \setminus S_{t+1})|\right] \leq 0,$$

from the Lemma D.4. Hence, $d_t$ is a non-negative super-martingale. By optional stopping theorem, at stopping time, we have $\mathbb{E}[d_t] \leq d_1 = \alpha n$. By Markov inequality, $d_t$ is less than $10\alpha n$ with probability $0.9$, i.e. $|S_t \cap S_{\text{good}}| \geq (1 - 10\alpha)n$. The desired bound follows from induction and Lemma G.7.

Now we bound the number of iterations under the conditions of Lemma D.5. Let $W_t = |S_t \setminus S_{t-1}|/n$. Since Eq. (5), we have

$$\mathbb{E}[W_t] \geq \frac{1}{n} \sum_{i \in \mathcal{T}_{2\alpha}} \frac{\tau_i}{dB^2} \geq \frac{0.7\|M(S_{t-1}) - \mathbf{I}\|_2}{\alpha dB^2} \geq \frac{0.7 C\alpha \log(1/\alpha)}{dB^2} \ .$$

Let $T$ be the stopping time. We know $\sum_{t=1}^T W_t \leq 10\alpha$. By Wald's equation, we have

$$\mathbb{E}[\sum_{t=1}^T W_t] = \mathbb{E}[\sum_{t=1}^T \mathbb{E}[W_t]] \geq \mathbb{E}[T] \frac{0.7 C\alpha \log(1/\alpha)}{dB^2} \ .$$

This means $\mathbb{E}[T] \leq (15dB^2)/(C \log(1/\alpha))$. By Markov inequality we know with probability $0.9$, we have $T = O(dB^2/\log(1/\alpha))$.

### D.3.5 Proof of Lemma D.4

The expected number of removed good points and bad points are proportional to the $\sum_{i \in S_{\text{good}} \cap \mathcal{T}_{2\alpha}} \tau_i$ and $\sum_{i \in S_{\text{bad}} \cap \mathcal{T}_{2\alpha}} \tau_i$. It suffices to show

$$\sum_{i \in S_{\text{good}} \cap \mathcal{T}_{2\alpha}} \tau_i \quad \leq \quad \sum_{i \in S_{\text{bad}} \cap \mathcal{T}_{2\alpha}} \tau_i \ .$$

Assuming we have $\|M(S_{t-1}) - I\|_2 \geq C\alpha \log 1/\alpha$ for some $C > 0$ sufficiently large, it suffices to show

$$\frac{1}{n} \sum_{i \in S_{\text{bad}} \cap \mathcal{T}_{2\alpha}} \tau_i \geq \frac{1}{1000}\|M(S_{t-1}) - \mathbf{I}\|_2 \ .$$

First of all, we have

$$\frac{1}{n} \sum_{i \in S_{t-1}} \tau_i - 1 = v_t^\top M(S_{t-1}) v_t - 1$$

$$= v_t^\top \left( M(S_{t-1}) - \mathbf{I} \right) v_t$$

Lemma G.6 shows that the magnitude of the largest eigenvalue of $M(S_{t-1}) - \mathbf{I}$ is positive since the magnitudes negative eigenvalues are all less than $c\alpha \log 1/\alpha$. So we have

$$\frac{1}{n} \sum_{i \in S_{t-1}} \tau_i - 1 \geq \|M(S_{t-1}) - \mathbf{I}\|_2 - O(\alpha \log 1/\alpha) \tag{2}$$

$$\geq 0.9 \|M(S_{t-1}) - \mathbf{I}\|_2 , \tag{3}$$

where the first inequality follows from Lemma D.6, and the second inequality follows from our choice of large constant $C$. The next lemma regularity conditions for $\tau_i$'s for each iteration is satisfied.

**Lemma D.5.** *If $n \gtrsim \frac{B^2 d^{3/2}}{\varepsilon_1 \alpha \log 1/\alpha} \log(1/\delta)$, then there exists a large constant $C > 0$ such that, with probability $1 - O(1/d)$, we have*

*1.*

$$\frac{1}{n} \sum_{i \in S_{\text{good}} \cap \mathcal{T}_{2\alpha} \cap S_{t-1}} \tau_i \leq \frac{1}{1000} \|M(S_{t-1}) - \mathbf{I}\|_2 . \tag{4}$$

*2. For all $i \notin \mathcal{T}_{2\alpha}$,*

$$\alpha \tau_i \leq \frac{1}{1000} \|M(S_{t-1}) - \mathbf{I}\|_2 .$$

*3.*

$$\frac{1}{n} \sum_{i \in S_{\text{good}} \cap S_{t-1}} (\tau_i - 1) \leq \frac{1}{1000} \|M(S_{t-1}) - \mathbf{I}\|_2 .$$

Thus, by combining with Lemma D.5, we have

$$\frac{1}{n} \sum_{i \in S_{t-1} \cap S_{\text{bad}}} \tau_i \geq 0.8 \|M(S_{t-1}) - \mathbf{I}\|_2 .$$

We now have

$$\frac{1}{n} \sum_{i \in S_{\text{bad}} \cap \mathcal{T}_{2\alpha}} \tau_i \geq 0.8 \|M(S_{t-1}) - \mathbf{I}\|_2 - \sum_{i \in S_{\text{bad}} \cap S_{t-1} \setminus \mathcal{T}_{2\alpha}} \tau_i$$

$$\geq 0.8 \|M(S_{t-1}) - \mathbf{I}\|_2 - \max_{i \in S_{\text{bad}} \cap S_{t-1} \setminus \mathcal{T}_{2\alpha}} \alpha \tau_i$$

$$\geq 0.8 \|M(S_{t-1}) - \mathbf{I}\|_2 - \frac{1}{1000} \|M(S_{t-1}) - \mathbf{I}\|_2 \tag{5}$$

$$\geq \frac{1}{n} \sum_{i \in S_{\text{good}} \cap \mathcal{T}_{2\alpha}} \tau_i ,$$

which completes the proof.

### D.3.6 Proof of Lemma D.5

By our choice of sample complexity $n$, with probability $1 - O(1/dB^2)$, we have $\|\mu(S_{t-1}) - \mu_t\|_2^2 \lesssim \alpha \log 1/\alpha$, $v_t^\top (M(S_{t-1}) - \mathbf{I}) v_t \gtrsim \|M(S_{t-1}) - \mathbf{I}\|_2 - \alpha \log 1/\alpha$ (Lemma D.6), and $\|M(S_{t-1}) - \mathbf{I}\|_2 \geq C\alpha \log 1/\alpha$ simultaneously hold before stopping.

**Lemma D.6.** *If*

$$n \gtrsim \frac{d^{3/2} B^2}{\eta \varepsilon_1} \sqrt{2 \ln \frac{1.25}{\delta}} \log \frac{1}{\zeta} \, ,$$

*then with probability $1 - \zeta$, we have*

$$v_t^\top \left( M(S_{t-1}) - \mathbf{I} \right) v_t \geq \|M(S_{t-1}) - \mathbf{I}\|_2 - 2\eta - \frac{2|S_{t-1}|}{n} \|\mu_t - \mu(S_{t-1})\|_2^2$$

We first consider the upper bound of the good points.

$$\frac{1}{n} \sum_{i \in S_{\mathrm{good}} \cap \mathcal{T}_{2\alpha} \cap S_{t-1}} \tau_i = \frac{1}{n} \sum_{i \in S_{\mathrm{good}} \cap \mathcal{T}_{2\alpha} \cap S_{t-1}} \langle x_i - \mu_t, v_t \rangle^2$$

$$\overset{(a)}{\leq} \frac{2}{n} \sum_{i \in S_{\mathrm{good}} \cap \mathcal{T}_{2\alpha} \cap S_{t-1}} \langle x_i - \mu, v_t \rangle^2 + \frac{2}{n} |S_{\mathrm{good}} \cap \mathcal{T}_{2\alpha} \cap S_{t-1}| \langle \mu - \mu_t, v_t \rangle^2$$

$$\leq O(\alpha \log 1/\alpha) + \alpha \left( \|\mu - \mu(S_{t-1})\|_2 + \|\mu_t - \mu(S_{t-1})\|_2 \right)^2$$

$$\overset{(b)}{\leq} O(\alpha \log 1/\alpha) + \alpha \left( O(\alpha \sqrt{\log 1/\alpha}) + \sqrt{\alpha \left( \|M(S_{t-1}) - \mathbf{I}\|_2 + O(\alpha \log 1/\alpha) \right)} + O(\sqrt{\alpha \log 1/\alpha}) \right)^2$$

$$\leq O(\alpha \log 1/\alpha) + \alpha^2 \|M(S_{t-1}) - \mathbf{I}\|_2$$

$$\overset{(c)}{\leq} \frac{1}{1000} \|M(S_{t-1}) - I\|_2$$

where the $(a)$ is implied by the fact that for any vector $x, y, z$, we have $(x - y)(x - y)^\top \preceq 2(x - z)(x - z)^\top + 2(y - z)(y - z)^\top$, $(b)$ follows from Lemma G.7 and $c$ follows from our choice of large constant $C$.

Since $|S_{\mathrm{bad}} \cap \mathcal{T}_{2\alpha}| \leq \alpha n$, we know $|S_{\mathrm{good}} \cap \mathcal{T}_{2\alpha}| \geq \alpha n$, so we have for $i \notin \mathcal{T}_{2\alpha}$,

$$\alpha \tau_i \leq \frac{\alpha}{|S_{\mathrm{good}} \cap \mathcal{T}_{2\alpha} \cap S_{t-1}|} \sum_{i \in S_{\mathrm{good}} \cap \mathcal{T}_{2\alpha} \cap S_{t-1}} \tau_i \leq \frac{1}{1000} \|M(S_{t-1}) - \mathbf{I}\|_2 \, .$$

Since $|S_{\mathrm{good}} \cap S_{t-1}| \geq (1 - 10\alpha)n$, we have

$$\frac{1}{n} \sum_{i \in S_{\mathrm{good}} \cap S_{t-1}} \tau_i = \frac{1}{n} \sum_{i \in S_{\mathrm{good}} \cap S_{t-1}} \langle x_i - \mu(S_{t-1}), v_t \rangle^2 \tag{6}$$

$$= \frac{1}{n} \sum_{i \in S_{\mathrm{good}} \cap S_{t-1}} \langle x_i - \mu(S_{\mathrm{good}} \cap S_{t-1}), v_t \rangle^2 + \frac{|S_{\mathrm{good}} \cap S_{t-1}|}{n} \langle \mu(S_{\mathrm{good}} \cap S_{t-1}) - \mu(S_{t-1}), v_t \rangle^2$$

$$\tag{7}$$

$$\overset{(a)}{\leq} c\alpha \log 1/\alpha + 1 + \|\mu(S_{\mathrm{good}} \cap S_{t-1}) - \mu(S_{t-1})\|_2^2 \tag{8}$$

$$\leq c\alpha \log 1/\alpha + 1 + \left( \|\mu(S_{\mathrm{good}} \cap S_{t-1}) - \mu\|_2 + \|\mu - \mu(S_{t-1})\|_2 \right)^2 \tag{9}$$

$$\overset{(b)}{\leq} c\alpha \log 1/\alpha + 1 + \alpha \|M(S_{t-1}) - \mathbf{I}\|_2 + O(\alpha \log 1/\alpha) \tag{10}$$

$$\overset{(c)}{\leq} \frac{1}{1000} \|M(S_{t-1}) - \mathbf{I}\|_2 \, , \tag{11}$$

where $(a)$ follows from Lemma G.6, and $(b)$ follows from Lemma G.7, and $(c)$ follows from our choice of large constant $C$.

### D.3.7 Proof of Lemma D.6

*Proof.* We have following identity.

$$\frac{1}{n} \sum_{i \in S_{t-1}} (x_i - \mu_t)(x_i - \mu_t)^\top$$

$$= \frac{1}{n} \sum_{i \in S_{t-1}} (x_i - \mu(S_{t-1}))(x_i - \mu(S_{t-1}))^\top + \frac{|S_{t-1}|}{n} (\mu(S_{t-1}) - \mu_t)(\mu(S_{t-1}) - \mu_t)^\top \, .$$

So we have,

$$v_t^\top \left( M(S_{t-1}) - \mathbf{I} \right) v_t$$

$$\geq \quad v_t^\top \left( \frac{1}{n} \sum_{i \in S_{t-1}} (x_i - \mu_t)(x_i - \mu_t)^\top - \mathbf{I} \right) v_t - \frac{|S_{t-1}|}{n} \|\mu_t - \mu(S_{t-1})\|_2^2$$

$$\geq \quad \|M(S_{t-1}) - \mathbf{I}\|_2 - 2\eta - \frac{2|S_{t-1}|}{n} \|\mu_t - \mu(S_{t-1})\|_2^2$$

where the last inequality follows from Lemma G.6, which shows that the magnitude of the largest eigenvalue of $M(S_{t-1}) - \mathbf{I}$ must be positive. $\qquad\square$

# E   PRIME: efficient algorithm for private and robust mean estimation

We provide our main algorithms, Algorithm 9 and Algorithm 10, in Appendix E.1 and the corresponding proof in Appendix F. We provide our novel DPTHRESHOLD and its anlysis in Appendix E.2.

We define $S_{\text{good}}$ as the original set of $n$ clean samples (as defined in Assumption 1 and 2) and $S_{\text{bad}}$ as the set of corrupted samples that replace $\alpha n$ of the clean samples. The (rescaled) covariance is denoted by $M(S^{(s)}) \triangleq (1/n) \sum_{i \in S^{(s)}} (x_i - \mu(S^{(s)}))(x_i - \mu(S^{(s)}))^\top$, where $\mu(S^{(s)}) \triangleq (1/|S^{(s)}|) \sum_{i \in S^{(s)}} x_i$ denotes the mean.

### E.1 PRIvate and robust Mean Estimation (PRIME)

---

**Algorithm 9:** PRIvate and robust Mean Estimation (PRIME)

---

**Input:** $S = \{x_i \in \mathbb{R}^d\}_{i=1}^n$, adversarial fraction $\alpha \in (0, 1/2)$, number of iterations
$\quad\quad T_1 = O(\log d), T_2 = O(\log d)$, target privacy $(\varepsilon, \delta)$

1 $(\bar{x}, B) \leftarrow q_{\text{range}}(\{x_i\}_{i=1}^n, 0.01\varepsilon, 0.01\delta)$             [Algorithm 5 in Appendix D.3.1]

2 Clip the data points: $\tilde{x}_i \leftarrow \mathcal{P}_{\bar{x}+[-B/2, B/2]^d}(x_i)$, for all $i \in [n]$

3 $\hat{\mu} \leftarrow$ DPMMWFILTER$(\{\tilde{x}_i\}_{i=1}^n, \alpha, T_1, T_2, 0.99\varepsilon, 0.99\delta)$          [Algorithm 10]

**Output:** $\hat{\mu}$

---

**Algorithm 10:** Differentially private filtering with matrix multiplicative weights (DPMMWFIL-TER)

---

**Input:** $S = \{x_i \in \bar{x} + [-B/2, B/2]^d\}_{i=1}^n$, $\alpha \in (0, 1/2)$, $T_1 = O(\log(B\sqrt{d}))$, $T_2 = O(\log d)$,
$\quad\quad$ privacy $(\varepsilon, \delta)$

1 Initialize $S^{(1)} \leftarrow [n]$, $\varepsilon_1 \leftarrow \varepsilon/(4T_1)$, $\delta_1 \leftarrow \delta/(4T_1)$, $\varepsilon_2 \leftarrow \min\{0.9, \varepsilon\}/(4\sqrt{10T_1 T_2 \log(4/\delta)})$,
$\quad \delta_2 \leftarrow \delta/(20T_1 T_2)$, a large enough constant $C > 0$

2 **if** $n < (4/\varepsilon_1)\log(1/(2\delta_1))$ **then Output:** $\emptyset$

3 **for** epoch $s = 1, 2, \ldots, T_1$ **do**

4     $\lambda^{(s)} \leftarrow \|M(S^{(s)}) - \mathbf{I}\|_2 + \text{Lap}(2B^2 d/(n\varepsilon_1))$

5     $n^{(s)} \leftarrow |S^{(s)}| + \text{Lap}(1/\varepsilon_1)$

6     **if** $n^{(s)} \leq 3n/4$ **then Output:** $\emptyset$

7     **if** $\lambda^{(s)} \leq C\,\alpha \log(1/\alpha)$ **then**

       **Output:**
$$\mu^{(s)} \leftarrow (1/|S^{(s)}|)\left(\sum_{i \in S^{(s)}} x_i\right) + \mathcal{N}(0, (2B\sqrt{2d\log(1.25/\delta_1)}/(n\,\varepsilon_1))^2 \mathbf{I}_{d \times d})$$

8     $\alpha^{(s)} \leftarrow 1/(100(0.1/C + 1.01)\lambda^{(s)})$

9     $S_1^{(s)} \leftarrow S^{(s)}$

10     **for** $t = 1, 2, \ldots, T_2$ **do**

11        $\lambda_t^{(s)} \leftarrow \|M(S_t^{(s)}) - \mathbf{I}\|_2 + \text{Lap}(2B^2 d/(n\varepsilon_2))$

12        **if** $\lambda_t^{(s)} \leq 0.5\lambda_0^{(s)}$ **then**

13           terminate epoch

14        **else**

15           $\Sigma_t^{(s)} \leftarrow M(S_t^{(s)}) + \mathcal{N}(0, (4B^2 d\sqrt{2\log(1.25/\delta_2)}/(n\varepsilon_2))^2 \mathbf{I}_{d^2 \times d^2})$

16           $U_t^{(s)} \leftarrow (1/\text{Tr}(\exp(\alpha^{(s)}\sum_{r=1}^t (\Sigma_r^{(s)} - \mathbf{I})))) \exp(\alpha^{(s)}\sum_{r=1}^t (\Sigma_r^{(s)} - \mathbf{I}))$

17           $\psi_t^{(s)} \leftarrow \left\langle M(S_t^{(s)}) - \mathbf{I}, U_t^{(s)}\right\rangle + \text{Lap}(2B^2 d/(n\varepsilon_2))$

18           **if** $\psi_t^{(s)} \leq (1/5.5)\lambda_t^{(s)}$ **then**

19              $S_{t+1}^{(s)} \leftarrow S_t^{(s)}$

20           **else**

21              $Z_t^{(s)} \leftarrow \text{Unif}([0, 1])$

22              $\mu_t^{(s)} \leftarrow (1/|S_t^{(s)}|)\left(\sum_{i \in S_t} x_i\right) + \mathcal{N}(0, (2B\sqrt{2d\log(1.25/\delta_2)}/(n\,\varepsilon_2)\mathbf{I}_{d \times d})^2)$

23              $\rho_t^{(s)} \leftarrow$ DPTHRESHOLD$(\mu_t^{(s)}, U_t^{(s)}, \alpha, \varepsilon_2, \delta_2, S_t^{(s)})$      [Algorithm 11]

24              $S_{t+1}^{(s)} \leftarrow S_t^{(s)} \setminus \{i \,|\, i \in \mathcal{T}_{2\alpha} \text{ for } \{\tau_j = (x_j - \mu_t^{(s)})^\top U_t^{(s)}(x_j - \mu_t^{(s)})\}_{j \in S_t^{(s)}}$ and
$\quad\quad\quad\quad\quad\quad \tau_i \geq \rho_t^{(s)} Z_t^{(s)}\}$, where $\mathcal{T}_{2\alpha}$ is defined in Definition C.1.

25     $S^{(s+1)} \leftarrow S_t^{(s)}$

**Output:** $\mu^{(T_1)}$

---

## E.2 Algorithm and analysis of DPTHRESHOLD

---

**Algorithm 11:** Differentially private estimation of the threshold (DPTHRESHOLD)

---

**Input:** $\mu, U, \alpha \in (0, 1/2)$, target privacy $(\varepsilon, \delta)$, $S = \{x_i \in \bar{x} + [-B/2, B/2]^d\}$

**1** Set $\tau_i \leftarrow (x_i - \mu)^\top U(x_i - \mu)$ for all $i \in S$

**2** Set $\tilde{\psi} \leftarrow (1/n) \sum_{i \in S}(\tau_i - 1) + \text{Lap}(2B^2 d/n\varepsilon))$

**3** Compute a histogram over geometrically sized bins

$$I_1 = [1/4, 1/2), I_2 = [1/2, 1), \ldots, I_{2+\log(B^2 d)} = [2^{\log(B^2 d)-1}, 2^{\log(B^2 d)}]$$

$$h_j \leftarrow \frac{1}{n} \cdot |\{i \in S \mid \tau_i \in [2^{-3+j}, 2^{-2+j})\}|, \quad \text{for all } j = 1, \ldots, 2 + \log(B^2 d)$$

**4** Compute a privatized histogram $\tilde{h}_j \leftarrow h_j + \mathcal{N}(0, (4\sqrt{2\log(1.25/\delta)}/(n\varepsilon))^2)$, for all $j \in [2 + \log(B^2 d)]$

**5** Set $\tilde{\tau}_j \leftarrow 2^{-3+j}$, for all $j \in [2 + \log(B^2 d)]$

**6** Find the largest $\ell \in [2 + \log(B^2 d)]$ satisfying $\sum_{j \geq \ell}(\tilde{\tau}_j - \tilde{\tau}_\ell) \tilde{h}_j \geq 0.31\tilde{\psi}$

**Output:** $\rho = \tilde{\tau}_\ell$

---

**Lemma E.1** (DPTHRESHOLD: picking threshold privately). *Algorithm DPTHRESHOLD($\mu, U, \alpha, \varepsilon, \delta, S$) running on a dataset $\{\tau_i = (x_i - \mu)^\top U(x_i - \mu)\}_{i \in S}$ is $(\varepsilon, \delta)$-DP. Define $\psi \triangleq \frac{1}{n} \sum_{i \in S}(\tau_i - 1)$. If $\tau_i$'s satisfy*

$$\frac{1}{n} \sum_{i \in S_{\text{good}} \cap \mathcal{T}_{2\alpha} \cap S} \tau_i \leq \psi/1000$$

$$\frac{1}{n} \sum_{i \in S_{\text{good}} \cap S} (\tau_i - 1) \leq \psi/1000,$$

*and $n \geq \widetilde{\Omega}\left(\frac{B^2 d\sqrt{\log(1/\delta)}}{\varepsilon\alpha}\right)$, then DPTHRESHOLD outputs a threshold $\rho$ such that with probability $1 - O(1/\log^3 d)$,*

$$\frac{1}{n} \sum_{\tau_i < \rho} (\tau_i - 1) \leq 0.75\psi \quad \text{and} \tag{12}$$

$$2\left(\sum_{i \in S_{\text{good}} \cap \mathcal{T}_{2\alpha}} \boldsymbol{I}\{\tau_i \leq \rho\}\frac{\tau_i}{\rho} + \boldsymbol{I}\{\tau_i > \rho\}\right) \leq \sum_{i \in S_{\text{bad}} \cap \mathcal{T}_{2\alpha}} \boldsymbol{I}\{\tau_i \leq \rho\}\frac{\tau_i}{\rho} + \boldsymbol{I}\{\tau_i > \rho\}. \tag{13}$$

## E.3 Proof of Lemma E.1

**1. Threshold $\rho$ sufficiently reduces the total score.**

Let $\rho$ be the threshold picked by the algorithm. Let $\hat{\tau}_i$ denote the minimum value of the interval of the bin that $\tau_i$ belongs to. It holds that

$$\frac{1}{n} \sum_{\tau_i \geq \rho, i \in [n]} (\tau_i - \rho) \geq \frac{1}{n} \sum_{\hat{\tau}_i \geq \rho, i \in [n]} (\hat{\tau}_i - \rho)$$

$$= \sum_{\tilde{\tau}_j \geq \rho, j \in [2+\log(B^2 d)]} (\tilde{\tau}_j - \rho)h_j$$

$$\overset{(a)}{\geq} \sum_{\tilde{\tau}_j \geq \rho, j \in [2+\log(B^2 d)]} (\tilde{\tau}_j - \rho)\tilde{h}_j - O\left(\log(B^2 d) \cdot B^2 d \cdot \frac{\sqrt{\log(\log(B^2 d)\log d)\log(1/\delta)}}{\varepsilon n}\right)$$

$$\overset{(b)}{\geq} 0.31\tilde{\psi} - \tilde{O}\left(\frac{B^2 d}{\varepsilon n}\right)$$

$$\overset{(c)}{\geq} 0.3\psi - \tilde{O}\left(\frac{B^2 d}{\varepsilon n}\right),$$

where $(a)$ holds due to the accuracy of the private histogram (Lemma G.12), $(b)$ holds by the definition of $\rho$ in our algorithm, and $(c)$ holds due to the accuracy of $\tilde{\psi}$. This implies if $\rho < 1$, then $\frac{1}{n}\sum_{\tau_i<\rho}(\tau_i-1)$ is negative and if $\rho \geq 1$, then

$$\frac{1}{n}\sum_{\tau_i<\rho}(\tau_i-1) = \psi - \frac{1}{n}\sum_{\tau_i\geq\rho}(\tau_i-1) \leq \psi - \frac{1}{n}\sum_{\tau_i\geq\rho}(\tau_i-\rho) \leq 0.7\psi + \tilde{O}(B^2 d/\varepsilon n).$$

By Lemma E.2, it holds that

$$
\begin{aligned}
\frac{1}{n}\sum_{i\in S\backslash\mathcal{T}_{2\alpha}}(\tau_i-1) &= \psi - \frac{1}{n}\sum_{i\in S_{\text{good}}\cap\mathcal{T}_{2\alpha}}(\tau_i-1) - \frac{1}{n}\sum_{i\in S_{\text{bad}}\cap\mathcal{T}_{2\alpha}}(\tau_i-1) \\
&\leq \psi - \frac{1}{n}\sum_{i\in S_{\text{bad}}\cap\mathcal{T}_{2\alpha}}(\tau_i-1) \\
&\leq (2/1000)\psi
\end{aligned}
$$

And we conclude that

$$\frac{1}{n}\sum_{\tau_i<\rho \text{ or } i\notin\mathcal{T}_{2\alpha}}(\tau_i-1) \leq 0.71\psi + \tilde{O}(B^2 d/\varepsilon n) \leq 0.75\psi$$

## 2. Threshold $\rho$ removes more bad data points than good data points.

Define $C_2$ to be the threshold such that $\frac{1}{n}\sum_{\tau_i>C_2}(\tau_i-C_2) = (2/3)\psi$. Suppose $2^b \leq C_2 \leq 2^{b+1}$, $\frac{1}{n}\sum_{\hat{\tau}_i\geq 2^{b-1}}(\hat{\tau}_i-2^{b-1}) \geq (1/3)\psi$ because $\forall \tau_i \geq C_2$, $(\hat{\tau}_i-2^{b-1}) \geq \frac{1}{2}(\tau_i-C_2)$. Trivially $C_2 \geq 1$ due to the fact that $\frac{1}{n}\sum_{\tau_i\geq 1}\tau_i - 1 \geq \psi$. Then we have the threshold picked by the algorithm $\rho \geq 2^{b-1}$, which implies $\rho \geq \frac{1}{4}C_2$. Suppose $\rho < C_2$, since $\rho \geq \frac{1}{4}C_2$, we have

$$
\begin{aligned}
\Big(\sum_{i\in S_{\text{bad}}\cap\mathcal{T}_{2\alpha},\tau_i<\rho}\tau_i + \sum_{i\in S_{\text{bad}}\cap\mathcal{T}_{2\alpha},\tau_i\geq\rho}\rho\Big) &\geq \frac{1}{4}\Big(\sum_{i\in S_{\text{bad}}\cap\mathcal{T}_{2\alpha},\tau_i<C_2}\tau_i + \sum_{i\in S_{\text{bad}}\cap\mathcal{T}_{2\alpha},\tau_i\geq C_2}C_2\Big) \\
&\overset{(a)}{\geq} \frac{10}{4}\Big(\sum_{i\in S_{\text{good}}\cap\mathcal{T}_{2\alpha},\tau_i<C_2}\tau_i + \sum_{i\in S_{\text{good}}\cap\mathcal{T}_{2\alpha},\tau_i\geq C_2}C_2\Big) \\
&\overset{(b)}{\geq} \frac{10}{4}\Big(\sum_{i\in S_{\text{good}}\cap\mathcal{T}_{2\alpha},\tau_i<\rho}\tau_i + \sum_{i\in S_{\text{good}}\cap\mathcal{T}_{2\alpha},\tau_i>=\rho}\rho\Big),
\end{aligned}
$$

where (a) holds by Lemma E.3, and (b) holds since $\rho \leq C_2$. If $\rho \geq C_2$, the statement of the Lemma E.3 directly implies Equation (13).

**Lemma E.2.** *[Conditions for $\tau_i$'s] Suppose*

$$\frac{1}{n}\sum_{i\in S_{\text{good}}\cap S}(\tau_i-1) \leq \psi/1000$$

$$\frac{1}{n}\sum_{i\in S_{\text{good}}\cap\mathcal{T}_{2\alpha}}\tau_i \leq \psi/1000$$

*then, we have*

$$\alpha\tau_{2\alpha n} \leq \psi/1000$$

$$\frac{1}{n}\sum_{i\in S_{\text{bad}}\cap\mathcal{T}_{2\alpha}}(\tau_i-1) \geq (998/1000)\psi$$

*Proof.* Since $|S_{\text{good}}\cap\mathcal{T}_{2\alpha}| \geq \alpha n$, it holds

$$\alpha\tau_{2\alpha n} \leq \psi/1000.$$

$$\frac{1}{n}\sum_{i\in S_{\text{bad}}\cap \mathcal{T}_{2\alpha}}(\tau_i-1) \;=\; \frac{1}{n}\sum_{i\in S_{\text{bad}}\cap S}(\tau_i-1) - \frac{1}{n}\sum_{i\in S_{\text{bad}}\cap S\backslash \mathcal{T}_{2\alpha}}(\tau_i-1)$$

$$\geq \; (999/1000)\psi - \frac{1}{n}\sum_{i\in S_{\text{bad}}\cap S\backslash \mathcal{T}_{2\alpha}}(\tau_i-1)$$

$$\geq \; (999/1000)\psi - (1/1000)\psi$$

$$= \; (998/1000)\psi$$

$\square$

**Lemma E.3.** *Assuming that the conditions in Lemma E.2 holds, and for any $C$ such that*

$$\frac{1}{n}\sum_{i\in S,\tau_i<C}(\tau_i-1) + \frac{1}{n}\sum_{i\in S,\tau_i\geq C}(C-1) \geq (1/3)\psi \;,$$

*we have*

$$\sum_{i\in S_{\text{bad}}\cap \mathcal{T}_{2\alpha},\tau_i<C}\tau_i + \sum_{i\in S_{\text{bad}}\cap \mathcal{T}_{2\alpha},\tau_i\geq C}C \geq 10\Big(\sum_{i\in S_{\text{good}}\cap \mathcal{T}_{2\alpha},\tau_i<C}\tau_i + \sum_{i\in S_{\text{good}}\cap \mathcal{T}_{2\alpha},\tau_i\geq C}C\Big)$$

*Proof.* First we show an upper bound on $S_{\text{good}}\cap \mathcal{T}_{2\alpha}$:

$$\frac{1}{n}\sum_{i\in S_{\text{good}}\cap \mathcal{T}_{2\alpha},\tau_i<C}\tau_i + \frac{1}{n}\sum_{i\in S_{\text{good}}\cap \mathcal{T}_{2\alpha},\tau_i\geq C}C \leq \frac{1}{n}\sum_{i\in S_{\text{good}}\cap \mathcal{T}_{2\alpha}}\tau_i \leq \psi/1000.$$

Then we show an lower bound on $S_{\text{bad}}\cap \mathcal{T}_{2\alpha}$:

$$\frac{1}{n}\sum_{i\in S_{\text{bad}}\cap S,\tau_i<C}(\tau_i-1) + \frac{1}{n}\sum_{i\in S_{\text{bad}}\cap S,\tau_i>C}(C-1)$$

$$= \; \frac{1}{n}\sum_{i\in S,\tau_i<C}(\tau_i-1) + \frac{1}{n}\sum_{i\in S,\tau_i\geq C}(C-1)$$

$$-\Big(\frac{1}{n}\sum_{i\in S_{\text{good}}\cap S,\tau_i<C}(\tau_i-1) + \frac{1}{n}\sum_{i\in S_{\text{good}}\cap S,\tau_i\geq C}(C-1)\Big)$$

$$\geq \; (1/3-1/1000)\psi \;.$$

We have

$$\frac{1}{n}\sum_{i\in S_{\text{bad}}\cap \mathcal{T}_{2\alpha},\tau_i<C}\tau_i + \frac{1}{n}\sum_{i\in S_{\text{bad}}\cap \mathcal{T}_{2\alpha},\tau_i>C}C \geq \frac{1}{n}\sum_{i\in S_{\text{bad}}\cap \mathcal{T}_{2\alpha},\tau_i<C}(\tau_i-1) + \frac{1}{n}\sum_{i\in S_{\text{bad}}\cap \mathcal{T}_{2\alpha},\tau_i>C}(C-1)$$

$$= \frac{1}{n}\sum_{i\in S_{\text{bad}}\cap S,\tau_i<\rho}(\tau_i-1) + \frac{1}{n}\sum_{i\in S_{\text{bad}}\cap S,\tau_i>C}(C-1)$$

$$- \Big(\frac{1}{n}\sum_{i\in S_{\text{bad}}\cap S\backslash \mathcal{T}_{2\alpha},\tau_i<C}(\tau_i-1) + \frac{1}{n}\sum_{i\in S_{\text{bad}}\cap S\backslash \mathcal{T}_{2\alpha},\tau_i>C}(C-1)\Big)$$

$$\geq (1/3-1/1000)\psi - \alpha\tau_{2\alpha n}$$

$$\geq (1/3-2/1000)\psi$$

Combing the lower bound and the upper bound yields the desired statement $\square$

# F  The analysis of PRIME and the proof of Theorem 6

## F.1  Proof of part 1 of Theorem 6 on differential privacy

Let $(\varepsilon_0, \delta_0)$ be the end-to-end target privacy guarantee. The $(0.01\varepsilon_0, 0.01\delta_0)$-DP guarantee of $q_{\text{range}}$ follows from Lemma D.2. We are left to show that DPMMWFILTER in Algorithm 10 satisfy $(0.99\varepsilon_0, 0.99\delta_0)$-DP. To this end, we explicitly write out how many times we access the database and how much privacy is lost each time in an interactive version of DPMMWFILTER in Algorithm 13, which performs the same operations as DPMMWFILTER.

In order to apply Lemma G.13, we cap $\varepsilon$ at 0.9 in initializing $\varepsilon_2$. We call $q_{\text{spectral}}$ and $q_{\text{size}}$ $T_1$ times, each with $(\varepsilon_1, \delta_1)$ guarantee. In total this accounts for $(0.5\varepsilon, 0.5\delta)$ privacy loss. The rest of the mechanisms are called $5T_1T_2$ times ($q_{\text{spectral}}(\cdot)$ and $q_{\text{MMW}}(\cdot)$ each call two DP mechanisms internally), each with $(\varepsilon_2, \delta_2)$ guarantee. In total this accounts for $(0.5\varepsilon, 0.5\delta)$ privacy loss. Altogether, this is within the privacy budget of $(\varepsilon = 0.99\varepsilon_0, \delta = 0.99\delta_0)$.

We are left to show privacy of $q_{\text{spectral}}$, $q_{\text{MMW}}$, and $q_{\text{1Dfilter}}$, and $q_{\text{size}}$ in Algorithm 12. We will assume for now that $|S_r^{(\ell)}| \geq n/2$ for all $\ell \in [T_1]$ and $r \in [T_2]$ and prove privacy. We show in the end that this happens with probability larger than $1 - \delta_1$. In all sub-routines, we run $\text{Filter}(\cdot)$ in Algorithm 12 to simulate the filtering process so far and get the current set of samples $S_{t_s}^{(s)}$. The following main technical lemma allows us to prove privacy of all interactive mechanisms. This is a counterpart of Lemma 2.2 used for DPFILTER. We provide a proof in Appendix D.3.2.

**Lemma F.1.** *Let $S(\mathcal{D}_n) \subseteq \mathcal{D}_n$ denote the output of the simulated filtering process* $\text{Filter}(\cdot)$ *on $\mathcal{D}_n$ for a given set of parameters $(\{\{\Psi_r^{(\ell)}\}_{r \in [t_\ell]}\}_{\ell \in [s]}, \{(\mu^{(\ell)}, \lambda^{(\ell)})\}_{\ell \in [s]})$ in Algorithm 12. Then we have $d_\triangle(S(\mathcal{D}_n), S(\mathcal{D}')_n) \leq d_\triangle(\mathcal{D}_n, \mathcal{D}'_n)$, where $d_\triangle(\mathcal{D}, \mathcal{D}') \triangleq \max\{|\mathcal{D} \setminus \mathcal{D}'|, |\mathcal{D}' \setminus \mathcal{D}|\}$.*

This is a powerful tool for designing private mechanisms, as it guarantees that we can safely simulate the filtering process with privatized parameters and preserve the neighborhood of the dataset; if $\mathcal{D}_n \sim \mathcal{D}'_n$ are neighboring (i.e., $d_\triangle(\mathcal{D}_n, \mathcal{D}'_n) \leq 1$) then so are the filtered pair $S(\mathcal{D}_n)$ and $S(\mathcal{D}'_n)$ (i.e., $d_\triangle(S(\mathcal{D}_n), S(\mathcal{D}'_n)) \leq 1$). Note that in all the interactive mechanisms in Algorithm 12, the noise we need to add is proportional to the set sensitivity of $\text{Filter}(\cdot)$ defined as $\Delta_{\text{set}} \triangleq \max_{\mathcal{D}_n \sim \mathcal{D}'_n} d_\triangle(S(\mathcal{D}_n), S(\mathcal{D}'_n))$. If the repeated application of the $\text{Filter}(\cdot)$ is not a contraction in $d_\triangle(\cdot, \cdot)$, this results in a sensitivity blow-up. Fortunately, the above lemma ensures contraction of the filtering, proving that $\Delta_{\text{set}} = 1$. Hence, it is sufficient for us to prove privacy for two neighboring filtered sets $S \sim S'$ (as opposed to proving privacy for two neighboring original datasets before filtering $\mathcal{D}_n \sim \mathcal{D}'_n$).

In $q_{\text{spectral}}$, $\lambda$ satisfy $(\varepsilon, 0)$-DP as the $L_1$ sensitivity is $\Delta_1 = (1/n)B^2d$ (Definition 1.2) and we add $\text{Lap}(\Delta_1/\varepsilon)$. The release of $\mu$ also satisfy $(\varepsilon, \delta)$-DP as the $L_2$ sensitivity is $\Delta_2 = 2B\sqrt{d}/n$, assuming $|S| \geq n/2$ as ensured by the stopping criteria, and we add $\mathcal{N}(0, \Delta_2(2\log(1.25/\delta))/\varepsilon)^2\mathbf{I})$. Note that in the outer loop call of $q_{\text{spectral}}$, we only release $\mu$ once in the end, and hence we count $q_{\text{spectral}}$ as one access. On the other hand, in the inner loop, we use both $\mu$ and $\lambda$ from $q_{\text{spectral}}$ so we count it as two accesses.

In $q_{\text{size}}$, the returned set size $(\varepsilon, 0)$-DP as the $L_1$ sensitivity is $\Delta_1 = 1$ and we add $\text{Lap}(\Delta_1/\varepsilon)$. One caveat is that we need to ensure that the stopping criteria of checking $n^{(s)} > 3n/4$ ensures that $|S_t^{(s)}| > n/2$ with probability at least $1 - \delta_1$. This guarantees that the rest of the private mechanisms can assume $|S_t^{(s)}| > n/2$ in analyzing the sensitivity. Since Laplace distribution follows $f(z) = (\varepsilon/2)e^{-\varepsilon|z|}$, we have $\mathbb{P}(n^{(s)} > 3n/4 \text{ and } |S_t^{(s)}| < n/2) \leq (1/2)e^{-n\varepsilon/4}$. Hence, the desired privacy is ensured for $(1/2)e^{-n\varepsilon/4} \leq \delta_1$ (i.e., $n \geq (4/\varepsilon_1)\log(1/(2\delta_1))$).

In $q_{\text{MMW}}$, $\Sigma$ is $(\varepsilon, \delta)$-DP as the $L_2$ sensitivity is $\Delta_2 = B^2d/n$, and we add $\mathcal{N}(0, \Delta_2(2\log(1.25/\delta))/\varepsilon)^2\mathbf{I})$. $\psi$ is $(\varepsilon, 0)$-DP as the $L_1$ sensitivity is $\Delta_1 = 2B^2d/n$ and we add $\text{Lap}(\Delta_1/\varepsilon)$. This is made formal in the following theorem with a proof. in Appendix F.1.1. This algorithm is identical to the MOD-SULQ algorithm introduced in [13] and analyzed in [18, Theorem 5], up to the choice of the noise variance. But a tighter analysis improves over the MOD-SULQ analysis from [18] by a factor of $d$ in the variance of added Gaussian noise as noted in [39].

**Lemma F.2** (Differentially Private PCA). *Consider a dataset $\{x_i \in \mathbb{R}^d\}_{i=1}^n$. If $\|x_i\|_2 \le 1$ for all $i \in [n]$, the following privatized second moment matrix satisfies $(\varepsilon, \delta)$-differential privacy:*

$$\frac{1}{n} \sum_{i=1}^n x_i x_i^\top + Z \ ,$$

*with $Z_{i,j} \sim \mathcal{N}(0, (\,(1/(n\varepsilon))\sqrt{2\log(1.25/\delta)}\,)^2)$ for $i \ge j$ and $Z_{i,j} = Z_{j,i}$ for $i < j$.*

In $q_{\mathrm{1Dfilter}}$, the $(\varepsilon, \delta)$ differential privacy follows from that of DPTHRESHOLD proved in Lemma E.1.

### F.1.1 Proof of Lemma F.2

Consider neighboring two databases $\mathcal{D} = \{x_i\}_{i=1}^n$ and $\tilde{\mathcal{D}} = \mathcal{D} \cup \{\tilde{x}_n\} \setminus \{x_n\}$, and let $A = (1/n)\sum_{x_i \in \mathcal{D}} x_i x_i^\top$ and $\tilde{A} = (1/n)\sum_{x_i \in \tilde{\mathcal{D}}} x_i x_i^\top$. Let $B$ and $\tilde{B}$ be the Gaussian noise matrix with $\beta^2$ as variance. Let $G = A + B$ and $\tilde{G} = \tilde{A} + \tilde{B}$. At point $H$, we have

$$
\begin{aligned}
\ell_{D,\tilde{D}} \;=\; \log \frac{f_G(H)}{f_{\tilde{G}}(H)} \;&=\; \sum_{1 \le i \le j \le d} \left( -\frac{1}{2\beta^2}(H_{ij} - A_{ij})^2 + \frac{1}{2\beta^2}\left( H_{ij} - \hat{A}_{ij} \right)^2 \right) \\
&=\; \frac{1}{2\beta^2} \sum_{1 \le i \le j \le d} \left( \frac{2}{n}(H_{ij} - A_{ij})(x_{n,i}x_{n,j} - \hat{x}_{n,i}\hat{x}_{n,j}) + \frac{1}{n^2}(\hat{x}_{n,i}\hat{x}_{n,j} - x_{n,i}x_{n,j})^2 \right) \ .
\end{aligned}
$$

Since $\|x_n\|_2 \le 1$ and $\|\tilde{x}_n\|_2 \le 1$, we have $\sum_{1 \le i \le j \le d} (\hat{x}_{n,i}\hat{x}_{n,j} - x_{n,i}x_{n,j})^2 = 1/2\|\tilde{x}_n\tilde{x}_n^\top - x_n x_n^\top\|_F^2 \le 2$.

Now we bound the first term,

$$
\begin{aligned}
2 \sum_{1 \le i \le j \le d}(H_{ij} - A_{ij})(x_{n,i}x_{n,j} - \hat{x}_{n,i}\hat{x}_{n,j}) \;&=\; \left\langle H - A, x_n x_n^\top - \tilde{x}_n \tilde{x}_n^\top \right\rangle \\
&=\; x_n^\top B x_n - \tilde{x}_n^\top B \tilde{x}_n \\
&\le\; 2\|B\|_2 \ .
\end{aligned}
$$

So we have $|\ell_{D,\tilde{D}}| \le \varepsilon$ whenever $\|B\|_2 \le n\varepsilon\beta^2 - 1/n$.

For any fixed unit vector $\|v\|_2 = 1$, we have

$$ v^\top B v = 2 \sum_{1 \le i \le j \le d} B_{ij} v_i v_j \sim \mathcal{N}\Big(0, 2\sum_{1 \le i \le j \le d} v_i^2 v_j^2\Big) \;=\; \mathcal{N}(0, 1) \ . $$

Then we have

$$
\begin{aligned}
\mathbb{P}\Big(|\ell_{D,\tilde{D}}| \ge \varepsilon\Big) \;&\le\; \mathbb{P}\big(\|B\|_2 \ge n\varepsilon\beta^2 - 1/n\big) \\
&=\; \mathbb{P}\left( \mathcal{N}(0,1) \ge n\varepsilon\beta^2 - \frac{1}{n} \right) \\
&=\; \Phi\left( \frac{1}{n} - n\varepsilon\beta^2 \right) \ ,
\end{aligned}
$$

where $\Phi$ is CDF of standard Gaussian. According to Gaussian mechanism, if $\beta = (1/(n\varepsilon))\sqrt{2\log(1.25/\delta)}$, we have $\Phi\left( \frac{1}{n} - n\varepsilon\beta^2 \right) \le \delta$.

### F.2 Proof of part 2 of Theorem 6 on accuracy

The accuracy of PRIME follows from the fact that $q_{\mathrm{range}}$ returns a hypercube that contains all the clean data with high probability (Lemma D.2) and that DPMMWFILTER achieves the desired accuracy (Theorem 11) if the original uncorrupted dataset $S_{\mathrm{good}}$ is $\alpha$-subgaussian good. $S_{\mathrm{good}}$ is $\alpha$-subgaussian good if we have $n = \tilde{\Omega}(d/\alpha^2)$ as shown in Lemma G.3. We present the proof of Theorem 11 below.

**Algorithm 12:** Interactive differentially private mechanisms for DPMMWFILTER

---

**1** $q_{\text{spectral}}(\{\{\Psi_r^{(\ell)}\}_{r\in[t_\ell]}\}_{\ell\in[s]}, \{(\mu^{(\ell)}, \lambda^{(\ell)})\}_{\ell\in[s]}, \varepsilon, \delta)$**:**

**2** $\quad S \leftarrow \text{Filter}(\{\{\Psi_r^{(\ell)}\}_{r\in[t_\ell]}\}_{\ell\in[s]}, \{(\mu^{(\ell)}, \lambda^{(\ell)})\}_{\ell\in[s]}, \varepsilon, \delta)$

**3** $\quad \mu \leftarrow (1/|S|)\left(\sum_{i\in S} x_i\right) + \mathcal{N}(0, (2B\sqrt{2d\log(1.25/\delta)}/(n\varepsilon))^2 \mathbf{I})$

**4** $\quad \lambda \leftarrow \|M(S) - \mathbf{I}\|_2 + \text{Lap}(2B^2 d/(n\varepsilon))$

**5** $\quad$ **return** $(\mu, \lambda)$

**6** $q_{\text{size}}(\{\{\Psi_r^{(\ell)}\}_{r\in[t_\ell]}\}_{\ell\in[s]}, \{(\mu^{(\ell)}, \lambda^{(\ell)})\}_{\ell\in[s]}, \varepsilon, \delta)$**:**

**7** $\quad S \leftarrow \text{Filter}(\{\{\Psi_r^{(\ell)}\}_{r\in[t_\ell]}\}_{\ell\in[s]}, \{(\mu^{(\ell)}, \lambda^{(\ell)})\}_{\ell\in[s]}, \varepsilon, \delta)$

**8** $\quad$ **return** $|S| + \text{Lap}(1/\varepsilon)$

**9** $q_{\text{MMW}}(\{\{\Psi_r^{(\ell)}\}_{r\in[t_\ell]}\}_{\ell\in[s]}, \{(\mu^{(\ell)}, \lambda^{(\ell)})\}_{\ell\in[s]}, \alpha^{(s)}, \mu_t^{(s)}, \varepsilon, \delta)$**:**

**10** $\quad S \leftarrow \text{Filter}(\{\{\Psi_r^{(\ell)}\}_{r\in[t_\ell]}\}_{\ell\in[s]}, \{(\mu^{(\ell)}, \lambda^{(\ell)})\}_{\ell\in[s]}, \varepsilon, \delta)$

**11** $\quad \Sigma_{t_s+1}^{(s)} \leftarrow M(S) + \mathcal{N}(0, (4B^2 d\sqrt{2\log(1.25/\delta)}/(n\varepsilon))^2 \mathbf{I})$

**12** $\quad U \leftarrow (1/\text{Tr}(\exp(\alpha^{(s)}\sum_{r=1}^{t_s+1}(\Sigma_r^{(s)} - \mathbf{I}))))\exp(\alpha^{(s)}\sum_{r=1}^{t_s+1}(\Sigma_r^{(s)} - \mathbf{I}))$

**13** $\quad \psi \leftarrow \langle M(S) - \mathbf{I}, U\rangle + \text{Lap}(2B^2 d/(n\varepsilon))$

**14** $\quad$ **return** $(\Sigma_{t_s+1}^{(s)}, U, \psi)$

**15** $q_{\text{1Dfilter}}(\{\{\Psi_r^{(\ell)}\}_{r\in[t_\ell]}\}_{\ell\in[s]}, \{(\mu^{(\ell)}, \lambda^{(\ell)})\}_{\ell\in[s]}, \mu, U, \alpha, \varepsilon, \delta)$**:**

**16** $\quad S \leftarrow \text{Filter}(\{\{\Psi_r^{(\ell)}\}_{r\in[t_\ell]}\}_{\ell\in[s]}, \{(\mu^{(\ell)}, \lambda^{(\ell)})\}_{\ell\in[s]}, \varepsilon, \delta)$

**17** $\quad$ **return** $\rho \leftarrow \text{DPTHRESHOLD}(\mu, U, \alpha, \varepsilon, \delta, S)$

**18** $\text{Filter}(\{\{\Psi_r^{(\ell)}\}_{r\in[t_\ell]}\}_{\ell\in[s]}, \{(\mu^{(\ell)}, \lambda^{(\ell)})\}_{\ell\in[s]})$**:**

**19** $\quad S^{(1)} \leftarrow [n]$

**20** $\quad$ **for** epoch $\ell = 1, \dots, s$ **do**

**21** $\quad\quad \alpha^{(\ell)} \leftarrow 1/(100(0.1/C + 1.01)\lambda^{(\ell)})$

**22** $\quad\quad S_1^{(\ell)} \leftarrow S^{(\ell)}$

**23** $\quad\quad$ **for** $r = 1, \dots, t_s$ **do**

**24** $\quad\quad\quad S_{r+1}^{(\ell)} \leftarrow S_r^{(\ell)} \setminus \{i \mid i \in \mathcal{T}_{2\alpha} \text{ for } \{\tau_j = (x_j - \mu_r^{(\ell)})^\top U_r^{(\ell)}(x_j - \mu_r^{(\ell)})\}_{j\in S_r^{(\ell)}}$ and

$\quad\quad\quad\quad \tau_i \geq \rho_r^{(\ell)} Z_r^{(\ell)}\}$, where $\mathcal{T}_{2\alpha}$ is defined in Definition C.1.

$\quad$ **Output:** $S_{t_s}^{(s)}$

---

**Theorem 11** (Analysis of accuracy of DPMMWFILTER)**.** *Let $S$ be an $\alpha$-corrupted sub-Gaussian dataset, where $\alpha \leq c$ for some universal constant $c \in (0, 1/2)$. Let $S_{\text{good}}$ be $\alpha$-subgaussian good with respect to $\mu \in \mathbb{R}^d$. Suppose $\mathcal{D} = \{x_i \in \bar{x} + [-B/2, B/2]^d\}_{i=1}^n$ be the projected dataset. If $n \geq \widetilde{\Omega}\left(\frac{d^{3/2}B^2\log(2/\delta)}{\varepsilon\alpha\log 1/\alpha}\right)$, then DPMMWFILTER terminates after at most $O(\log dB^2)$ epochs and outputs $S^{(s)}$ such that with probability $0.9$, we have $|S_t^{(s)} \cap S_{\text{good}}| \geq (1 - 10\alpha)n$ and*

$$\|\mu(S^{(s)}) - \mu\|_2 \lesssim \alpha\sqrt{\log 1/\alpha}\,.$$

*Moreover, each epoch runs for at most $O(\log d)$ iterations.*

*Proof.* In $s = O(\log_{0.98}((C\alpha\log(1/\alpha))/\|M(S^{(1)}) - \mathbf{I}\|_2))$ epochs, following Lemma F.3 guarantees that we find a candidate set $S^{(s)}$ of samples with $\|M(S^{(s)}) - \mathbf{I}\|_2 \leq C\alpha\log(1/\alpha)$. We provide proof of Lemma F.3 in the Appendix F.3.

**Lemma F.3.** *Let $S$ be an $\alpha$-corrupted sub-Gaussian dataset under Assumption 1. For an epoch $s \in [T_1]$ and an iteration $t \in [T_2]$, under the hypotheses of Lemma F.4, if $S_{\text{good}}$ is $\alpha$-subgaussian good with respect to $\mu \in \mathbb{R}^d$ as in Definition G.2, $n = \widetilde{\Omega}(d^{3/2}\log(1/\delta)/(\varepsilon\alpha))$, and $|S_t^{(s)} \cap S_{\text{good}}| \geq (1 - 10\alpha)n$ then with probability $1 - O(1/\log^3 d)$ the conditions in Eqs. (14) and (15) hold. When these two conditions hold, more corrupted samples are removed in expectation than the uncorrupted*

**Algorithm 13:** Interactive version of DPMMWFILTER

**Input:** $\alpha \in (0,1)$, $T_1, T_2$, $\varepsilon_1 = \varepsilon/(4T_1)$, $\delta_1 = \delta/(4T_1)$,
$\qquad \varepsilon_2 = \min\{0.9, \varepsilon\}/(4\sqrt{10T_1 T_2 \log(4/\delta)})$, $\delta_2 = \delta/(20T_1 T_2)$

1 **if** $n < (4/\varepsilon_1)\log(1/(2\delta_1))$ **then Output:** $\emptyset$
2 **for** epoch $s = 1, 2, \ldots, T_1$ **do**
3 $\quad$ $(\mu^{(s)}, \lambda^{(s)}) \leftarrow q_{\text{spectral}}(\{\{\Psi_r^{(\ell)}\}_{r\in[t_\ell]}\}_{\ell\in[s-1]}, \{(\mu^{(\ell)}, \lambda^{(\ell)})\}_{\ell\in[s-1]}, \varepsilon_1, \delta_1)$
4 $\quad$ $n^{(s)} \leftarrow q_{\text{size}}(\{\{\Psi_r^{(\ell)}\}_{r\in[t_\ell]}\}_{\ell\in[s-1]}, \{(\mu^{(\ell)}, \lambda^{(\ell)})\}_{\ell\in[s-1]}, \varepsilon_1, \delta_1)$
5 $\quad$ **if** $n^{(s)} \leq 3n/4$ **then** terminate
6 $\quad$ **if** $\lambda^{(s)} \leq C\alpha \log(1/\alpha)$ **then**
$\qquad$ $\lfloor$ **Output:** $\mu^{(s)}$
7 $\quad$ $\alpha^{(s)} \leftarrow 1/(100(0.1/C + 1.01)\lambda^{(s)})$
8 $\quad$ $t_s \leftarrow 0$
9
10 $\quad$ **for** $t = 1, 2, \ldots, T_2$ **do**
11 $\qquad$ $(\mu_t^{(s)}, \lambda_t^{(s)}) \leftarrow q_{\text{spectral}}(\{\{\Psi_r^{(\ell)}\}_{r\in[t_\ell]}\}_{\ell\in[s]}, \{(\mu^{(\ell)}, \lambda^{(\ell)})\}_{\ell\in[s]}, \varepsilon_2, \delta_2)$
12 $\qquad$ **if** $\lambda_t^{(s)} \leq 0.5\lambda^{(s)}$ **then**
13 $\qquad\quad$ $\lfloor$ terminate epoch
14 $\qquad$ **else**
15 $\qquad\quad$ $(\Sigma_t^{(s)}, U_t^{(s)}, \psi_t^{(s)}) \leftarrow$
$\qquad\qquad$ $q_{\text{PMMW}}(\{\{\Psi_r^{(\ell)}\}_{r\in[t_\ell]}\}_{\ell\in[s]}, \{(\mu^{(\ell)}, \lambda^{(\ell)})\}_{\ell\in[s]}, \alpha^{(s)}, \mu_t^{(s)}, \varepsilon_2, \delta_2)$
16 $\qquad\quad$ **if** $\psi_t^{(s)} \leq (1/5.5)\lambda_t^{(s)}$ **then**
17 $\qquad\qquad$ $\lfloor$ $\alpha_t^{(s)} \leftarrow 0$
18 $\qquad\quad$ **else**
19 $\qquad\qquad$ $Z_t^{(s)} \leftarrow \text{Unif}([0,1])$
20 $\qquad\qquad$ $\rho_t^{(s)} \leftarrow q_{\text{1Dfilter}}(\{\{\Psi_r^{(\ell)}\}_{r\in[t_\ell]}\}_{\ell\in[s]}, \{(\mu^{(\ell)}, \lambda^{(\ell)})\}_{\ell\in[s]}, \mu_t^{(s)}, U_t^{(s)}, \alpha, \varepsilon_2, \delta_2)$
21 $\qquad\qquad$ $\lfloor$ $\alpha_t^{(s)} \leftarrow \alpha$
22 $\qquad$ $\Psi_t^{(s)} \leftarrow (\mu_t^{(s)}, \lambda_t^{(s)}, \Sigma_t^{(s)}, U_t^{(s)}, \psi_t^{(s)}, Z_t^{(s)}, \rho_t^{(s)}, \alpha_t^{(s)})$
23 $\qquad$ $t_s \leftarrow t$

$\quad$ **Output:** $\mu_{t_{T_1}}^{(T_1)}$

---

samples, i.e., $\mathbb{E}|(S_t^{(s)} \setminus S_{t+1}^{(s)}) \cap S_{\text{good}}| \leq \mathbb{E}|(S_t^{(s)} \setminus S_{t+1}^{(s)}) \cap S_{\text{bad}}|$. *Further, for an epoch $s \in [T_1]$ there exists a constant $C > 0$ such that if $\|M(S^{(s)}) - \mathbf{I}\|_2 \geq C\alpha \log(1/\alpha)$, then with probability $1 - O(1/\log^2 d)$, the $s$-th epoch terminates after $O(\log d)$ iterations and outputs $S^{(s+1)}$ such that $\|M(S^{(s+1)}) - \mathbf{I}\|_2 \leq 0.98\|M(S^{(s)}) - \mathbf{I}\|_2$.*

Lemma G.7 ensures that we get the desired bound of $\|\mu(S^{(s)}) - \mu\|_2 = O(\alpha\sqrt{\log(1/\alpha)})$ as long as $S^{(s)}$ has enough clean data, i.e., $|S^{(s)} \cap S_{\text{good}}| \geq n(1 - \alpha)$. Since Lemma F.3 gets invoked at most $O((\log d)^2)$ times, we can take a union bound, and the following argument conditions on the good events in Lemma F.3 holding, which happens with probability at least 0.99. To turn the average case guarantee of Lemma F.3 into a constant probability guarantee, we apply the optional stopping theorem. Recall that the $s$-th epoch starts with a set $S^{(s)}$ and outputs a filtered set $S_t^{(s)}$ at the $t$-th inner iteration. We measure the progress by by summing the number of clean samples removed up to epoch $s$ and iteration $t$ and the number of remaining corrupted samples, defined as $d_t^{(s)} \triangleq |(S_{\text{good}} \cap S^{(1)}) \setminus S_t^{(s)}| + |S_t^{(s)} \setminus (S_{\text{good}} \cap S^{(1)})|$. Note that $d_1^{(1)} = \alpha n$, and $d_t^{(s)} \geq 0$. At each epoch and iteration, we have

$$\mathbb{E}[d_{t+1}^{(s)} - d_t^{(s)} | d_1^{(1)}, d_2^{(1)}, \cdots, d_t^{(s)}] = \mathbb{E}\left[|S_{\text{good}} \cap (S_t^{(s)} \setminus S_{t+1}^{(s)})| - |S_{\text{bad}} \cap (S_t^{(s)} \setminus S_{t+1}^{(s)})|\right] \leq 0,$$

from part 1 of Lemma F.3. Hence, $d_t^{(s)}$ is a non-negative super-martingale. By the optional stopping theorem, at stopping time, we have $\mathbb{E}[d_t^{(s)}] \leq d_1^{(1)} = \alpha n$. By the Markov inequality, $d_t^{(s)}$ is less than $10\alpha n$ with probability 0.9, i.e., $|S_t^{(s)} \cap S_{\text{good}}| \geq (1 - 10\alpha)n$. The desired bound in Theorem 11 follows from Lemma G.7.

$\square$

## F.3 Proof of Lemma F.3

Lemma F.3 is a combination of Lemma F.4 and Lemma F.5. We state the technical lemmas and subsequently provide the proofs.

**Lemma F.4.** *For an epoch $s$ and an iteration $t$ such that $\lambda^{(s)} > C\alpha \log(1/\alpha)$, $\lambda_t^{(s)} > 0.5\lambda_0^{(s)}$, and $n^{(s)} > 3n/4$, if $n \gtrsim \frac{B^2(\log B)d^{3/2}\log(1/\delta)}{\varepsilon\alpha}$ and $|S_t^{(s)} \cap S_{\text{good}}| \geq (1 - 10\alpha)n$ then with probability $1 - O(1/\log^3 d)$, the conditions in Eqs. (14) and (15) hold. When these two conditions hold we have $\mathbb{E}|S_t^{(s)} \setminus S_{t+1}^{(s)} \cap S_{\text{good}}| \leq \mathbb{E}|S_t^{(s)} \setminus S_{t+1}^{(s)} \cap S_{\text{bad}}|$. If $n \gtrsim \frac{B^2(\log B)d^{3/2}\log(1/\delta)}{\varepsilon\alpha}$, $\psi_t^{(s)} > \frac{1}{5.5}\lambda_t^{(s)}$, and $n^{(s)} > 3n/4$, then we have with probability $1 - O(1/\log^3 d)$, $\left\langle M(S_{t+1}^{(s)}) - \mathbf{I}, U_t^{(s)} \right\rangle \leq 0.76 \left\langle M(S_t^{(s)}) - \mathbf{I}, U_t^{(s)} \right\rangle$.*

**Lemma F.5.** *For an epoch $s$ and for all $t = 0, 1, \cdots, T_2 = O(\log d)$ if Lemma F.4 holds, $n^{(s)} > 3n/4$, and $n \gtrsim \frac{B^2(\log B)d^{3/2}\log(1/\delta)}{\varepsilon\alpha}$, then we have $\|M(S^{(s+1)}) - \mathbf{I}\|_2 \leq 0.98\|M(S^{(s)}) - \mathbf{I}\|_2$ with probability $1 - O(1/\log^2 d)$.*

### F.3.1 Proof of Lemma F.4

*Proof of Lemma F.4.* To prove that we make progress for each iteration, we first show our dataset satisfies regularity conditions in Eqs. (14) and (15) that we need for DPTHRESHOLD. Following Lemma F.6 implies with probability $1 - 1/(\log^3 d)$, our scores satisfies the regularity conditions needed in Lemma E.1.

**Lemma F.6.** *For each epoch $s$ and iteration $t$, under the hypotheses of Lemma F.4, with probability $1 - O(1/\log^3 d)$, we have*

$$\frac{1}{n}\sum_{i \in S_{\text{good}} \cap \mathcal{T}_{2\alpha}} \tau_i \leq \psi/1000 \tag{14}$$

$$\frac{1}{n}\sum_{i \in S_{\text{good}} \cap S_t^{(s)}} (\tau_i - 1) \leq \psi/1000 \,, \tag{15}$$

*where $\psi \triangleq \frac{1}{n}\sum_{i \in S_t^{(s)}}(\tau_i - 1)$.*

Then by Lemma E.1 our DPTHRESHOLD gives us a threshold $\rho$ such that

$$\sum_{i \in S_{\text{good}} \cap \mathcal{T}_{2\alpha}} \mathbf{1}\{\tau_i \leq \rho\}\frac{\tau_i}{\rho} + \mathbf{1}\{\tau_i > \rho\} \leq \sum_{i \in S_{\text{bad}} \cap \mathcal{T}_{2\alpha}} \mathbf{1}\{\tau_i \leq \rho\}\frac{\tau_i}{\rho} + \mathbf{1}\{\tau_i > \rho\} \,.$$

Conditioned on the hypotheses and the claims of Lemma E.1, according to our filter rule from Algorithm 10, we have

$$\mathbb{E}|(S_t^{(s)} \setminus S_{t+1}^{(s)}) \cap S_{\text{good}}| = \sum_{i \in S_{\text{good}} \cap \mathcal{T}_{2\alpha}} \mathbf{1}\{\tau_i \leq \rho\}\frac{\tau_i}{\rho} + \mathbf{1}\{\tau_i > \rho\}$$

and

$$\mathbb{E}|(S_t^{(s)} \setminus S_{t+1}^{(s)}) \cap S_{\text{bad}}| = \sum_{i \in S_{\text{bad}} \cap \mathcal{T}_{2\alpha}} \mathbf{1}\{\tau_i \leq \rho\}\frac{\tau_i}{\rho} + \mathbf{1}\{\tau_i > \rho\} \,.$$

This implies $\mathbb{E}|(S_t^{(s)} \setminus S_{t+1}^{(s)}) \cap S_{\text{good}}| \leq \mathbb{E}|(S_t^{(s)} \setminus S_{t+1}^{(s)}) \cap S_{\text{bad}}|$. At the same time, Lemma E.1 gives us a $\rho$ such that with probability $1 - O(\log^3 d)$

$$\frac{1}{n} \sum_{i \in S_{t+1}^{(s)}} (\tau_i - 1) - 2\alpha \leq \frac{1}{n} \sum_{\tau_i \leq \rho} (\tau_i - 1) \leq \frac{3}{4} \cdot \frac{1}{n} \sum_{i \in S_t^{(s)}} (\tau_i - 1) .$$

Hence, we have

$$
\left\langle M(S_t^{(s)}) - \mathbf{I}, U_t^{(s)} \right\rangle - \left\langle M(S_{t+1}^{(s)}) - \mathbf{I}, U_t^{(s)} \right\rangle = \frac{1}{n} \sum_{i \in S_t^{(s)} \setminus S_{t+1}^{(s)}} (\tau_i - 1)
$$

$$
\geq \frac{1}{4n} \sum_{i \in S_t^{(s)}} (\tau_i - 1) - 2\alpha
$$

$$
\overset{(a)}{\geq} \frac{1}{4} \cdot \frac{998}{1000} \left\langle M(S_t^{(s)}) - \mathbf{I}, U_t^{(s)} \right\rangle ,
$$

where $(a)$ follows from our assumption on $\lambda_t$ and stopping criteria. Rearranging the terms completes the proof. $\qquad\square$

### F.3.2 Proof of Lemma F.6

*Proof of Lemma F.6.* First of all, Lemma G.9, Lemma G.10 and Lemma G.11 gives us following Lemma F.7, which basically shows with enough samples, we can make sure the noises added for privacy guarantees are small enough with probability $1 - O(1/\log^3 d)$.

**Lemma F.7.** *For $\alpha \in (0, 0.5)$, if $n \gtrsim \frac{B^2(\log B)d^{3/2}\log(1/\delta)}{\varepsilon \alpha}$ and $n^{(s)} > 3n/4$ then we have with probability $1 - O(1/\log^3 d)$, following conditions simultaneously hold:*

1. $\|\mu_t^{(s)} - \mu(S_t^{(s)})\|_2^2 \leq 0.001\alpha \log 1/\alpha$

2. $|\psi_t^{(s)} - \left\langle M(S_t^{(s)}) - \mathbf{I}, U_t^{(s)} \right\rangle| \leq 0.001\alpha \log 1/\alpha$

3. $\left| \lambda_t^{(s)} - \|M(S_t^{(s)}) - \mathbf{I}\|_2 \right| \leq 0.001\alpha \log 1/\alpha$

4. $\left| \lambda^{(s)} - \|M(S^{(s)}) - \mathbf{I}\|_2 \right| \leq 0.001\alpha \log 1/\alpha$

5. $\left\| M(S_{t+1}^{(s)}) - \Sigma_t^{(s)} \right\|_2 \leq 0.001\alpha \log 1/\alpha$

6. $\|\mu^{(s)} - \mu(S^{(s)})\|_2^2 \leq 0.001\alpha \log 1/\alpha$

Now under above conditions, since $\lambda_1^{(s)} > C\alpha \log 1/\alpha$, we have $\|M(S_t^{(s)}) - \mathbf{I}\|_2 > 0.5(C - 0.002)\alpha \log 1/\alpha$. Using the fact that $\mu(S_t^{(s)}) = (1/n)\sum_{i \in S_t^{(s)}} x_i$, we also have

$$
\frac{1}{n} \sum_{i \in S_t^{(s)}} (\tau_i - 1)
$$

$$
= \frac{1}{n} \sum_{i \in S_t^{(s)}} \left\langle \left(x_i - \mu_t^{(s)}\right)\left(x_i - \mu_t^{(s)}\right)^\top - \mathbf{I}, U_t^{(s)} \right\rangle
$$

$$
= \frac{1}{n} \sum_{i \in S_t^{(s)}} \left\langle \left(x_i - \mu(S_t^{(s)})\right)\left(x_i - \mu(S_t^{(s)})\right)^\top - \mathbf{I}, U_t^{(s)} \right\rangle
$$

$$
+ \frac{|S_t^{(s)}|}{n} \left\langle \left(\mu(S_t^{(s)}) - \mu_t^{(s)}\right)\left(\mu(S_t^{(s)}) - \mu_t^{(s)}\right)^\top, U_t^{(s)} \right\rangle
$$

$$
= \left\langle M(S_t^{(s)}) - \mathbf{I}, U_t^{(s)} \right\rangle + \frac{|S_t^{(s)}|}{n} \left\langle \left(\mu(S_t^{(s)}) - \mu_t^{(s)}\right)\left(\mu(S_t^{(s)}) - \mu_t^{(s)}\right)^\top, U_t^{(s)} \right\rangle .
$$

Thus, from the first and the second claims in Lemma F.7, we have

$$|\psi - \psi_t^{(s)}| \le 0.002\,\alpha \log 1/\alpha\,. \tag{16}$$

For an epoch $s$ and an iteration $t$, since $\alpha n \le S_{\text{good}} \cap \mathcal{T}_{2\alpha} \cap S_t^{(s)} \le 2\alpha n$, we have

$$
\begin{aligned}
\frac{1}{n} \sum_{i \in S_{\text{good}} \cap \mathcal{T}_{2\alpha} \cap S_t^{(s)}} \tau_i &= \frac{1}{n} \sum_{i \in S_{\text{good}} \cap \mathcal{T}_{2\alpha} \cap S_t^{(s)}} \left\langle (x_i - \mu_t^{(s)})(x_i - \mu_t^{(s)})^\top, U_t^{(s)} \right\rangle \\
&\overset{(a)}{\le} \frac{2}{n} \sum_{i \in S_{\text{good}} \cap \mathcal{T}_{2\alpha} \cap S_t^{(s)}} \left\langle (x_i - \mu)(x_i - \mu)^\top, U_t^{(s)} \right\rangle + \frac{2|S_{\text{good}} \cap \mathcal{T}_{2\alpha} \cap S_t^{(s)}|}{n} \left\langle (\mu - \mu_t^{(s)})(\mu - \mu_t^{(s)})^\top, U_t^{(s)} \right\rangle \\
&\overset{(b)}{\le} O(\alpha \log 1/\alpha) + 4\alpha \left\langle (\mu - \mu_t^{(s)})(\mu - \mu_t^{(s)})^\top, U_t^{(s)} \right\rangle \\
&\le O(\alpha \log 1/\alpha) + 4\alpha \|\mu_t^{(s)} - \mu\|_2^2 \\
&\le O(\alpha \log 1/\alpha) + 4\alpha \left( \|\mu - \mu(S_t^{(s)})\|_2 + \|\mu(S_t^{(s)}) - \mu_t^{(s)}\|_2 \right)^2 \\
&\overset{(c)}{\le} O(\alpha \log 1/\alpha) + 4\alpha \left( O\left(\alpha\sqrt{\log 1/\alpha}\right) + \sqrt{\alpha \left( O(\alpha \log 1/\alpha) + \|M(S_t^{(s)}) - \mathbf{I}\|_2 \right)} + \|\mu(S_t^{(s)}) - \mu_t^{(s)}\|_2 \right)^2 \\
&\le O(\alpha \log 1/\alpha) + 8\alpha^2 \left( \|M(S_t^{(s)}) - \mathbf{I}\|_2 + O(\alpha \log 1/\alpha) \right) + O(8\alpha^3 \log 1/\alpha) + 8\alpha^2 \log 1/\alpha \\
&\overset{(d)}{\le} \frac{1}{1000} \left( \frac{\|M(S_t^{(s)}) - \mathbf{I}\|_2 - 0.001\,\alpha \log 1/\alpha}{5.5} - 0.002\,\alpha \log 1/\alpha \right) \\
&\le \frac{\psi_t^{(s)} - 0.002\,\alpha \log 1/\alpha}{1000} \\
&\le \frac{\psi}{1000}\,,
\end{aligned}
$$

where $(a)$ follows from the fact that for any vector $x, y, z$, we have $(x-y)(x-y)^\top \preceq 2(x-z)(x-z)^\top + 2(y-z)(y-z)^\top$, $(b)$ follows from Lemma G.4, $(c)$ follows from Lemma G.7, $(d)$ follows from our choice of large constant $C$, and in the last inequality we used Eq. (16).

Similarly we have

$$
\frac{1}{n} \sum_{i \in S_{\text{good}} \cap S_t^{(s)}} (\tau_i - 1)
$$

$$
= \frac{1}{n} \sum_{i \in S_{\text{good}} \cap S_t^{(s)}} \left\langle (x_i - \mu_t^{(s)})(x_i - \mu_t^{(s)})^\top - \mathbf{I}, U_t^{(s)} \right\rangle
$$

$$
= \frac{1}{n} \sum_{i \in S_{\text{good}} \cap S_t^{(s)}} \left\langle \left( x_i - \mu(S_{\text{good}} \cap S_t^{(s)}) \right) \left( x_i - \mu(S_{\text{good}} \cap S_t^{(s)}) \right)^\top - \mathbf{I}, U_t^{(s)} \right\rangle
$$

$$
+ \frac{|S_{\text{good}} \cap S_t^{(s)}|}{n} \left\langle \left( \mu(S_{\text{good}} \cap S_t^{(s)}) - \mu_t^{(s)} \right) \left( \mu(S_{\text{good}} \cap S_t^{(s)}) - \mu_t^{(s)} \right)^\top, U_t^{(s)} \right\rangle
$$

$$
\overset{(a)}{\leq} O\left( \alpha \log 1/\alpha \right) + \left\| \mu(S_{\text{good}} \cap S_t^{(s)}) - \mu_t^{(s)} \right\|_2^2
$$

$$
\leq O\left( \alpha \log 1/\alpha \right) + \left( \left\| \mu(S_{\text{good}} \cap S_t^{(s)}) - \mu \right\|_2 + \left\| \mu - \mu(S_t^{(s)}) \right\|_2 \right)^2 + 0.001\, \alpha \log 1/\alpha
$$

$$
\overset{(b)}{\leq} O\left( \alpha \log 1/\alpha \right) + \left( O(\alpha \sqrt{\log 1/\alpha}) + \sqrt{\alpha(\|M(S_t^{(s)}) - \mathbf{I}\|_2 + O(\alpha \log 1/\alpha))} \right)^2 + 0.001\, \alpha \log 1/\alpha
$$

$$
\leq O\left( \alpha \log 1/\alpha \right) + \alpha \left( \|M(S_t^{(s)}) - \mathbf{I}\|_2 + O\left( \alpha \log 1/\alpha \right) \right) + O(\alpha^2 \log 1/\alpha) + + 0.001\, \alpha \log 1/\alpha
$$

$$
\overset{(c)}{\leq} \frac{1}{1000} \left( \frac{\|M(S_t^{(s)}) - \mathbf{I}\|_2 - 0.001\, \alpha \log 1/\alpha}{5.5} - 0.002\, \alpha \log 1/\alpha \right)
$$

$$
\leq \frac{\psi_t^{(s)} - 0.002\, \alpha \log 1/\alpha}{1000}
$$

$$
\leq \frac{\psi}{1000},
$$

where $(a)$ follows from Lemma G.4, $(b)$ follows from Lemma G.5 and Lemma G.7 and $(c)$ follows from our choice of large constant $C$.

$\square$

### F.3.3   Proof of Lemma F.5

*Proof of Lemma F.5.* Under the conditions of Lemma F.7, we have picked $n$ large enough such that with probability $1 - O(1/\log^3 d)$, we have

$$
\|\Sigma_{t+1}^{(s)} - \mathbf{I}\|_2 \approx_{0.01} \|M(S_{t+1}^{(s)}) - \mathbf{I}\|_2 .
$$

By Lemma F.4, we now have

$$
\left\langle M(S_{t+1}^{(s)}) - \mathbf{I}, U_t^{(s)} \right\rangle \leq 0.76 \left\langle M(S_t^{(s)}) - \mathbf{I}, U_t^{(s)} \right\rangle
$$

$$
\leq 0.76 \left\langle M(S_1^{(s)}) - \mathbf{I}, U_t^{(s)} \right\rangle
$$

$$
\leq 0.76 \|M(S_1^{(s)}) - \mathbf{I}\|_2 . \tag{17}
$$

Since $\lambda_1^{(s)} > C \alpha \log 1/\alpha$, we have $\|M(S_{t+1}^{(s)}) - \mathbf{I}\|_2 > 0.5(C - 0.002) \alpha \log 1/\alpha$. Combining the above inequality and the fifth claim of Lemma F.7 together, we have

$$
\left\langle \Sigma_{t+1}^{(s)} - \mathbf{I}, U_t^{(s)} \right\rangle \leq \left\langle M(S_{t+1}^{(s)}) - \mathbf{I}, U_t^{(s)} \right\rangle + \|\Sigma_{t+1}^{(s)} - M(S_{t+1}^{(s)})\|_2 \leq 0.77 \|M(S_1^{(s)}) - \mathbf{I}\|_2 .
$$

By Lemma G.1, we have $M(S_{t+1}^{(s)}) - \mathbf{I} \preceq M(S_1^{(s)}) - \mathbf{I}$. By our choice of $\alpha^{(s)}$, we have $\alpha^{(s)} \left( M(S_{t+1}^{(s)}) - \mathbf{I} \right) \preceq \frac{1}{100} \mathbf{I}$ and $\alpha^{(s)} \left( \Sigma_{t+1}^{(s)} - \mathbf{I} \right) \preceq \frac{1}{100} \mathbf{I}$. Therefore, by Lemma G.14, we have

$$\left\| \sum_{t=1}^{T_2} \Sigma_{t+1}^{(s)} - \mathbf{I} \right\|_2$$

$$\leq \sum_{t=1}^{T_2} \left\langle \Sigma_{t+1}^{(s)} - \mathbf{I}, U_t^{(s)} \right\rangle + \alpha^{(s)} \sum_{t=1}^{T_2} \left\langle U_t^{(s)}, \left| \Sigma_{t+1}^{(s)} - \mathbf{I} \right| \right\rangle \|\Sigma_{t+1}^{(s)} - \mathbf{I}\|_2 + \frac{\log(d)}{\alpha^{(s)}}$$

$$\overset{(a)}{\leq} \sum_{t=1}^{T_2} \left\langle \Sigma_{t+1}^{(s)} - \mathbf{I}, U_t^{(s)} \right\rangle + \frac{1}{100} \sum_{t=1}^{T_2} \left\langle U_t^{(s)}, \left| \Sigma_{t+1}^{(s)} - \mathbf{I} \right| \right\rangle + 200 \log(d) \|M(S_1^{(s)}) - \mathbf{I}\|_2$$

where $(a)$ follows from our choice of $\alpha^{(s)}$ and $C$. By Lemma G.6, $M(S_{t+1}^{(s)}) - \mathbf{I} \succeq -c_1 \alpha \log 1/\alpha \cdot I$ for $t = 1, 2, \cdots, T_2$, we have

$$|M(S_{t+1}^{(s)}) - \mathbf{I}| \preceq M(S_{t+1}^{(s)}) - \mathbf{I} + 2c_1 \alpha \log 1/\alpha \, \mathbf{I},$$

and hence

$$\left\langle U_t^{(s)}, \left| M(S_{t+1}^{(s)}) - \mathbf{I} \right| \right\rangle \leq \left\langle U_t^{(s)}, M(S_{t+1}^{(s)}) - \mathbf{I} \right\rangle + 2c_1 \alpha \log 1/\alpha$$

Meanwhile, we have

$$M(S_{t+1}^{(s)}) - \mathbf{I} - \|\Sigma_{t+1}^{(s)} - M(S_{t+1}^{(s)})\|_2 \, \mathbf{I} \preceq \Sigma_{t+1}^{(s)} - \mathbf{I} \preceq M(S_{t+1}^{(s)}) - \mathbf{I} + \|\Sigma_{t+1}^{(s)} - M(S_{t+1}^{(s)})\|_2 \, \mathbf{I} \,.$$

Hence,

$$|\Sigma_{t+1}^{(s)} - \mathbf{I}| \preceq M(S_{t+1}^{(s)}) - \mathbf{I} + (3\|\Sigma_{t+1}^{(s)} - M(S_{t+1}^{(s)})\|_2 + 2c_1 \alpha \log 1/\alpha) \, \mathbf{I}$$

Together with Eq. (17), we have

$$\left\langle U_t^{(s)}, \left| \Sigma_{t+1}^{(s)} - \mathbf{I} \right| \right\rangle$$

$$\leq \left\langle U_t^{(s)}, M(S_{t+1}^{(s)}) - \mathbf{I} \right\rangle + 3\|\Sigma_{t+1}^{(s)} - M(S_{t+1}^{(s)})\|_2 + 2c_1 \alpha \log 1/\alpha$$

$$\leq 0.79 \left\| M(S_1^{(s)}) - \mathbf{I} \right\|_2 + 2c_1 \alpha \log 1/\alpha \,.$$

By Lemma G.6, we have $M(S_{t+1}^{(s)}) - \mathbf{I} \succeq -c_1 \alpha \log 1/\alpha \, \mathbf{I}$. Also, we know $M(S_{t+1}^{(s)}) - \mathbf{I} \preceq M(S_1^{(s)}) - \mathbf{I}$. Then we have

$$\left\| M(S_{T_2+1}^{(s)}) - \mathbf{I} \right\|_2$$

$$\leq \frac{1}{T_2} \left\| \sum_{i=1}^{T_2} M(S_{t+1}^{(s)}) - \mathbf{I} \right\|_2$$

$$\leq \frac{1}{T_2} \left\| \sum_{i=1}^{T_2} \Sigma_{t+1}^{(s)} - \mathbf{I} \right\|_2 + 0.001 \, \alpha \log 1/\alpha$$

$$\leq \frac{1}{T_2} \left( \sum_{t=1}^{T_2} \left\langle \Sigma_{t+1}^{(s)} - \mathbf{I}, U_t^{(s)} \right\rangle + \frac{1}{100} \sum_{t=1}^{T_2} \left\langle U_t^{(s)}, \left| \Sigma_{t+1}^{(s)} - \mathbf{I} \right| \right\rangle + 200 \log(d) \|M(S_1^{(s)}) - \mathbf{I}\|_2 \right) + 0.001 \, \alpha \log 1/\alpha$$

$$\leq 0.79 \|M(S_1^{(s)}) - \mathbf{I}\|_2 + 2c_1 \alpha \log 1/\alpha + \frac{200 \log(d)}{T_2} \|M(S_1^{(s)}) - \mathbf{I}\|_2 + 0.001 \, \alpha \log 1/\alpha$$

$$\leq 0.98 \, \|M(S_1^{(s)}) - \mathbf{I}\|_2 \,,$$

where the last inequality follows from our assumption that $\lambda_0^{(s)} > C\alpha \log 1/\alpha$, and conditions of Lemma F.7 hold and we have $\|M(S_{t+1}^{(s)}) - \mathbf{I}\|_2 > 0.5(C - 0.002)\alpha \log 1/\alpha$. $\qquad \square$

# G Technical lemmas

## G.1 Lemmata for sub-Gaussian regularity from [36]

**Lemma G.1** ([36, Lemma 3.4] ). *If $S' \subset S$, then $M(S') \preceq M(S)$.*

**Definition G.2** ([36, Definition 4.1] ). *Let $D$ be a distribution with mean $\mu \in \mathbb{R}^d$ and covariance $\mathbf{I}$. For $0 < \alpha < 1/2$, we say a set of points $S = \{X_1, X_2, \cdots, X_n\}$ is $\alpha$-subgaussian good with respect to $\mu \in \mathbb{R}^d$ if following inequalities are satisfied:*

- $\|\mu(S) - \mu\|_2 \lesssim \alpha\sqrt{\log 1/\alpha}$ *and* $\left\|\frac{1}{|S|} \sum_{i \in S} (X_i - \mu(S)) (X_i - \mu(S))^\top - \mathbf{I}\right\|_2 \lesssim \alpha \log 1/\alpha$.

- *for any subset $T \subset S$ so that $|T| = 2\alpha|S|$, we have*

$$\left\|\frac{1}{|T|} \sum_{i \in T} X_i - \mu\right\|_2 \lesssim \sqrt{\log 1/\alpha} \quad and \quad \left\|\frac{1}{|T|} \sum_{i \in T} (X_i - \mu(S)) (X_i - \mu(S))^\top - \mathbf{I}\right\|_2 \lesssim \log 1/\alpha .$$

**Lemma G.3** ([36, Lemma 4.1] ). *A set of i.i.d. samples from an identity covariance sub-Gaussian distribution of size $n = \Omega\left(\frac{d + \log 1/\delta}{\alpha^2 \log 1/\alpha}\right)$ is $\alpha$-subgaussian good with respect to $\mu$ with probability $1 - \delta$.*

**Lemma G.4** ([36, Fact 4.2] ). *Let $S$ be an $\alpha$-corrupted sub-Gaussian dataset under Assumption 1. If $S_{\text{good}}$ is $\alpha$-subgaussian good with respect to $\mu \in \mathbb{R}^d$, then for any $T \subset S$ such that $|T| \leq 2\alpha|S|$, we have for any unit vector $v \in \mathbb{R}^d$*

$$\frac{1}{|S|} \sum_{X_i \in T} \langle (X_i - \mu), v\rangle^2 \lesssim \alpha \log 1/\alpha .$$

*For any subset $T \subset S$ such that $|T| \geq (1 - 2\alpha)|S|$, we have*

$$\left\|\frac{1}{|S|} \sum_{i \in T} (x_i - \mu)(x_i - \mu)^\top - \mathbf{I}\right\|_2 \lesssim \alpha \log 1/\alpha \quad and \ ,$$

$$\left\|\frac{1}{|S|} \sum_{i \in T} (x_i - \mu(T))(x_i - \mu(T))^\top - \mathbf{I}\right\|_2 \lesssim \alpha \log 1/\alpha$$

**Lemma G.5** ([36, Corollary 4.3] ). *Let $S$ be an $\alpha$-corrupted sub-Gaussian dataset under Assumption 1. If $S_{\text{good}}$ is $\alpha$-subgaussian good with respect to $\mu \in \mathbb{R}^d$, then for any $T \subset S$ such that $|T| \leq 2\alpha|S|$, we have*

$$\left\|\frac{1}{|S|} \sum_{X_i \in T} (X_i - \mu)\right\|_2 \lesssim \alpha\sqrt{\log 1/\alpha} .$$

*For any subset $T \subset S$ such that $|T| \geq (1 - 2\alpha)|S|$, we have*

$$\|\mu(T) - \mu\|_2 \lesssim \alpha\sqrt{\log 1/\alpha} .$$

**Lemma G.6** ([36, Lemma 4.5] ). *Let $S$ be an $\alpha$-corrupted sub-Gaussian dataset under Assumption 1. If $S_{\text{good}}$ is $\alpha$-subgaussian good with respect to $\mu \in \mathbb{R}^d$, then for any $T \subset S$ such that $|T \cap S_{\text{good}}| \geq (1 - 2\alpha)|S|$, then there is some universal constant $c_1$ such that*

$$\frac{1}{|S|} \sum_{i \in T} (x_i - \mu(T)) (x_i - \mu(T))^\top \succeq (1 - c_1 \alpha \log 1/\alpha)\mathbf{I} .$$

**Lemma G.7** ([36] Lemma 4.6 ). *Let $S$ be an $\alpha$-corrupted sub-Gaussian dataset under Assumption 1. If $S_{\text{good}}$ is $\alpha$-subgaussian good with respect to $\mu \in \mathbb{R}^d$, then for any $T \subset S$ such that $|T \cap S_{\text{good}}| \geq (1 - 2\alpha)|S|$, we have*

$$\|\mu(T) - \mu\|_2 \leq \frac{1}{1 - \alpha} \cdot \left(\sqrt{\alpha\left(\|M(T) - \mathbf{I}\|_2 + O\left(\alpha \log 1/\alpha\right)\right)} + O\left(\alpha\sqrt{\log 1/\alpha}\right)\right) .$$

## G.2 Auxiliary Lemmas on Laplace and Gaussian mechanism

**Lemma G.8** (Theorem A.1 in [38]). *Let $\varepsilon \in (0,1)$ be arbitrary. For $c^2 \geq 2\ln(1.25/\delta)$, the Gaussian Mechanism with parameter $\sigma^2 \geq c^2 \Delta_2 f / \varepsilon$ is $(\varepsilon, \delta)$-differentially private.*

**Lemma G.9.** *Let $Y \sim \mathrm{Lap}(b)$. Then for all $h > 0$, we have $\mathbb{P}(|Y| \geq hb) = e^{-h}$.*

**Lemma G.10** (Tail bound of $\chi$-square distribution [77]). *Let $x_i \sim \mathcal{N}(0, \sigma^2)$ for $i = 1, 2, \cdots, d$. Then for all $\zeta \in (0,1)$, we have $\mathbb{P}(\|X\|_2 \geq \sigma\sqrt{d\log(1/\zeta)}) \leq \zeta$.*

**Lemma G.11** ([75, Corollary 2.3.6] ). *Let $Z \in \mathbb{R}^{d \times d}$ be a matrix such that $Z_{i,j} \sim \mathcal{N}(0, \sigma^2)$ for $i \geq j$ and $Z_{i,j} = Z_{j,i}$ for $i < j$. For $\forall \zeta \in (0,1)$, then with probability $1 - \zeta$ we have $\|Z\|_2 \leq \sigma\sqrt{d\log(1/\zeta)}$.*

**Lemma G.12** (Accuracy of the histogram using Gaussian Mechanism). *Let $f : \mathcal{X}^n \to \mathbb{R}^{\mathcal{S}}$ be a histogram over $K$ bins. For any dataset $D \in \mathcal{X}^n$ and $\varepsilon$, Gaussian Mechanism is an $(\varepsilon, \delta)$-differentially private algorithm $M(D)$ such that given*

*with probability $1 - \zeta$ we have*

$$\|M(D) - f(D)\|_\infty \leq O\left(\frac{\sqrt{\log(K/\zeta)\log(1/\delta)}}{\varepsilon n}\right) .$$

*Proof.* First notice that the $\ell_2$ sensitivity of histogram function $f$ is $\sqrt{2}/n$. Thus, by Lemma G.8, by adding noise $\mathcal{N}(0, (\frac{2\sqrt{2\log(1.25/\delta)}}{n\varepsilon})^2)$ to each entry of $f$, we have a $(\varepsilon, \delta)$ differentially private algorithm. Since Gaussian tail bound implies that $\mathbb{P}_{x \sim \mathcal{N}(0,\sigma^2)}[x \geq \Omega(\sqrt{\log(K/\eta)}\sigma)] \leq \eta/K$, we have that with probability $1 - \eta$, the $\ell_\infty$ norm of the added noise is bounded by $O(\frac{\sqrt{\log(1/\delta)\log(K/\eta)}}{n\varepsilon})$. This concludes the proof. $\square$

**Lemma G.13** (Composition theorem of [51, Theorem 3.4]). *For $\varepsilon \leq 0.9$, an end-to-end guarantee of $(\varepsilon, \delta)$-differential privacy is satisfied if a dataset is accessed $k$ times, each with a $(\varepsilon/2\sqrt{2k\log(2/\delta)}, \delta/2k)$-differential private mechanism.*

## G.3 Analysis of $\|M(S_t^{(s)}) - \mathbf{I}\|_2$ shrinking

For any symmetric matrix $A = \sum_{i=1}^d \lambda_i v_i v_i^\top$, we let $|A|$ denote $|A| = \sum_{i=1}^d |\lambda_i| v_i v_i^\top$.

**Lemma G.14** (Regret bound, Special case of [4, Theorem 3.1]). *Let*

$$U_t = \frac{\exp(\alpha \sum_{k=1}^{t-1}(\Sigma_k - \mathbf{I}))}{\mathrm{Tr}(\exp(\alpha \sum_{k=1}^{t-1}(\Sigma_k - \mathbf{I})))} ,$$

*and $\alpha$ satisfies $\alpha(\Sigma_{t+1} - \mathbf{I}) \preceq I$ for all $k \in [T]$, then for all $U \succeq 0$, $\mathrm{Tr}(U) = 1$, it holds that*

$$\sum_{t=1}^T \langle (\Sigma_{t+1} - \mathbf{I}), U - U_t \rangle \leq \alpha \sum_{t=1}^T \langle |(\Sigma_{t+1} - \mathbf{I}), U_t| \rangle \cdot \|(\Sigma_{t+1} - \mathbf{I})\|_2 + \frac{\log d}{\alpha}.$$

*Rearranging terms, and taking a supremum over $U$, we obtain that*

$$\|\sum_{t=1}^T (\Sigma_{t+1} - \mathbf{I})\|_2 \leq \sum_{t=1}^T \langle U_t, (\Sigma_{t+1} - \mathbf{I}) \rangle + \alpha \sum_{t=1}^T \langle |(\Sigma_{t+1} - \mathbf{I}), U_t| \rangle \cdot \|(\Sigma_{t+1} - \mathbf{I})\|_2 + \frac{\log d}{\alpha}.$$

## H Exponential time DP robust mean estimation of sub-Gaussian and heavy tailed distributions (Algorithm 2)

In this section, we give a self-contained proof of the privacy and utility of our exponential time robust mean estimation algorithm for sub-Gaussian and heavy tailed distributions. The proof relies on the resilience property of the uncorrupted data as shown in the following lemmas.

**Lemma H.1** (Lemma 10 in [73]). *If a set of points $\{x_i\}_{i \in S}$ lying in $\mathbb{R}^d$ is $(\sigma, \alpha)$-resilient around a point $\mu$, then*

$$\|\frac{1}{|T'|} \sum_{i \in T'} (x_i - \mu)\|_2 \leq \frac{2 - \alpha}{\alpha} \sigma.$$

*for all sets $T'$ of size at least $\alpha |S|$.*

**Lemma H.2** (Finite sample resilience of sub-Gaussian distributions [81, Theorem G.1]). *Let $S_{\text{good}}$ be a set of i.i.d. points from a sub-Gaussian distribution $\mathcal{D}$ with a parameter $\mathbf{I}_d$. Given that $|S_{\text{good}}| = \Omega((d + \log(1/\zeta))/(\alpha^2 \log 1/\alpha))$, $S_{\text{good}}$ is $(\alpha\sqrt{\log(1/\alpha)}, \alpha)$-resilient around its mean $\mu$ with probability $1 - \zeta$.*

**Lemma H.3** (Finite sample resilience of heavy-tailed distributions [81, Theorem G.2]). *Let $S_{\text{good}}$ be a set of i.i.d. samples drawn from distribution $\mathcal{D}$ whose mean and covariance are $\mu, \Sigma$ respectively, and that $\Sigma \preceq I$. Given that $|S| = \Omega(d/(\zeta\alpha))$, there exists a constant $c_\zeta$ that only depends on $\zeta$ such that $S_{\text{good}}$ is $(c_\zeta\sqrt{\alpha}, \alpha)$-resilient around $\mu$ with probability $1 - \zeta$.*

### H.1 Case of heavy-tailed distributions and a proof of Theorem 8

Lemma K.1 ensures that $q_{\text{range-ht}}$ returns samples in a bounded support of Euclidean distance $\sqrt{d}B/2$ with $B = 50/\sqrt{\alpha}$ where $(1 - 2\alpha)n$ samples are uncorrupted ($\alpha n$ is corrupted by adversary and $\alpha n$ can be corrupted by the pre-processing step). For a $(c_\zeta\sqrt{3\alpha}, 3\alpha)$-resilient dataset, we first show that $R(S)$ is robust against corruption.

**Lemma H.4** ($\alpha$-corrupted data has small $R(S)$). *Let $S$ be the set of $2\alpha$-corrupted data. Given that $n = \Omega(d/(\zeta\alpha))$, with probability $1 - \zeta$, $R(S) \leq c_\zeta\sqrt{3\alpha}$.*

This follows immediately by selecting $S'$ to be the uncorrupted $(1 - 2\alpha)$ fraction of the dataset and applying $(c_\zeta\sqrt{3\alpha}, 3\alpha)$-resilience. After pre-processing, we have that $\|x_i - \bar{x}\|_2 \leq B\sqrt{d}/2$, and then clearly $R(\cdot)$ has sensitivity $\Delta_R \leq B\sqrt{d}/n$.

**Lemma H.5** (Sensitivity and Privacy of $\hat{R}(S)$). *Given that $\hat{R}(S) = R(S) + \text{Lap}(\frac{3B\sqrt{d}}{n\varepsilon})$, $\hat{R}(S)$ is $(\varepsilon/3, 0)$-differentially private. Further, with probability $1 - \delta/3$, $|\hat{R}(S) - R(S)| \leq \frac{3B\sqrt{d}\log(3/\delta)}{n\varepsilon}$.*

In the algorithm, we first compute $\hat{R}(S)$. If $\hat{R}(S) \geq 2c_\zeta\sqrt{\alpha}$, we stop and output $\emptyset$. Otherwise, we use exponential mechanism with score function $d(\hat{\mu}, S)$ to find an estimate $\hat{\mu}$. We prove the privacy guarantee of our algorithm as follows.

**Lemma H.6** (Privacy). *Algorithm 2 is $(\varepsilon, \delta)$-differentially private if $n \geq 6B\sqrt{d}\log(3/\delta)/(c_\zeta\varepsilon\sqrt{\alpha})$.*

*Proof.* We consider neighboring datasets $S, S'$ under the following two scenario

1. $R(S) > 3c_\zeta\sqrt{\alpha}$

   In this case, given that $n \geq \frac{6B\sqrt{d}\log(3/\delta)}{c_\zeta\sqrt{\alpha}\varepsilon}$, we have $\hat{R}(S) > 2c_\zeta\sqrt{\alpha}$ and the output of the algorithm $\mathcal{A}(S) = \emptyset$ with probability at least $1 - \delta/3$, and $\mathcal{A}(S') = \emptyset$ with probability at least $1 - \delta/3$. Thus, for any set $Q$, $\mathbb{P}[\mathcal{A}(S) \in Q] \leq \mathbb{P}[\mathcal{A}(S') \in Q] + \delta/3$.

2. $R(S) \leq 3c_\zeta\sqrt{\alpha}$

   **Lemma H.7** (Sensitivity of $d(\hat{\mu}, S)$). *Given that $R(S) \leq 3c_\zeta\sqrt{\alpha}$, for any neighboring dataset $S'$, $|d(\hat{\mu}, S) - d(\hat{\mu}, S')| \leq 12c_\zeta/(n\sqrt{\alpha})$.*

In this case, the privacy guarantee of $\hat{R}(S)$ yields that $\mathbb{P}[\hat{R}(S) \in Q] \leq \exp(\varepsilon/3) \cdot \mathbb{P}[\hat{R}(S') \in Q]$. Lemma H.7 yields that $\mathbb{P}[\hat{\mu}(S) \in Q] \leq \exp(\varepsilon) \cdot \mathbb{P}[\hat{\mu}(S') \in Q]$. A simple composition of the privacy guarantee with $q_{\mathrm{range-ht}}(\cdot)$ and the exponential mechanism gives that

$$\mathbb{P}[(\hat{R}(S), \hat{\mu}(S)) \in Q] \leq \exp(\varepsilon) \cdot \mathbb{P}[(\hat{R}(S'), \hat{\mu}(S')) \in Q] + \delta/3$$

This implies that $\mathbb{P}[\mathcal{A}(S) \in Q] \leq \exp(\varepsilon) \cdot \mathbb{P}[\mathcal{A}(S') \in Q] + \delta/3$.

$\square$

**Lemma H.8** (Utility of the algorithm). *For an $2\alpha$-corrupted dataset $S$, Algorithm 2 achieves $\|\hat{\mu} - \mu^*\|_2 \leq c_\zeta \sqrt{\alpha}$ with probability $1 - \zeta$, if $n = \Omega(d/(\alpha\zeta) + (d\log(d/\alpha^{1.5}) + \log(1/\zeta))/(\varepsilon\alpha))$.*

*Proof of Lemma H.8.* We use the following lemma showing that $d(\hat{\mu}, S)$ is a good approximation of $\|\hat{\mu} - \mu^*\|_2$.

**Lemma H.9** ($d(\mu, S)$ approximates $\|\mu - \mu^*\|$). *Let $S$ be the set of $2\alpha$-corrupted data. Given that $n = \Omega(d/(\zeta\alpha))$, with probability $1 - \zeta$,*

$$\left| d(\hat{\mu}, S) - \|\hat{\mu} - \mu^*\|_2 \right| \leq 7c_\zeta \sqrt{\alpha} .$$

This implies that the exponential mechanism achieves the following bounds.

$$\mathbb{P}(\|\hat{\mu} - \mu^*\| \leq c_\zeta \sqrt{\alpha}) \geq \frac{1}{A} e^{-\frac{\varepsilon\alpha n}{3}} \mathrm{Vol}(c_\zeta \sqrt{\alpha}, d), \text{ and}$$

$$\mathbb{P}(\|\hat{\mu} - \mu^*\| \geq 22c_\zeta \sqrt{\alpha}) \leq \frac{1}{A} e^{-\frac{5\varepsilon\alpha n}{8}} B^d ,$$

where $A$ denotes the normalizing factor for the exponential mechanism and $\mathrm{Vol}(r, d)$ is the volume of a ball of radius $r$ in $d$ dimensions. It follows that

$$\log\left(\frac{\mathbb{P}(\|\hat{\mu} - \mu^*\|_2 \leq c_\zeta \sqrt{\alpha})}{\mathbb{P}(\|\hat{\mu} - \mu^*\|_2 \geq 22c_\zeta \sqrt{\alpha})}\right) \geq \frac{7}{24}\varepsilon\alpha n - C\,d\log(dB/\alpha)$$

$$\geq \log(1/\zeta) ,$$

for $n = \Omega((d\log(d/\alpha^{1.5}) + \log(1/\zeta))/(\varepsilon\alpha))$.

$\square$

### H.1.1 Proof of Lemma H.7

Since $R(S) \leq 3c_\zeta \sqrt{\alpha}$, define $S_{\mathrm{good}}$ as the minimizing subset in Definition 4.2 such that

$$R(S) = \max_{T \subset S_{\mathrm{good}}, |T| = (1-\alpha)|S_{\mathrm{good}}|} \|\mu(T) - \mu(S_{\mathrm{good}})\|_2 .$$

By this definition of $S_{\mathrm{good}}$ and Lemma H.1,

$$|v^\top(\mu(S_{\mathrm{good}} \cap \mathcal{T}^v) - \mu(S_{\mathrm{good}}))| \leq 6c_\zeta \sqrt{1/\alpha}, \text{ and}$$

$$|v^\top(\mu(S_{\mathrm{good}} \cap \mathcal{B}^v) - \mu(S_{\mathrm{good}}))| \leq 6c_\zeta \sqrt{1/\alpha}.$$

Therefore,

$$\min_{i \in S_{\mathrm{good}} \cap \mathcal{T}^v} |v^\top(x_i - \mu(S_{\mathrm{good}}))| \leq |v^\top(\mu(S_{\mathrm{good}} \cap \mathcal{T}^v) - \mu(S_{\mathrm{good}}))| \leq 6c_\zeta \sqrt{1/\alpha},$$

and similarly

$$\min_{i \in S_{\mathrm{good}} \cap \mathcal{B}^v} |v^\top(x_i - \mu(S_{\mathrm{good}}))| \leq |v^\top(\mu(S_{\mathrm{good}} \cap \mathcal{B}^v) - \mu(S_{\mathrm{good}}))| \leq 6c_\zeta \sqrt{1/\alpha}$$

This implies

$$\min_{i \in S_{\mathrm{good}} \cap \mathcal{T}^v} v^\top x_i - \max_{i \in S_{\mathrm{good}} \cap \mathcal{B}^v} v^\top x_i \leq 12c_\zeta \sqrt{1/\alpha} . \tag{18}$$

This implies that distribution of one-dimensional points $S_{(v)} = \{v^\top x_i\}$ is dense at the boundary of top and bottom $\alpha$ quantiles, and hence cannot be changed much by changing one entry. Formally, consider a neighboring dataset $S'$ (and the corresponding $S'_{(v)}$) where one point $x_i$ in $\mathcal{M}^{(v)}(S)$ is replaced by another point $\tilde{x}_i$. If $v^\top \tilde{x}_i \in [\max_{i \in S_{\text{good}} \cap \mathcal{B}^v} v^\top x_i, \min_{i \in S_{\text{good}} \cap \mathcal{T}^v} v^\top x_i]$, then Eq. (18) implies that this only changes the mean by $6c_\zeta/(\sqrt{\alpha}n)$. Otherwise, $\mathcal{M}^v(S')$ will have $x_i$ replaced by either $\arg\min_{i \in S_{\text{good}} \cap \mathcal{T}^v} v^\top x_i$ or $\arg\max_{i \in S_{\text{good}} \cap \mathcal{B}^v} v^\top x_i$. In both cases, Eq. (18) implies that this only changes the mean by $12c_\zeta/(\sqrt{\alpha}n)$. The other case of when the replaced sample $x_i \in S$ is not in $\mathcal{M}^v(S)$ follows similarly. From this, we upper bounds the maximum difference between $S$ and $S'$ when projected on $v$, that is

$$\left| v^\top \left( \mu(\mathcal{M}^v(S)) - \mu(\mathcal{M}^v(S')) \right) \right| \le \frac{12c_\zeta}{\sqrt{\alpha}n} \ .$$

This implies the sensitivity of $d(\mu, S)$ is bounded by $6c_\zeta/(\sqrt{\alpha}n)$:

$$
\begin{aligned}
|d(\mu, S) - d(\mu, S')| &= \left| \max_{v \in \mathbb{S}^{d-1}} v^\top \mu(M^v(S)) - \max_{\tilde{v} \in \mathbb{S}^{d-1}} \tilde{v}^\top \mu(M^v(S')) \right| \\
&\le \max_{v \in \mathbb{S}^{d-1}} \left| v^\top (\mu(M^v(S)) - \mu(M^v(S'))) \right| \le \frac{12c_\zeta}{\sqrt{\alpha}\, n}
\end{aligned}
$$

### H.1.2 Proof of Lemma H.9

First we show $|v^\top (\mu(\mathcal{M}^v) - \mu^*)| \le 7c_\zeta\sqrt{\alpha}$. Notice that $|S_{\text{good}} \cap \mathcal{T}^v| \le 3\alpha|S|$, and $|S_{\text{good}} \cap \mathcal{B}^v| \le 3\alpha|S|$. By the $(c_\zeta\sqrt{3\alpha}, 3\alpha)$-resilience property, we have $|v^\top (\mu(S_{\text{good}} \cap \mathcal{T}^v) - \mu^*)| \le c_\zeta\sqrt{3/\alpha}$, and $|v^\top (\mu(S_{\text{good}} \cap \mathcal{B}^v) - \mu^*)| \le c_\zeta\sqrt{3/\alpha}$. Since $|S_{\text{good}} \cap \mathcal{M}^v| \ge (1 - 8\alpha)|S_{\text{good}}|$, by the $(c_\zeta\sqrt{8\alpha}, 8\alpha)$-resilience property,

$$|v^\top (\mu(S_{\text{good}} \cap \mathcal{M}^v) - \mu^*)| \le c_\zeta\sqrt{8\alpha} \ .$$

Since $\mathcal{T}^v, \mathcal{B}^v$ are the largest and smallest $3\alpha n$ points respectively and $|S_{\text{bad}}| \le 2\alpha n$, we get

$$|v^\top (\mu(S_{bad} \cap \mathcal{M}^v) - \mu^*)| \le 2c_\zeta\sqrt{3/\alpha}.$$

Combining $S_{\text{good}} \cap \mathcal{M}^v$ and $S_{\text{bad}} \cap \mathcal{M}^v$ we get

$$
\begin{aligned}
&|v^\top (\mu(\mathcal{M}^v) - \mu^*)| \\
&\le \frac{|S_{bad} \cap \mathcal{M}^v|}{|\mathcal{M}^v|} |v^\top (\mu(S_{bad} \cap \mathcal{M}^v) - \mu^*)| + \frac{|\mu(S_{\text{good}} \cap \mathcal{M}^v|}{|\mathcal{M}^v|} |v^\top (\mu(S_{\text{good}} \cap \mathcal{M}^v) - \mu^*)| \\
&\le 7c_\zeta\sqrt{\alpha}.
\end{aligned}
$$

Finally we get that

$$
\begin{aligned}
\left| d(\hat{\mu}, S) - \|\hat{\mu} - \mu^*\|_2 \right| &\overset{(a)}{=} \left| \max_{v \in \mathbb{S}^{d-1}} \left| v^\top \left( \mu(\mathcal{M}^{(v)}) - \hat{\mu} \right) \right| - \max_{v \in \mathbb{S}^{d-1}} |v^\top (\hat{\mu} - \mu^*)| \right| \\
&\overset{(b)}{\le} \max_{v \in \mathbb{S}^{d-1}} \left| v^\top \left( \mu(\mathcal{M}^{(v)}) - \mu^* \right) \right| \\
&\le 7c_\zeta\sqrt{\alpha},
\end{aligned}
$$

where $(a)$ holds by the definition of the distance :

$$\|\mu - \mu^*\|_2 = \max_{v \in \mathbb{S}^{d-1}} |v^\top (\mu - \mu^*)|,$$

and $(b)$ holds by triangle inequality.

### H.2 Case of sub-Gaussian distributions and a proof of Theorem 7

Th proof is analogous to the previous section, we only state the lemmas that differ. $q_{\text{range}}$ returns a hypercube $\bar{x} + [-B/2, B/2]^d$ that includes all uncorrupted data points with a high probability.

**Lemma H.10** ($\alpha$-corrupted data has small $R(S)$)**.** *Let $S$ be the set of $\alpha$-corrupted data. Given that $n = \Omega(\frac{d + \log(1/\zeta)}{\alpha^2 \log 1/\alpha})$, with probability $1 - \zeta$, $R(S) \leq 3\,\alpha\sqrt{\log(1/3\alpha)}$.*

**Lemma H.11** (Privacy)**.** *Algorithm 2 is $(\varepsilon, \delta)$-differentially private if $n \geq 3B\sqrt{d}\log(3/\delta)/(\varepsilon\alpha\sqrt{\log(1/\alpha)})$.*

This follows from the following lemma.

**Lemma H.12** (Sensitivity of $d(\hat{\mu}, S)$)**.** *Given that $R(S) \leq 3\alpha\sqrt{\log(1/\alpha)}$, for any neighboring dataset $S'$, $|d(\hat{\mu}, S) - d(\hat{\mu}, S')| \leq 12\sqrt{\log 1/\alpha}/n$.*

**Lemma H.13** ($d(\hat{\mu}, S)$ approximates $\|\hat{\mu} - \mu^*\|$)**.** *Let $S$ be the set of $\alpha$-corrupted data. Given that $n = \Omega(\frac{d + \log(1/\zeta)}{\alpha^2 \log 1/\alpha})$, with probability $1 - \zeta$,*

$$\left| d(\hat{\mu}, S) - \|\hat{\mu} - \mu^*\|_2 \right| \;\; \leq \;\; 14\,\alpha\,\sqrt{\log 1/\alpha} \;.$$

This implies the following utility bound.

**Lemma H.14** (Utility of the algorithm)**.** *For an $\alpha$-corrupted dataset $S$, Algorithm 2 achieves $\|\hat{\mu} - \mu^*\|_2 \leq \alpha\sqrt{\log 1/\alpha}$ with probability $1 - \zeta$, if $n = \Omega((d + \log(1/\zeta))/(\alpha^2 \log(1/\alpha)) + (d \log(d\sqrt{\log(dn/\zeta)}/\alpha) + \log(1/\zeta)/(\varepsilon\alpha))$.*

## I  Background on exponential time approaches for Gaussian distributions

In this section, we provide a background on exponential time algorithms that achieve optimal guarantees but only applies to and heavily relies on the assumption that samples are drawn from a *Gaussian* distribution. In §4, we introduce a novel exponential time approach that seamlessly generalizes to both sub-Gaussian and covariance-bounded distributions.

We introduce Algorithm 1, achieving the optimal sample complexity of $\widetilde{O}(d/\min\{\alpha\varepsilon, \alpha^2\})$ (Theorem 12). The main idea is to find an approximate Tukey median (which is known to be a robust estimate of the mean [82]), using the exponential mechanism of [69] to preserve privacy.

**Tukey median set.** For any set of points $S = \{x_i \in \mathbb{R}^d\}_{i=1}^n$ and $\hat{\mu} \in \mathbb{R}^d$, the *Tukey depth* is defined as the minimal empirical probability density on one side of a hyperplane that includes $\hat{\mu}$:

$$D_{\text{Tukey}}(S, \hat{\mu}) \;=\; \inf_{v \in \mathbb{R}^d} \mathbb{P}_{x \sim \hat{p}_n}(v^\top(x - \hat{\mu}) \geq 0) \;,$$

where $\hat{p}_n$ is the empirical distribution of $S$. The *Tukey median set* is defined as the set of points achieving the maximum Tukey depth, which might not be unique. Tukey median reduces to median for $d = 1$, and is a natural generalization of the median for $d > 1$. Inheriting robustness of one-dimensional median, Tukey median is known to be a robust estimator of the multi-dimensional mean under an adversarial perturbation. In particular, under our model, it achieves the optimal sample complexity and accuracy. This optimality follows from the well-known fact that the sample complexity of $O((1/\alpha^2)(d + \log(1/\zeta)))$ cannot be improved upon even if we have no corruption, and the fact that the accuracy of $O(\alpha)$ cannot be improved upon even if we have infinite samples [82]. However, finding a Tukey median takes exponential time scaling as $\tilde{O}(n^d)$ [67].

**Corollary I.1** (Corollary of [82, Theorem 3])**.** *For a dataset of $n$ i.i.d. samples from a $d$-dimensional Gaussian distribution $\mathcal{N}(\mu, \mathbf{I}_d)$, an adversary corrupts an $\alpha \in (0, 1/4)$ fraction of the samples as defined in Assumption 1. Then, any $\hat{\mu}$ in the Tukey median set of a corrupted dataset $S$ satisfies $\|\hat{\mu} - \mu\|_2 = O(\alpha)$ with probability at least $1 - \zeta$ if $n = \Omega((1/\alpha^2)(d + \log(1/\zeta)))$.*

**Exponential mechanism.** The exponential mechanism was introduced in [69] to elicit approximate truthfulness and remains one of the most popular private mechanisms due to its broad applicability. It can seamlessly handle queries with non-numeric outputs, such as routing a flow or finding a graph. Consider a utility function $u(S, \hat{\mu}) \in \mathbb{R}$ on a dataset $S$ and a variable $\hat{\mu}$, where higher utility is preferred. Instead of truthfully outputting $\arg\max_{\hat{\mu}} u(S, \hat{\mu})$, the exponential mechanism outputs a randomized approximate maximizer sampled from the following distribution:

$$r_S(\hat{\mu}) \;=\; \frac{1}{Z_S} e^{\frac{\varepsilon}{2\Delta_u} u(S, \hat{\mu})} \;, \tag{19}$$

where $\Delta_u = \max_{\hat{\mu}, S \sim S'} |u(S, \hat{\mu}) - u(S', \hat{\mu})|$ is the sensitivity of $u$ (from Definition 1.2) and $Z_S$ ensures normalization to one. This mechanism is $(\varepsilon, 0)$-differentially private, since $e^{\frac{\varepsilon}{2\Delta_u}|u(S,\hat{\mu})-u(S',\hat{\mu})|} \leq e^{\varepsilon/2}$ and $e^{-\varepsilon/2} \leq Z_S/Z_{S'} \leq e^{\varepsilon/2}$.

**Proposition I.2** ([69, Theorem 6] ). *The sampled $\hat{\mu}$ from the distribution* (19) *is $(\varepsilon, 0)$-differentially private.*

This naturally leads to the following algorithm. The privacy guarantee follows immediately since the Tukey depth has sensitivity $1/n$, i.e., $|D_{\text{Tukey}}(S_n, \hat{\mu}) - D_{\text{Tukey}}(S'_n, \hat{\mu})| \leq 1/n$ for all $\hat{\mu} \in \mathbb{R}^d$ and two neighboring databases $S_n \sim S'_n$ of size $n$. In this section, for the analysis of private Tukey median, we assume the mean is from a known bounded set of the form $[-R, R]^d$ for some known $R > 0$.

---
**Algorithm 1** Private Tukey median
---
Output a random data point $\hat{\mu} \in [-2R, 2R]^d$ sampled from a density $r(\hat{\mu}) \propto e^{(1/2)\varepsilon n D_{\text{Tukey}}(S,\hat{\mu})}$ .
---

The private Tukey median achieves the following near optimal guarantee, whose proof is provided in §J. The accuracy of $O(\alpha)$ and sample complexity of $n = \Omega((1/\alpha^2)(d + \log(1/\zeta)))$ cannot be improved even without privacy (cf. Corollary I.1), and $n = \tilde{\Omega}(d/(\alpha\varepsilon))$ is necessary even without any corruption [52, Theorem 6.5].

**Theorem 12.** *Under the hypotheses of Corollary I.1, there exists a universal constant $c > 0$ such that if $\mu \in [-R, R]^d$, $\alpha \leq \min\{c, R\}$ and $n = \Omega((1/\alpha^2)(d + \log(1/\zeta)) + (1/\alpha\varepsilon)d\log(dR/\zeta\alpha))$, then Algorithm 1 is $(\varepsilon, 0)$-differentially private and achieves $\|\hat{\mu} - \mu\|_2 = O(\alpha)$ with probability $1 - \zeta$.*

The private Tukey median, however, is a conceptual algorithm since we cannot sample from $r(\hat{\mu})$. The $\mathcal{A}_{FindTukey}$ algorithm from [9] approximately finds the Tukey median privately. This achieves $O(\alpha)$ accuracy with $n = \tilde{\Omega}(d^{3/2}\log(1/\delta)/(\alpha\varepsilon) + (1/\alpha^2)(d + \log(1/\zeta)))$, but it still requires a runtime of $O(n^{\text{poly}(d)})$. Alternatively, we can sample from an $\alpha$-cover of $[-2R, 2R]^d$, which has $O((dR/\alpha)^d)$ points. However, evaluating the Tukey depth of a point is an NP-hard problem [5], requiring a runtime of $\tilde{O}(n^{d-1})$ [66]. The runtime of the discretized private Tukey median is $\tilde{O}(n^{-1}(dnR/\alpha)^d)$. Similarly, [14] introduced an exponential mechanism over the $\alpha$-cover with a novel utility function achieving the same guarantee as Theorem 12, but this requires a runtime of $O(n(dR/\alpha)^{2d})$.

## J Proof of Theorem 12 on the accuracy of the exponential mechanism for Tukey median

First, the $(\varepsilon, 0)$-differential privacy guarantee of private Tukey median follows as a corollary of Proposition I.2, by noting that sensitivity of $n D_{\text{Tukey}}(\mathcal{D}_n, x)$ is one, where $\mathcal{D}_n$ is a dataset of size $n$. This follows from the fact that for any fixed $x$ and $v$, $|\{z \in \mathcal{D}_n : (v^\top(x - z)) \geq 0\}|$ is the number of samples on one side of the hyperplane, which can change at most by one if we change one sample in $\mathcal{D}$.

Next, given $n$ i.i.d samples $X_1, X_2, \ldots X_n$ from distribution $p$, denote $\hat{p}_n$ as the empirical distribution defined by the samples $X_1, X_2, \ldots X_n$. Denote $\tilde{p}_n$ as the distribution that is corrupted from $\hat{p}_n$. We slightly overload the definition of Tukey depth to denote $D_{\text{Tukey}}(p, x)$ as the Tukey depth of point $x \in \mathbb{R}^d$ under distribution $p$, which is defined as

$$D_{\text{Tukey}}(p, x) = \inf_{v \in \mathbb{R}^d} \mathbb{P}_{z \sim p}(v^\top(x - z) \geq 0).$$

Note that this is the standard definition of Tukey depth. First we show that for $n$ large enough, the Tueky depth for the empirical distribution is close to that of the true distribution. We provide proofs of the following lemmas later in this section.

**Lemma J.1.** *With probability $1 - \delta$, for any $p$ and $x \in \mathbb{R}^d$,*

$$|D_{Tukey}(p, x) - D_{Tukey}(\hat{p}_n, x)| \leq C \cdot \sqrt{\frac{d + 1 + \log(1/\delta)}{n}}.$$

The proof of Lemma J.1can be found in §J.1. This allows us to use the known Tukey depths of a Gaussian distribution to bound the Tukey depths of the corrupted empirical one. We use this to show

that there is a strict separation between the Tueky depth of a point in $S_1 = \{x : \|x - \mu\| \leq \alpha\}$ and a point in $S_2 = \{x : \|x - \mu\| \geq 10\alpha\}$. The proof of Lemma J.2 can be found in §J.2.

**Lemma J.2.** *Define $p = \mathcal{N}(\mu, I)$, and assume $\alpha < 0.01$. Given that $n = \Omega(\alpha^{-2}(d + \log(1/\delta)))$, with probability $1 - \delta$,*

1. *For any point $x \in \mathbb{R}^d$, $\|x - \mu\| \leq \alpha$, it holds that*

$$D_{\textit{Tukey}}(\tilde{p}_n, x) \geq \frac{1}{2} - 2\alpha$$

2. *For any point $x \in \mathbb{R}^d$, $\|x - \mu\| \geq 10\alpha$, it holds that*

$$D_{\textit{Tukey}}(\tilde{p}_n, x) \leq \frac{1}{2} - 5\alpha.$$

This implies that most of the probability mass of the exponential mechanism is concentrated inside a ball of radius $O(\alpha)$ around the true mean $\mu$. Hence, with high probability, the exponential mechanism outputs an approximate mean that is $O(\alpha)$ close to the true one. The following lemma finishes the proof the the desired claim, whose proof can be found in §J.3.

**Lemma J.3** (Utility). *Denote $\tilde{p}_n$ as the distribution that is corrupted from $\hat{p}_n$. Suppose $x$ is sampled from $[-2R, 2R]^d$ with density $r(x) \propto \exp(-(1/2)\varepsilon n D_{\textit{Tukey}}(\tilde{p}_n, x))$, then given $n = \Omega\big( (d/(\alpha\varepsilon)) \log(dR/\eta\alpha) + (1/\alpha^2)(d + \log(1/\eta)) \big)$ and $\mu \in [-R, R]^d$, and $R \geq \alpha$,*

$$\mathbb{P}(\|x - \mu\| \leq 5\alpha) \geq 1 - \eta .$$

## J.1 Proof of Lemma J.1

From the VC inequality ([25], Chap 2, Chapter 4.3) and the fact that the family of sets $\{\{z | v^\top z \geq t\} | \|v\| = 1, t \in \mathbb{R}, v \in \mathbb{R}^d\}$ has VC dimension $d + 1$, there exists some universal constant $C$ such that with probability at least $1 - \delta$

$$\sup_{t \in \mathbb{R}, v \in \mathbb{R}^d, \|v\|=1} |\mathbb{P}_{z \sim p}(v^\top z \geq t) - \mathbb{P}_{z \sim \hat{p}_n}(v^\top z \geq t)| \leq C \cdot \sqrt{\frac{d + 1 + \log(1/\delta)}{n}},$$

which implies, for any $x \in \mathbb{R}^d$,

$$\sup_{v \in \mathbb{R}^d} |\mathbb{P}_{z \sim p}(v^\top(x - z) \geq 0) - \mathbb{P}_{z \sim \hat{p}_n}(v^\top(x - z) \geq 0)| \leq C \cdot \sqrt{\frac{d + 1 + \log(1/\delta)}{n}},$$

by letting $t = v^\top x$. We conclude the proof since

$$|D_{\text{Tukey}}(p, x) - D_{\text{Tukey}}(\hat{p}_n, x)|$$
$$= |\inf_{v \in \mathbb{R}^d} \mathbb{P}_{z \sim p}(v^\top(x - z) \geq 0) - \inf_{v \in \mathbb{R}^d} \mathbb{P}_{z \sim \hat{p}_n}(v^\top(x - z) \geq 0)|$$
$$\leq \sup_{v \in \mathbb{R}^d} |\mathbb{P}_{z \sim p}(v^\top(x - z) \geq 0) - \mathbb{P}_{\hat{p}_n}(v^\top(x - z) \geq 0)|$$
$$\leq C \cdot \sqrt{\frac{d + 1 + \log(1/\delta)}{n}}.$$

## J.2 Proof of Lemma J.2

For the first claim, we first prove a lower bound on $D_{\text{Tukey}}(p, x)$. Since $p = \mathcal{N}(\mu, I)$, for any $v \in \mathbb{R}^d$ such that $\|v\|_2 = 1$,

$$\mathbb{P}_{z \sim p}(v^\top (z - x) \geq 0)$$

$$= \mathbb{P}_{z \sim N(0,1)}(z \geq v^\top(x - \mu))$$

$$= \int_{v^\top(x-\mu)}^\infty \frac{1}{\sqrt{2\pi}} \exp(-z^2/2) dz$$

$$\geq \frac{1}{2} - \frac{1}{\sqrt{2\pi}} v^\top(x - \mu)$$

$$\geq \frac{1}{2} - \frac{1}{\sqrt{2\pi}} \|x - \mu\|_2$$

$$\geq \frac{1}{2} - \frac{1}{\sqrt{2\pi}} \alpha$$

Thus,

$$D_{\text{Tukey}}(p, x)$$

$$= \inf_{v \in \mathbb{R}^d} \mathbb{P}_{z \sim p}(v^\top(x - z) \geq 0)$$

$$\geq \frac{1}{2} - \frac{1}{\sqrt{2\pi}} \alpha$$

Then Lemma J.1 implies that with probability $1 - \delta$

$$D_{\text{Tukey}}(\hat{p}_n, x) \geq \frac{1}{2} - \frac{1}{\sqrt{2\pi}} \alpha - C \cdot \sqrt{\frac{d + 1 + \log(1/\delta)}{n}}.$$

Since the corruption can change at most $\alpha$ probability mass, it holds that $|D_{\text{Tukey}}(\tilde{p}_n, x) - D_{\text{Tukey}}(\hat{p}_n, x)| \leq \alpha$. Setting $n = \Omega(\alpha^{-2}(d + \log(1/\delta)))$ yields

$$D_{\text{Tukey}}(\tilde{p}_n, x) \geq \frac{1}{2} - \frac{1}{\sqrt{2\pi}} \|x - \mu\|_2 - C \cdot \sqrt{\frac{d + 1 + \log(1/\delta)}{n}} - \alpha$$

$$\geq \frac{1}{2} - 2\alpha.$$

For the second claim, note that

$$D_{\text{Tukey}}(p, x)$$

$$\leq \int_{v^\top(x-\mu)}^\infty \frac{1}{\sqrt{2\pi}} \exp(-z^2/2) dz$$

$$\overset{(a)}{\leq} \frac{1}{2} - \frac{1}{\sqrt{2\pi}} \exp(-(20\alpha)^2/2) \cdot 20\alpha$$

$$\overset{(b)}{\leq} \frac{1}{2} - 7\alpha$$

where (a) holds since $\|x - \mu\| \geq 20\alpha$, and it is easy to verify that (b) holds for $\alpha \leq 0.01$. The second claim holds since

$$D_{\text{Tukey}}(\tilde{p}_n, x)$$

$$\leq D_{\text{Tukey}}(\hat{p}_n, x) + \alpha$$

$$\leq D_{\text{Tukey}}(p, x) + \alpha + C \cdot \sqrt{\frac{d + 1 + \log(1/\delta)}{n}}$$

$$\overset{(a)}{\leq} D_{\text{Tukey}}(p, x) + 2\alpha$$

$$\leq \frac{1}{2} - 5\alpha,$$

where $(a)$ holds by setting $n = \Omega(\alpha^{-2}(d + \log(1/\delta)))$.

### J.3 Proof of Lemma J.3

Let $r(x) = \frac{1}{A} \exp(-\varepsilon n D_{\text{Tukey}}(\tilde{p}_n, x))$ where $A$ is the normalization factor. Then

$$\mathbb{P}(\|x - \mu\| \leq \alpha) \geq \frac{1}{A} \exp(\varepsilon n(\frac{1}{2} - 2\alpha)) \cdot \frac{\pi^{d/2}}{\Gamma(d/2 + 1)} \alpha^d,$$

using the fact that $\mu \in [-R, R]^d$ and that $R \geq \alpha$, and

$$\mathbb{P}(\|x - \mu\| \geq 5\alpha) \leq \frac{1}{A} \exp(\varepsilon n(\frac{1}{2} - 10\alpha)) \cdot (4R)^d.$$

Hence

$$\log\left(\frac{\mathbb{P}(\|x - \mu\| \leq \alpha)}{\mathbb{P}(\|x - \mu\| \geq 5\alpha)}\right) \geq \varepsilon n(3\alpha) - C \cdot d \log(dR/\alpha),$$

where $C$ is an absolute constant. If we set $n = \Omega(\frac{d \log(dB/\delta\alpha)}{\alpha\varepsilon})$, we get that

$$\frac{\mathbb{P}(\|x - \mu\| \leq \alpha)}{\mathbb{P}(\|x - \mu\| \geq 5\alpha)} \geq \frac{10}{\delta},$$

which implies that with probability at least $1 - \delta$, $\|x - \mu\| \leq 5\alpha$.

# K  The algorithmic details and the analysis of PRIME-HT for covariance bounded distributions

We provide the algorithm and the analysis for the range estimation query $q_{\text{range}-\text{ht}}$, and then prove the result on analyzing PRIME-HT.

## K.1  Range estimation with $q_{\text{range}-\text{ht}}$

---

**Algorithm 14:** Differentially private range estimation for covariance bounded distributions ($q_{\text{range}-\text{ht}}$) [54, Algorithm 2]

---

**Input:** $S = \{x_i\}_{i=1}^n, \varepsilon, \delta, \zeta$

1  Randomly partition the dataset $S = \cup_{\ell \in [m]} S^{(\ell)}$ with $m = 200 \log(2/\zeta)$

2  $\bar{x}^{(\ell)} \leftarrow q_{\text{range}}(S^{(\ell)}, \varepsilon/m, \delta/m, \sigma = 40)$ for all $\ell \in [m]$

3  $\hat{x}_j \leftarrow \text{median}(\{\bar{x}_j^{(\ell)}\}_{\ell \in [m]})$ for all $j \in [d]$

**Output:** $(\hat{x}, B = 50/\sqrt{\alpha})$

---

**Lemma K.1.** $q_{\text{range}-\text{ht}}$ *is $(\varepsilon, \delta)$-differentially private. Under Assumption 2 and for $\alpha \in (0, 0.01)$, if $n = \Omega((1/\alpha)\log(1/\zeta) + (\sqrt{d \log(1/\delta)} \log(1/\zeta) \log(d/\delta)/\varepsilon))$, $q_{\text{range}-\text{ht}}$ returns a ball $\mathcal{B}_{\sqrt{d}B/2}(\bar{x})$ of radius $\sqrt{d}B/2$ centered at $\bar{x}$ that includes $(1 - 2\alpha)n$ uncorrupted samples where $B = 50/\sqrt{\alpha}$ with probability $1 - \zeta$.*

We first show that applying the private histogram to each coordinate provides a robust estimate of the range, but with a constant probability 0.9.

**Lemma K.2** (Robustness of a single private histogram). *Under the $\alpha$-corruption model of Assumption 2, if $n = \Omega(\sqrt{d \log(1/\delta)} \log(d/\delta)/\varepsilon)$, for $\alpha \in (0, 0.01)$, $q_{\text{range}}$ in Algorithm 5 with a choice of $\sigma = 40$ and $B = 120$ returns intervals $\{I_j\}_{j=1}^d$ of size $|I_j| = 240$ such that $\mu_j \in I_j$ with probability 0.9 for each $j \in [d]$.*

*Proof of Lemma K.2.* The proof is analogous to Appendix D.1 and we only highlight the differences here. By Lemma D.1 we know that $|\tilde{p}_k - \hat{p}_k| \leq 0.01$ with the assumption on $n$. The corruption can change the normalized count in each bin by $\alpha \leq 0.01$ by assumption. It follows from Chebyshev inequality that $\mathbb{P}(|x_{i,j} - \mu_j|^2 > \sigma^2) \leq 1/\sigma^2$. It follows from (e.g. [54, Lemma A.3]) that $\mathbb{P}(|\{i : x_{i,j} \notin [\mu - \sigma, \mu + \sigma]\}| > (100/\sigma^2)n) < 0.05$. Hence the maximum bin has $\tilde{p}_k \geq 0.5(1 - 100/\sigma^2) - 0.02$ and the true mean is in the maximum bin or in an adjacent bin. The largest non-adjacent bucket is at most $100/\sigma^2 + 0.02$. Hence, the choice of $\sigma = 40$ ensures that we find the $\mu$ within $3\sigma = 120$. $\square$

Following [54, Algorithm 2], we partition the dataset into $m = 200 \log(2/\zeta)$ subsets of an equal size $n/m$ and apply the median-of-means approach. Applying Lemma K.2, it is ensured (e.g., by [54, Lemma A.4]) that more than half of the partitions satisfy that the center of the interval is within 240 away from $\mu$, with probability $1 - \zeta$. Therefore the median of those $m$ centers is within 240 from the true mean in each coordinate. This requires the total sample size larger only by a factor of $\log(d/\zeta)$.

To choose a radius $\sqrt{d}B/2$ ball around this estimated mean that includes $1 - \alpha$ fraction of the points, we choose $B = 25/\sqrt{\alpha}$. Since $\|\hat{\mu} - \mu\|_2 \leq 120\sqrt{d} \ll \sqrt{d}B/2$ for $\alpha \leq 0.01$, this implies that we can choose $\sqrt{d}B/2$-ball around the estimated mean with $B = 50/\sqrt{\alpha}$.

Let $z_i = \mathbb{I}(\|x_i - \mu\|_2 > \sqrt{d}B/2)$. We know that $\mathbb{E}[z_i] = \mathbb{P}[(\|x_i - \mu\|_2 > \sqrt{d}B/2)] \leq \mathbb{E}[\|x_i - \mu\|_2^2(2/dB^2)] = (1/1250)\alpha$. Applying multiplicative Chernoff bound (e.g., in [54, Lemma A.3]), we get $|\{i : \|x_i - \mu\|_2 \leq \sqrt{d}B/2\}| \geq 1 - (3/2500)\alpha$ with probability $1 - \zeta$, if $n = \Omega((1/\alpha)\log(1/\zeta))$. This ensures that with high probability, $(1 - \alpha)$ fraction of the original uncorrupted points are included in the ball. Since the adversary can corrupt $\alpha n$ samples, at least $(1 - 2\alpha)n$ of the remaining good points will be inside the ball.

## K.2 Proof of Theorem 9

The proof of the privacy guarantee of Algorithm 15 follows analogously from the proof of the privacy of PRIME and is omitted here. The accuracy guarantee follows form the following theorem and Lemma K.1.

**Theorem 13** (Analysis of accuracy of DPMMWFILTER-HT). *Let $S$ be an $\alpha$-corrupted covariance bounded dataset under Assumption 2, where $\alpha \leq c$ for some universal constant $c \in (0, 1/2)$. Let $S_{\text{good}}$ be $\alpha$-good with respect to $\mu \in \mathbb{R}^d$. Suppose $\mathcal{D} = \{x_i \in \mathcal{B}_{\sqrt{d}B/2}(\bar{x})\}_{i=1}^n$ be the projected dataset. If $n \geq \widetilde{\Omega}\left(\frac{d^{3/2}B^2 \log(1/\delta)}{\varepsilon}\right)$, then DPMMWFILTER-HT terminates after at most $O(\log dB^2)$ epochs and outputs $S^{(s)}$ such that with probability $0.9$, we have $|S_t^{(s)} \cap S_{\text{good}}| \geq (1 - 10\alpha)n$ and*

$$\|\mu(S^{(s)}) - \mu\|_2 \lesssim \sqrt{\alpha} \ .$$

*Moreover, each epoch runs for at most $O(\log d)$ iterations.*

### K.2.1 Analysis of DPMMWFILTER-HT and a proof of Theorem 13

Algorithm 15 is a similar matrix multiplicative weights based filter algorithm for distributions with bounded covariance. Similarly, we first state following Lemma K.3 and prove Theorem 13 given Lemma K.3

**Lemma K.3.** *Let $S$ be an $\alpha$-corrupted bounded covariance dataset under Assumption 2. For an epoch $s$ and an iteration $t$ such that $\lambda^{(s)} > C$, $\lambda_t^{(s)} > 2/3\lambda_0^{(s)}$, and $n^{(s)} > 3n/4$, if $n \gtrsim \frac{B^2(\log B)d^{3/2}\log(1/\delta)}{\varepsilon}$ and $|S_t^{(s)} \cap S_{\text{good}}| \geq (1 - 10\alpha)n$, then with probability $1 - O(1/\log(d)^3)$, we have the condition in Eq. (20) holds. When this condition holds, we have more corrupted samples are removed in expectation than the uncorrupted samples, i.e., $\mathbb{E}|(S_t^{(s)} \setminus S_{t+1}^{(s)}) \cap S_{\text{good}}| \leq \mathbb{E}|(S_t^{(s)} \setminus S_{t+1}^{(s)}) \cap S_{\text{bad}}|$. Further, for an epoch $s \in [T_1]$ there exists a constant $C > 0$ such that if $\|M(S^{(s)})\|_2 \geq C$, then with probability $1 - O(1/\log^2 d)$, the $s$-th epoch terminates after $O(\log d)$ iterations and outputs $S^{(s+1)}$ such that $\|M(S^{(s+1)})\|_2 \leq 0.98\|M(S^{(s)})\|_2$.*

Now we define $d_t^{(s)} \triangleq |(S_{\text{good}} \cap S^{(1)}) \setminus S_t^{(s)}| + |S_t^{(s)} \setminus (S_{\text{good}} \cap S^{(1)})|$. Note that $d_1^{(1)} = \alpha n$, and $d_t^{(s)} \geq 0$. At each epoch and iteration, we have

$$\mathbb{E}[d_{t+1}^{(s)} - d_t^{(s)} | d_1^{(1)}, d_2^{(1)}, \cdots, d_t^{(s)}] \quad = \quad \mathbb{E}\left[|S_{\text{good}} \cap (S_t^{(s)} \setminus S_{t+1}^{(s)})| - |S_{\text{bad}} \cap (S_t^{(s)} \setminus S_{t+1}^{(s)})|\right] \quad \leq \quad 0,$$

from the part 1 of Lemma K.3. Hence, $d_t^{(s)}$ is a non-negative super-martingale. By optional stopping theorem, at stopping time, we have $\mathbb{E}[d_t^{(s)}] \leq d_1^{(1)} = \alpha n$. By Markov inequality, $d_t^{(s)}$ is less than $10\alpha n$ with probability $0.9$, i.e. $|S_t^{(s)} \cap S_{\text{good}}| \geq (1 - 10\alpha)n$. The desired bound in Theorem 13 follows from Lemma K.11.

### K.2.2 Proof of Lemma K.3

Lemma K.3 is a combination of Lemma K.4, Lemma K.5 and Lemma K.6. We state the technical lemmas and subsequently provide the proofs.

**Lemma K.4.** *For each epoch $s$ and iteration $t$, under the hypotheses of Lemma K.3 then with probability $1 - O(1/\log^3 d)$, we have*

$$\frac{1}{n}\sum_{i \in S_{\text{good}} \cap S_t^{(s)}} \tau_i \quad \leq \quad \psi/1000 \ , \tag{20}$$

*where $\psi \triangleq \frac{1}{n}\sum_{i \in S_t^{(s)}} \tau_i$.*

**Lemma K.5.** *For each epoch $s$ and iteration $t$, under the hypotheses of Lemma K.3, if condition Eq. (20) holds, then we have $\mathbb{E}|S_t^{(s)} \setminus S_{t+1}^{(s)} \cap S_{\text{good}}| \leq \mathbb{E}|S_t^{(s)} \setminus S_{t+1}^{(s)} \cap S_{\text{bad}}|$ and with probability $1 - O(1/\log^3 d)$, and $\left\langle M(S_{t+1}^{(s)}), U_t^{(s)} \right\rangle \leq 0.76 \left\langle M(S_t^{(s)}), U_t^{(s)} \right\rangle$.*

**Algorithm 15:** Differentially private filtering with matrix multiplicative weights (DPMMWFILTER-HT) for distributions with bounded covariance

**Input:** $S = \{x_i \in \mathcal{B}_{\sqrt{d}B/2}(\bar{x})\}_{i=1}^n$, $\alpha \in (0,1)$, $T_1 = O(\log B\sqrt{d})$, $T_2 = O(\log d)$, $B \in \mathbb{R}_+$, $(\varepsilon, \delta)$

1 **if** $n < (4/\varepsilon_1)\log(1/(2\delta_1))$ **then Output:** $\emptyset$

2 Initialize $S^{(1)} \leftarrow [n]$, $\varepsilon_1 \leftarrow \varepsilon/(4T_1)$, $\delta_1 \leftarrow \delta/(4T_1)$, $\varepsilon_2 \leftarrow \min\{0.9, \varepsilon\}/(4\sqrt{10T_1T_2\log(4/\delta)})$, $\delta_2 \leftarrow \delta/(20T_1T_2)$, a large enough constant $C > 0$

3 **for** epoch $s = 1, 2, \ldots, T_1$ **do**

4      $\lambda^{(s)} \leftarrow \|M(S^{(s)})\|_2 + \mathrm{Lap}(2B^2d/(n\varepsilon_1))$

5      $n^{(s)} \leftarrow |S^{(s)}| + \mathrm{Lap}(1/\varepsilon_1)$

6      **if** $n^{(s)} \leq 3n/4$ **then** terminate

7      **if** $\lambda^{(s)} \leq C$ **then**
        **Output:**
$$\mu^{(s)} \leftarrow (1/|S^{(s)}|)\Big(\sum_{i \in S^{(s)}} x_i\Big) + \mathcal{N}(0, (2B\sqrt{2d\log(1.25/\delta_1)}/(n\,\varepsilon_1))^2\mathbf{I}_{d\times d})$$

8      $\alpha^{(s)} \leftarrow 1/(100(0.1/C + 1.05)\lambda^{(s)})$

9      $S_1^{(s)} \leftarrow S^{(s)}$

10      **for** $t = 1, 2, \ldots, T_2$ **do**

11         $\lambda_t^{(s)} \leftarrow \|M(S_t^{(s)})\|_2 + \mathrm{Lap}(2B^2d/(n\varepsilon_2))$

12         **if** $\lambda_t^{(s)} \leq 2/3\lambda_0^{(s)}$ **then**

13            terminate epoch

14         **else**

15            $\Sigma_t^{(s)} \leftarrow M(S_t^{(s)}) + \mathcal{N}(0, (4B^2d\sqrt{2\log(1.25/\delta_2)}/(n\varepsilon_2))^2\mathbf{I}_{d^2\times d^2})$

16            $U_t^{(s)} \leftarrow (1/\mathrm{Tr}(\exp(\alpha^{(s)}\sum_{r=1}^t(\Sigma_r^{(s)})))) \exp(\alpha^{(s)}\sum_{r=1}^t(\Sigma_r^{(s)}))$

17            $\psi_t^{(s)} \leftarrow \langle M(S_t^{(s)}), U_t^{(s)}\rangle + \mathrm{Lap}(2B^2d/(n\varepsilon_2))$

18            **if** $\psi_t^{(s)} \leq (1/5.5)\lambda_t^{(s)}$ **then**

19               $S_{t+1}^{(s)} \leftarrow S_t^{(s)}$

20            **else**

21               $Z_t^{(s)} \leftarrow \mathrm{Unif}([0,1])$

22               $\mu_t^{(s)} \leftarrow (1/|S_t^{(s)}|)\big(\sum_{i \in S_t} x_i\big) + \mathcal{N}(0, (2B\sqrt{2d\log(1.25/\delta_2)}/(n\,\varepsilon_2)\mathbf{I}_{d\times d})^2)$

23               $\rho_t^{(s)} \leftarrow \mathrm{DPTHRESHOLD\text{-}HT}(\mu_t^{(s)}, U_t^{(s)}, \alpha, \varepsilon_2, \delta_2, S_t^{(s)})$         [Algorithm 16]

24               $S_{t+1}^{(s)} \leftarrow S_t^{(s)} \setminus \{i \mid \{\tau_j = (x_j - \mu_t^{(s)})^\top U_t^{(s)}(x_j - \mu_t^{(s)})\}_{j \in S_t^{(s)}}$ and $\tau_i \geq \rho_t^{(s)} Z_t^{(s)}\}$.

25      $S^{(s+1)} \leftarrow S_t^{(s)}$

**Output:** $\mu^{(T_1)}$

---

**Lemma K.6.** *For epoch $s$, suppose for $t = 0, 1, \cdots, T_2$ where $T_2 = O(\log d)$, if Lemma K.5 holds, $n \gtrsim \frac{B^2(\log B)d^{3/2}\log(1/\delta)}{\varepsilon\alpha}$, and $n^{(s)} > 3n/4$, then we have $\|M(S^{(s+1)})\|_2 \leq 0.98\|M(S^{(s)})\|_2$ with probability $1 - O(1/\log^2 d)$.*

### K.2.3 Proof of Lemma K.4

*Proof.* By Lemma G.9, Lemma G.10 and Lemma G.11, we can pick $n = \widetilde{\Omega}\left(\frac{B^2d^{3/2}\log}{\varepsilon}\right)$ such that with probability $1 - O(1/\log^3 d)$, following conditions simultaneously hold:

     1. $\|\mu_t^{(s)} - \mu(S_t^{(s)})\|_2^2 \leq 0.001$

     2. $|\psi_t^{(s)} - \langle M(S_t^{(s)}), U_t^{(s)}\rangle| \leq 0.001$

**Algorithm 16:** Differentially private estimation of the threshold for bounded covariance DPTHRESHOLD-HT

**Input:** $\mu, U, \alpha \in (0,1)$, target privacy $(\varepsilon, \delta)$, $S = \{x_i \in \mathcal{B}_{B\sqrt{d}/2}(\bar{x})\}$

1 Set $\tau_i \leftarrow (x_i - \mu)^\top U(x_i - \mu)$ for all $i \in S$

2 Set $\tilde{\psi} \leftarrow (1/n)\sum_{i \in S}\tau_i + \mathrm{Lap}(2B^2 d/n\varepsilon))$

3 Compute a histogram over geometrically sized bins
$$I_1 = [1/4, 1/2), I_2 = [1/2, 1), \ldots, I_{2+\log(B^2 d)} = [2^{\log(B^2 d)-1}, 2^{\log(B^2 d)}]$$

$$h_j \leftarrow \frac{1}{n} \cdot |\{i \in S \,|\, \tau_i \in [2^{-3+j}, 2^{-2+j})\}|, \quad \text{for all } j = 1, \cdots, 2 + \log(B^2 d)$$

4 Compute a privatized histogram $\tilde{h}_j \leftarrow h_j + \mathcal{N}(0, (4\sqrt{2d\log(1.25/\delta)}/(n\varepsilon))^2)$, for all $j \in [2 + \log(B^2 d)]$

5 Set $\tilde{\tau}_j \leftarrow 2^{-3+j}$, for all $j \in [2 + \log(B^2 d)]$

6 Find the largest $\ell \in [2 + \log(B^2 d)]$ satisfying $\sum_{j \geq \ell}(\tilde{\tau}_j - \tilde{\tau}_\ell)\tilde{h}_j \geq 0.31\tilde{\psi}$

**Output:** $\rho = \tilde{\tau}_\ell$

---

3. $\left|\lambda_t^{(s)} - \|M(S_t^{(s)})\|_2\right| \leq 0.001$

4. $\left|\lambda^{(s)} - \|M(S^{(s)})\|_2\right| \leq 0.001$

5. $\left\|M(S_{t+1}^{(s)}) - \Sigma_t^{(s)}\right\|_2 \leq 0.001$

6. $\|\mu^{(s)} - \mu(S^{(s)})\|_2^2 \leq 0.001$ .

Then we have

$$
\frac{1}{n}\sum_{i \in S_{\text{good}} \cap S_t^{(s)}} \tau_i = \frac{1}{n}\sum_{i \in S_{\text{good}} \cap S_t^{(s)}} \left\langle (x_i - \mu_t^{(s)})(x_i - \mu_t^{(s)})^\top, U_t^{(s)}\right\rangle
$$

$$
\stackrel{(a)}{\leq} \frac{2}{n}\sum_{i \in S_{\text{good}} \cap S_t^{(s)}} \left\langle (x_i - \mu(S_{\text{good}} \cap S_t^{(s)}))(x_i - \mu(S_{\text{good}} \cap S_t^{(s)}))^\top, U_t^{(s)}\right\rangle
$$

$$
+ \frac{2|S_{\text{good}} \cap S_t^{(s)}|}{n}\left\langle (\mu(S_{\text{good}} \cap S_t^{(s)}) - \mu_t^{(s)})(\mu(S_{\text{good}} \cap S_t^{(s)}) - \mu_t^{(s)})^\top, U_t^{(s)}\right\rangle
$$

$$
\leq 2\left\langle M((S_{\text{good}} \cap S_t^{(s)}), U_t^{(s)}\right\rangle + 2\|\mu_t^{(s)} - \mu(S_{\text{good}} \cap S_t^{(s)})\|_2^2
$$

$$
\stackrel{(b)}{\leq} 2 + 2\left(\|\mu_t^{(s)} - \mu\|_2 + \|\mu(S_{\text{good}} \cap S_t^{(s)}) - \mu\|_2\right)^2
$$

$$
\stackrel{(c)}{\leq} 2 + 2\left(0.01 + 2\sqrt{\alpha\|M(S_t^{(s)})\|_2} + 3\sqrt{\alpha}\right)^2
$$

$$
\leq 3 + 8\alpha\|M(S_t^{(s)})\|_2 + 32\alpha
$$

$$
\stackrel{(d)}{\leq} \frac{\psi_t^{(s)} - 0.002}{1000}
$$

$$
\leq \frac{\psi}{1000} ,
$$

where $(a)$ follows from the fact that for any vector $x, y, z$, we have $(x - y)(x - y)^\top \preceq 2(x - z)(x - z)^\top + 2(y - z)(y - z)^\top$, $(b)$ follows from $\alpha$-goodness of $S_{\text{good}}$, $(c)$ follows from Lemma K.11 and $(d)$ follows from our choice of large constant $C$ and sample complexity $n$.

$\square$

### K.2.4 Proof of Lemma K.5

*Proof.* Lemma K.4 implies with probability $1 - O(1/\log^3 d)$, our scores satisfies the condition in Eq. (20). Then by Lemma K.7 our DPTHRESHOLD-HT gives us a threshold $\rho$ such that

$$\sum_{i \in S_{\text{good}} \cap S_t^{(s)}} \mathbf{1}\{\tau_i \leq \rho\}\frac{\tau_i}{\rho} + \mathbf{1}\{\tau_i > \rho\} \leq \sum_{i \in S_{\text{bad}} \cap S_t^{(s)}} \mathbf{1}\{\tau_i \leq \rho\}\frac{\tau_i}{\rho} + \mathbf{1}\{\tau_i > \rho\} .$$

According to our filter rule from Algorithm 16, we have

$$\mathbb{E}|(S_t^{(s)} \setminus S_{t+1}^{(s)}) \cap S_{\text{good}}| = \sum_{i \in S_{\text{good}} \cap S_t^{(s)}} \mathbf{1}\{\tau_i \leq \rho\}\frac{\tau_i}{\rho} + \mathbf{1}\{\tau_i > \rho\}$$

and

$$\mathbb{E}|(S_t^{(s)} \setminus S_{t+1}^{(s)}) \cap S_{\text{bad}}| = \sum_{i \in S_{\text{bad}} \cap S_t^{(s)}} \mathbf{1}\{\tau_i \leq \rho\}\frac{\tau_i}{\rho} + \mathbf{1}\{\tau_i > \rho\} .$$

This implies $\mathbb{E}|(S_t^{(s)} \setminus S_{t+1}^{(s)}) \cap S_{\text{good}}| \leq \mathbb{E}|(S_t^{(s)} \setminus S_{t+1}^{(s)}) \cap S_{\text{bad}}|$.

At the same time, Lemma K.7 gives us a $\rho$ such that with probability $1 - O(\log^3 d)$, we have

$$\frac{1}{n}\sum_{i \in S_{t+1}^{(s)}} \tau_i \leq \frac{1}{n}\sum_{\tau_i \leq \rho, i \in S_t^{(s)}} \tau_i \leq \frac{3}{4} \cdot \frac{1}{n}\sum_{i \in S_t^{(s)}} \tau_i .$$

Hence, we have

$$
\begin{aligned}
\left\langle M(S_{t+1}^{(s)}), U_t^{(s)} \right\rangle &= \left\langle \frac{1}{n}\sum_{i \in S_{t+1}^{(s)}} (x_i - \mu(S_{t+1}^{(s)}))(x_i - \mu(S_{t+1}^{(s)}))^\top, U_t^{(s)} \right\rangle \\
&\leq \left\langle \frac{1}{n}\sum_{i \in S_{t+1}^{(s)}} (x_i - \mu(S_t^{(s)}))(x_i - \mu(S_t^{(s)}))^\top, U_t^{(s)} \right\rangle \\
&\leq \frac{1}{n}\sum_{i \in S_{t+1}^{(s)}} \tau_i + \|\mu_t^{(s)} - \mu(S_t^{(s)})\|_2^2 \\
&\leq \frac{3}{4n}\sum_{i \in S_t^{(s)}} \tau_i + 0.01 \\
&\overset{(a)}{\leq} 0.76 \left\langle M(S_t^{(s)}), U_t^{(s)} \right\rangle ,
\end{aligned}
$$

where $(a)$ follows from our assumption that $\psi_t^{(s)} > \frac{1}{5.5}\lambda_t^{(s)} > \frac{2}{16.5}C$.

$\square$

### K.2.5 Proof of Lemma K.6

*Proof.* If Lemma K.5 holds, we have

$$
\begin{aligned}
\left\langle M(S_t^{(s)}), U_t^{(s)} \right\rangle &\leq 0.76 \left\langle M(S_{t-1}^{(s)}), U_t^{(s)} \right\rangle \\
&\leq 0.76 \left\langle M(S_1^{(s)}), U_t^{(s)} \right\rangle \\
&\leq 0.76\|M(S_1^{(s)})\|_2
\end{aligned}
$$

We pick $n$ large enough such that with probability $1 - O(\log^3 d)$,

$$\|\Sigma_t^{(s)}\|_2 \approx_{0.05} \|M(S_t^{(s)})\|_2 .$$

Thus, we have

$$\left\langle \Sigma_t^{(s)}, U_t^{(s)} \right\rangle \le 0.81 \|M(S_1^{(s)})\|_2 .$$

By Lemma G.1, we have $M(S_t^{(s)}) \preceq M(S_1^{(s)})$. by our choice of $\alpha^{(s)}$, we have $\alpha^{(s)} M(S_{t+1}^{(s)}) \preceq \frac{1}{100}\mathbf{I}$ and $\alpha^{(s)} \Sigma_t^{(s)} \preceq \frac{1}{100}\mathbf{I}$. Therefore, by Lemma G.14 we have

$$\left\| \sum_{i=1}^{T_2} \Sigma_t^{(s)} \right\|_2$$

$$\le \sum_{t=1}^{T_2} \left\langle \Sigma_t^{(s)}, U_t^{(s)} \right\rangle + \alpha^{(s)} \sum_{t=0}^{T_2} \left\langle U_t^{(s)}, \left| \Sigma_t^{(s)} \right| \right\rangle \|\Sigma_t^{(s)}\|_2 + \frac{\log(d)}{\alpha^{(s)}}$$

$$\overset{(a)}{\le} \sum_{t=1}^{T_2} \left\langle \Sigma_t^{(s)}, U_t^{(s)} \right\rangle + \frac{1}{100} \sum_{t=1}^{T_2} \left\langle U_t^{(s)}, \left| \Sigma_t^{(s)} \right| \right\rangle + 200 \log(d) \|M(S_1^{(s)})\|_2$$

where $(a)$ follows from our choice of $\alpha^{(s)}$, $C$, and $n$.

Meanwhile, we have

$$|\Sigma_t^{(s)}| \preceq M(S_t^{(s)}) + 0.15\, \mathbf{I} .$$

Thus we have

$$\left\langle U_t^{(s)}, \left| \Sigma_t^{(s)} \right| \right\rangle \le 0.91 \left\| M(S_1^{(s)}) \right\|_2$$

Then we have

$$\left\| M(S_{T_2}^{(s)}) \right\|_2 \le \frac{1}{T_2} \left\| \sum_{i=1}^{T_2} M(S_t^{(s)}) \right\|_2$$

$$\le \frac{1}{T_2} \left\| \sum_{i=1}^{T_2} \Sigma_t^{(s)} \right\|_2 + 0.05 \, \|M(S_1^{(s)})\|_2$$

$$\le \frac{1}{T_2} \left( \sum_{t=1}^{T_2} \left\langle \Sigma_t^{(s)}, U_t^{(s)} \right\rangle + \frac{1}{100} \sum_{t=1}^{T_2} \left\langle U_t^{(s)}, \left| \Sigma_t^{(s)} \right| \right\rangle + 200 \log(d) \|M(S_1^{(s)})\|_2 \right) + 0.05 \, \|M(S_1^{(s)})\|_2$$

$$\le 0.91 \|M(S_1^{(s)})\|_2 + \frac{200 \log(d)}{T_2} \|M(S_1^{(s)})\|_2 + 0.05 \, \|M(S_1^{(s)})\|_2$$

$$\le 0.98 \, \|M(S_1^{(s)})\|_2$$

$\qquad\qquad\qquad\qquad\qquad\qquad\qquad\qquad\qquad\qquad\qquad\qquad\qquad\qquad\qquad\qquad \square$

### K.2.6 Proof of DPTHRESHOLD-HT for distributions with bounded covariance

**Lemma K.7** (DPTHRESHOLD-HT: picking threshold privately for distributions with bounded covariance). *Algorithm* DPTHRESHOLD-HT$(\mu, U, \alpha, \varepsilon, \delta, S)$ *running on a dataset* $\{\tau_i = (x_i - \mu)^\top U (x_i - \mu)\}_{i \in S}$ *is* $(\varepsilon, \delta)$-*DP. Define* $\psi \triangleq \frac{1}{n} \sum_{i \in S} \tau_i$. *If* $\tau_i$*'s satisfy*

$$\frac{1}{n} \sum_{i \in S_{\text{good}} \cap S} \tau_i \quad \le \quad \psi/1000 ,$$

*and* $n \ge \tilde{\Omega}\left( \frac{B^2 d}{\varepsilon} \right)$ *then* DPTHRESHOLD-HT *outputs a threshold* $\rho$ *such that*

$$2\Big( \sum_{i \in S_{\text{good}} \cap S} \mathbf{1}\{\tau_i \le \rho\} \frac{\tau_i}{\rho} + \mathbf{1}\{\tau_i > \rho\} \Big) \le \sum_{i \in S_{\text{bad}} \cap S} \mathbf{1}\{\tau_i \le \rho\} \frac{\tau_i}{\rho} + \mathbf{1}\{\tau_i > \rho\} , \qquad (21)$$

*and with probability* $1 - O(1/\log^3 d)$,

$$\frac{1}{n} \sum_{\tau_i < \rho} \tau_i \quad \le \quad 0.75\psi .$$

*Proof.* **1. $\rho$ cuts enough**

Let $\rho$ be the threshold picked by the algorithm. Let $\hat{\tau}_i$ denote the minimum value of the interval of the bin that $\tau_i$ belongs to. It holds that

$$\frac{1}{n}\sum_{\tau_i \geq \rho, i \in [n]}(\tau_i - \rho) \geq \frac{1}{n}\sum_{\hat{\tau}_i \geq \rho, i \in [n]}(\hat{\tau}_i - \rho)$$

$$= \sum_{\tilde{\tau}_j \geq \rho, j \in [2+\log(B^2 d)]}(\tilde{\tau}_j - \rho)h_j$$

$$\overset{(a)}{\geq} \sum_{\tilde{\tau}_j \geq \rho, j \in [2+\log(B^2 d)]}(\tilde{\tau}_j - \rho)\tilde{h}_j - O\left(\log(B^2 d) \cdot B^2 d \cdot \frac{\sqrt{\log(\log(B^2 d)\log d)\log(1/\delta)}}{\varepsilon n}\right)$$

$$\overset{(b)}{\geq} 0.31\tilde{\psi} - \tilde{O}(\frac{B^2 d}{\varepsilon n})$$

$$\overset{(c)}{\geq} 0.3\psi - \tilde{O}(\frac{B^2 d}{\varepsilon n}),$$

where $(a)$ holds due to the accuracy of the private histogram (Lemma G.12), $(b)$ holds by the definition of $\rho$ in our algorithm, and $(c)$ holds due to the accuracy of $\tilde{\psi}$. This implies

$$\frac{1}{n}\sum_{\tau_i < \rho}\tau_i \leq \psi - \frac{1}{n}\sum_{\tau_i \geq \rho}(\tau_i - \rho) \leq 0.7\psi + \tilde{O}(B^2 d/\varepsilon n).$$

**2. $\rho$ doesn't cut too much**

Define $C_2$ to be the threshold such that $\frac{1}{n}\sum_{\tau_i > C_2}(\tau_i - C_2) = (2/3)\psi$. Suppose $2^b \leq C_2 \leq 2^{b+1}$, we have $\sum_{\hat{\tau}_i \geq 2^{b-1}}(\hat{\tau}_i - 2^{b-1}) \geq (1/3)\psi$ because $\forall \tau_i \geq C_2$, $(\hat{\tau}_i - 2^{b-1}) \geq \frac{1}{2}(\tau_i - C_2)$. Then the threshold picked by the algorithm $\rho \geq 2^{b-1}$, which implies $\rho \geq \frac{1}{4}C_2$. Suppose $\rho < C_2$, since $\rho \geq \frac{1}{4}C_2$

$$\sum_{i \in S_{\text{bad}} \cap S, \tau_i < \rho}\tau_i + \sum_{i \in S_{\text{bad}} \cap S, \tau_i \geq \rho}\rho \geq \frac{1}{4}\left(\sum_{i \in S_{\text{bad}} \cap S, \tau_i < C_2}\tau_i + \sum_{i \in S_{\text{bad}} \cap S, \tau_i \geq C_2}C_2\right)$$

$$\overset{(a)}{\geq} \frac{10}{4}\left(\sum_{i \in S_{\text{good}} \cap S, \tau_i < C_2}\tau_i + \sum_{i \in S_{\text{good}} \cap S, \tau_i \geq C_2}C_2\right)$$

$$\overset{(b)}{\geq} \frac{10}{4}\left(\sum_{i \in S_{\text{good}} \cap S, \tau_i < \rho}\tau_i + \sum_{i \in S_{\text{good}} \cap S, \tau_i \geq \rho}\rho\right),$$

where (a) holds by Lemma K.8, and (b) holds since $\rho \leq C_2$. If $\rho \geq C_2$, the statement of the Lemma K.8 directly implies Equation (21).

**Lemma K.8.** *Assuming that the condition in Eq.(20) holds, then for any $C$ such that*

$$\frac{1}{n}\sum_{i \in S, \tau_i < C}\tau_i + \frac{1}{n}\sum_{i \in S, \tau_i \geq C}C \geq (1/3)\psi,$$

*we have*

$$\sum_{i \in S_{\text{bad}} \cap S, \tau_i < C}\tau_i + \sum_{i \in S_{\text{bad}} \cap S, \tau_i \geq C}C \geq 10\left(\sum_{i \in S_{\text{good}} \cap S, \tau_i < C}\tau_i + \sum_{i \in S_{\text{good}} \cap S, \tau_i \geq C}C\right)$$

*Proof.* First we show an upper bound on $S_{\text{good}}$:

$$\frac{1}{n}\sum_{i \in S_{\text{good}} \cap S, \tau_i < C}\tau_i + \frac{1}{n}\sum_{i \in S_{\text{good}} \cap S, \tau_i \geq C}C \leq \frac{1}{n}\sum_{i \in S_{\text{good}} \cap S}\tau_i \leq \psi/1000.$$

Then we show an lower bound on $S_{\text{bad}}$:

$$\frac{1}{n} \sum_{i \in S_{\text{bad}} \cap S, \tau_i < C} \tau_i + \frac{1}{n} \sum_{i \in S_{\text{bad}} \cap S, \tau_i > C} C$$

$$= \quad \frac{1}{n} \sum_{i \in S, \tau_i < C} \tau_i + \frac{1}{n} \sum_{i \in S, \tau_i \geq C} C$$

$$- (\frac{1}{n} \sum_{i \in S_{\text{good}} \cap S, \tau_i < C} \tau_i + \frac{1}{n} \sum_{i \in S_{\text{good}} \cap S, \tau_i \geq C} C)$$

$$\geq \quad (1/3 - 1/1000)\psi .$$

Combing the lower bound and the upper bound yields the desired statement $\qquad\square$

$\square$

### K.2.7 Regularity lemmas for distributions with bounded covariance

**Definition K.9** ([36, Definition 3.1] ). *Let $D$ be a distribution with mean $\mu \in \mathbb{R}^d$ and covariance $\Sigma \preceq \mathbf{I}$. For $0 < \alpha < 1/2$, we say a set of points $S = \{X_1, X_2, \cdots, X_n\}$ is $\alpha$-good with respect to $\mu \in \mathbb{R}^d$ if following inequalities are satisfied:*

- $\|\mu(S) - \mu\|_2 \leq \sqrt{\alpha}$

- $\left\| \frac{1}{|S|} \sum_{i \in S} (X_i - \mu(S)) (X_i - \mu(S))^\top \right\|_2 \leq 1.$

**Lemma K.10** ([36, Lemma 3.1] ). *Let $D$ be a distribution with mean $\mu \in \mathbb{R}^d$ and covariance $\Sigma \preceq \mathbf{I}$. Let $S = \{X_1, X_2, \cdots, X_n\}$ be a set of i.i.d. samples of $D$. If $n = \Omega(d \log(d)/\alpha)$, then with probability $1 - O(1)$, there exists a set $S_{\text{good}} \subseteq S$ such that $S_{\text{good}}$ is $\alpha$-good with respect to $\mu$ and $|S_{\text{good}}| \geq (1 - \alpha)n$.*

**Lemma K.11** ([36, Lemma 3.2] ). *Let $S$ be an $\alpha$-corrupted bounded covariance dataset under Assumption 2. If $S_{\text{good}}$ is $\alpha$-good with respect to $\mu$, then for any $T \subset S$ such that $|T \cap S_{\text{good}}| \geq (1 - \alpha)|S|$, we have*

$$\|\mu(T) - \mu\|_2 \leq \frac{1}{1 - 2\alpha} \cdot \left( 2\sqrt{\alpha \|M(T)\|_2} + 3\sqrt{\alpha} \right) .$$

# L   Experiments

We evaluate PRIME and compare with a DP mean estimator of [52] on synthetic dataset in Figure 1 and Figure 2, which consists of samples from $(1 - \alpha)\mathcal{N}(0, \mathbf{I}) + \alpha\mathcal{N}(\mu_{\text{bad}}, \mathbf{I})$. The main focus of this evaluation is to compare the estimation error and demonstrate the robustness of PRIME under differential privacy guarantees. Our choice of experimental settings and hyper parameters are as follows: $1 \le d \le 100$, $\mu_{\text{bad}} = (1.5, 1.5, \cdots, 1.5)_d$, $0.001 \le \varepsilon \le 100$, $0.01 \le \alpha \le 0.1$ , $C = 1$.

Figure 2 shows additional experiments including the regime where we do not have enough number of samples. When $n \le cd^{1.5}/\alpha\varepsilon$, the utility guarantee (Theorem 5) does not hold. The noise we add on the final output becomes large as $n$ decreases and dominates the estimation error. The DP Mean [52] has lower error compared to PRIME when $n$ is small because PRIME spends some privacy budget to perform operations other than those in DP Mean in the Algorithm 10. In practice, we can check whether there are enough number of samples based on known parameters $(\varepsilon, \delta, n, \alpha)$, and choose to use DP Mean (or adjust how the privacy budget is distributed in PRIME).

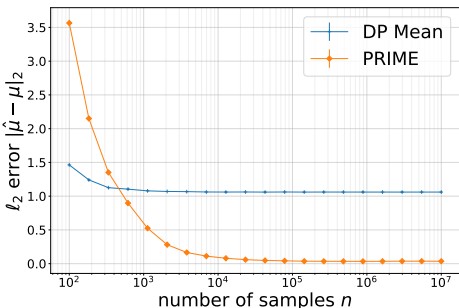

Figure 2: Estimation error achieved by PRIME significantly improves upon that of DP Mean in the large sample regime where our theoretical guarantees apply. In the small sample regime, the noise from the DP mechanisms dominate the error, which increases with decreasing $n$. We choose $(\alpha, \varepsilon, \delta, d) = (0.1, 100, 0.01, 50)$. Each data point is repeated 50 runs and standard error is shown in the error bar.

Our implementation is based on Python with basic Numpy library. We run on a 2018 Macbook Pro machine. For each choice of $d$ in our settings, it takes less than 2 minutes and PRIME stops after at most 3 epochs. We have attached our code as supplementary materials.