# OpenReview forum: "Robust and differentially private mean estimation"
_NeurIPS.cc/2021/Conference — NeurIPS 2021 Poster_

### Official Review · Reviewer_n7D3 · 2021-07-06

**Rating:** 7
**Confidence:** 4

**Summary:**

The paper considers the problem of robust mean estimation under adversarial contamination in the differentially private setting. In the canonical setting of robust mean estimation, one is given access to $n$ i.i.d samples from a well-behaved distribution over $\mathbb{R}^d$ and the goal is to estimate the mean of the distribution. The catch is that an adversary is allowed to inspect the samples and arbitrarily corrupt an $\alpha$ fraction of them. Standard assumptions on the distribution include assuming a spectral bound on the covariance matrix of the distribution although the optimal recovery guarantees improve under additional assumptions such as gaussianity. The key difference in the setup considered in this paper is that the algorithm is also required to be differentially private with respect to the data points. The problems of building robust and differentially private algorithms have received much attention in the recent part in isolation and building ones which satisfy both of these properties is a natural one.

**Limitations And Societal Impact:**

Yes

**Main Review:**

The paper considers the problem of robust mean estimation under adversarial contamination in the differentially private setting. In the canonical setting of robust mean estimation, one is given access to $n$ i.i.d samples from a well-behaved distribution over $\mathbb{R}^d$ and the goal is to estimate the mean of the distribution. The catch is that an adversary is allowed to inspect the samples and arbitrarily corrupt an $\alpha$ fraction of them. Standard assumptions on the distribution include assuming a spectral bound on the covariance matrix of the distribution although the optimal recovery guarantees improve under additional assumptions such as gaussianity. The key difference in the setup considered in this paper is that the algorithm is also required to be differentially private with respect to the data points. The problems of building robust and differentially private algorithms have received much attention in the recent part in isolation and building ones which satisfy both of these properties is a natural one.

The paper proposes two main algorithms for the problem. The first is an algorithm which runs in polynomial time but suffers a sub-optimal sample complexity of $O(d^{3/2} / \alpha^2)$ (as opposed to the information theoretically optimal rate of $O(d / \alpha^2)$). The second is an exponential time algorithm which nevertheless obtains the optimal sample complexity up to log factors ($O(d / \alpha^2)$). The efficient algorithm itself seems to be largely an adaption of the filtering based approach to robust mean estimation where one first obtains a spectral certificate determining which data points are likely to be outliers and uses this certificate to either remove corrupted data points or downweighting them in the computation of the mean estimate. The looseness in the sample complexity is due to the differentially private singular vector computation step that is used to compute the spectral certificate. The in-efficient algorithm in the paper essentially computes a noisy estimate of the solution to a direction wise trimmed mean problem. Roughly speaking, one searches for a mean estimate whose projection onto any direction $v$ is close to the trimmed mean of the data points when projected onto $v$ where the trimming eliminates the top and bottom $O(\alpha)$ percentiles of the data points respectively. While this geometric insight existed in prior work as well, the adaptation to the differentially private setting (as done in Algorithm 2 of this paper) seems quite clever and novel even if it isn't emphasized as much.

Overall, the contributions of the paper are valuable and interesting as the first work at the intersection of robustness and differential privacy. While the efficient algorithm is sub-optimal in terms of sample complexity, there seem to be clever insights in the construction of the inefficient algorithm which may enable future work on the topic.


**Time Spent Reviewing:**

4

---

> ### Author Response · Authors · 2021-08-10
> **Thank you for your insightful feedback.**
>
> Thank you for your constructive feedback. The geometric insight of Algorithm 2 is one of the main technical contributions, and we will emphasize it more in the revision.

---

### Official Review · Reviewer_i7TG · 2021-07-08

**Rating:** 7
**Confidence:** 4

**Summary:**

This paper shows results on robust and differentially private mean estimation of sub-Gaussian and covariance bounded distributions. So far, work has mostly been done in the direction of either robustness or of differentially private statistics, but this paper puts both of them together. Specifically, it provides near-optimal (but exponential time) algorithm and sub-optimal (but polynomial time) algorithms for mean estimation of the aforementioned types of distributions.

For their polynomial-time algorithm, PRIME, they use an MMW filter-based approach, which they privatise in a novel filtering algorithm (which they call, "DP-Threshold"). This reduces the iteration complexity of the filter and the number of database accesses in each iteration.

Their exponential-time algorithm exploits the resilience property of the distributions to bound the sensitivity of the robust mean.

Finally, they provide experimental results to back their theoretical results.

**Limitations And Societal Impact:**

Limitations have been mentioned in the detailed review. There doesn't seem to be any negative societal impact of this work.

**Main Review:**

Update: Upon the authors' response, I have decided to update my final evaluation.

Strengths:
1. The paper is very technical, and clearly a lot of work went into it.
2. The paper is also quite well-written. It's organised and explains the background well enough.
3. It is the first paper to give results for robust and differentially private mean estimation of sub-Gaussians and covariance bounded distributions. So, there's novelty in it.
4. Their exponential-time algorithms are nearly optimal.
5. The experimental results indicate significant improvements over any kinds of prior work in differential privacy. However, it is important to note that the comparison with the prior work of [KLSU'19] is a bit apples-to-oranges, as their work does not focus on robustness at all.

Weaknesses:
1. Their polynomial-time algorithms have $O(d^{3/2})$ sample complexity. The authors do highlight this deficiency in their work. New techniques altogether may be required to solve this problem, it seems, since going via the route of private PCA would incur such big costs.
2. It is a bit surprising to see $(\varepsilon,\delta)$-differentially private algorithms needing a range on the mean $[-R,R]^d$. I say this because of the existence of stability-based methods in privacy. Is there a way to remove that dependence completely?
3. This is kind of a nitpick, and the authors discuss this, too. The third graph in Figure 1. The accuracy of their algorithm is worse than the differentially private prior methods for value of $\varepsilon$ less than $0.05$. This raises the question: are these algorithms really practical?

Comments:
1. Typo: Figure 1 - second graph - y axis should be labelled as $||\hat{\mu}-\mu||_2$.
2. Line 355 should read as: We show that robustness can be achieved...

**Time Spent Reviewing:**

4

---

> ### Author Response · Authors · 2021-08-10
> **Thank you for the insightful comments.**
>
> We thank the reviewer for the constructive feedback.
>
> **Re: (empirical) comparison with the prior work of [KLSU'19].**
>
> We agree with the reviewer that the experimental comparisons are apple to oranges, and we will explicitly state that DPmean is not designed for robustness. This choice of comparison was made for the following reasons: 1) there is no competing method that provides robustness and privacy jointly, and hence there is no other baseline that is apples-to-apples; and 2) [KLSU ‘19] provides some robustness **“coincidentally”** due to the fact that they use histogram-based method (without any formal robustness guarantees), and they are more robust than, for example, the simplest baseline of just taking the empirical mean. We wanted to convey the message that this coincidental robustness is not enough (as the error now increases with the dimension), and we will make it more clear in the writing when we revise the manuscript.
>
> **Re: 1. Cost of $d^{3/2}$.**
>
> We agree that new techniques are required, which opens up an exciting line of new research questions: Can we tightly analyze the sensitivity of filtering algorithms, such that we can bypass the release of private covariance matrix? Is there a computational lower bound (in the spirit of SQ lower bounds for statistical complexity) to robust and private mean estimation? What is the fundamental tradeoff of private PCA under statistical assumptions on the data?
>
> **Re: 2. Removing bounded support [-R,R]^d.**
> We want to note that this is necessary for the $q_{\rm range}$ to work, and we believe that it is possible to completely remove the dependence on $\log R$ from the sample complexity and also remove the condition that we need $\mu$ from this bounded support. In short, we can take $R\to \infty$ and all the analysis will hold with a little change. Here are the changes that need to be made.
> The first change is in the theorem statement. In the sample complexity of PRIME (Theorems 2 and 6), there will be an extra term scaling as
> $$ \frac{\sqrt{d\log\frac1\delta} \log\frac{d}{\zeta\delta}}{\varepsilon} .$$
> Note that this additional sample complexity only scales as $\sqrt{d}$, and hence has little effect on the overall sample complexity (unless $\delta$ is really small). Technically, this factor comes from Lemma D.2 in the supplementary material, where we can upper bound $\min ( \log(dR/\zeta), \log\frac{d}{\zeta\delta} ) \leq \log\frac{d}{\zeta\delta}$ to get rid of the sample complexity dependence on $\log R$.
>
> Second change is in the algorithm. To achieve this while requiring no assumption on the boundedness of $\mu$, we need the private histogram mechanism that does not depend on $R$. Here is the slightly modified one that satisfies our goal. Instead of binning (coordinatewise) the interval [-R,R] with a bin-size of 2, we bin the entire number line [-\infty,\infty] with a bin-size of 2. This can be done efficiently as we only need to bin those bins that are occupied by samples, requiring at most $n$ bins to be considered. This is possible as we follow the advanced private histogram scheme [56] that does not add any noise to unoccupied bins. This satisfies the same privacy and utility guarantee while removing any requirement on the boundedness of $\mu$.
>
> This is possible as we are already using a “stability-based” private histogram mechanism from [56], which does not add noise to empty bins, and hence adapts to the topology of the data. It is possible to use mechanisms that are more explicitly stability-based also. For example, [3] achieves optimal DP mean estimation guarantee that improves the extra term of $ \frac{\sqrt{d\log\frac1\delta} \log\frac{d}{\zeta\delta}}{\varepsilon}$ down to $\frac{\log(1/\delta)}{\varepsilon}$, which is information theoretically optimal.
>
> **Re: 3. Practicality of the algorithms.**
>
> The practical choice of $\varepsilon$ depends on the application, but we believe $\varepsilon$ close to one is a common choice. Further, we did not try to optimize over the choices of constants in PRIME to get the best “threshold” where the DP noise dominates even the adversarial corruption. One could more carefully track all the choices in the algorithm to get further gain, which does not change the sample complexity as only constant factors change, but can potentially improve the practical performance. This we believe is outside the scope of this paper.

---

### Official Review · Reviewer_nCbz · 2021-07-15

**Rating:** 7
**Confidence:** 3

**Summary:**

The paper proposes an algorithm which is both DP and robust for "poisoning attacks" for mean estimation. Namely, it can deal with an adversary corrupting \alpha*n many points and still produces a mean which is fairly close to the true mean (of the remainder of the points); and do which without directly looking at the point but rather statistical properties that are randomly perturbed in order to preserve DP. In addition to the theoretical work, there was also a few experiments conducted on the output of the mechanism.

**Limitations And Societal Impact:**

Irrelevant.

**Main Review:**

I think this is an interesting paper that tackles DP-mean estimation from a new angle -- dealing with robustness and removing outliers. It is likely that it has implications - in terms of outlier detection - beyond what is discussed in paper and I am very fond of the idea of DP outlier detection/removal, as it means that the approach is likely to work even without some outlier.

On the flip side, I think that (a) the paper is poorly written (some mistakes are just plain embarrassing) and (b) I am baffled about the experiments. Firstly, as a theory-focused paper, I see little contribution in adding figures showing "the math works"; and more importantly, the first two figures are based on a very large \eps value (unless I misunderstood the experiment section) so they are of very little importance. So I am left with the rightmost figure (pun intended), in which the error doesn't decay to 0 as \eps increases. Is this a result of the fact that the robust mean estimation tends to be somewhat "off" even without privacy? A comparison to the non-private baseline would have helped here.

Still, all in all I think this is an interesting paper and I advocate acceptance.

Typos:
* Line 110: The Lap-noise is to the power k, as each coordinate is perturbed.
* Line 121: \forall v.
* Line 139: The current set S_t (not S_0)
* Line 212: What's S'? Is this a general lemma for any S' or just for neighboring datasets?
* U_t^{(s)} has a completely unclear definition. You sum over r yet you do not use the index r at all! The summation -- should it be in the numerator? Is it in the right place in the denominator? (Maybe I am missing something here...)
* Section 2.3.2 needs a lot of work (at the expense of the discussion post Thm 6 + Sec. 5 which is mostly repetitive to the intro). What's the intuition for this \rho? Where did the 0.31 factor come from?
* Def 4.2: Don't get why S' has size (1-2\alpha) rather than (1-\alpha) and T should have size >= |S'|-\alpha*n, no? Some discussion as to the bounds in the resilience score would be much appreciated here.
* Line 351: Theta *inside* big O definition???



**Time Spent Reviewing:**

Refuse to time myself.

---

> ### Author Response · Authors · 2021-08-10
> **Thank you for your careful reading and insightful comments.**
>
> We thank the reviewer for the constructive feedback.
>
> **Re: Experimental results.**
>
> We agree that there is little contribution in showing synthetic data experiments on a theory-focused paper. It was not meant to give new information, but rather to make it easy to understand the theorems via examples. As in the “rightmost” figure, even without privacy (and even with infinite samples), one cannot achieve a vanishing error when $\alpha>0$. There is a fundamental lower bound of $\min_{\hat\mu} \max_{P_\mu,{\rm adversary}} ||\hat\mu-\mu || \geq  2 \Phi^{-1}(\frac{1}{2}(1+\frac{\alpha}{2})) = \Omega( \alpha)$, where $\Phi$ is the CDF of a standard 1-D Gaussian and we are minimizing over all estimators and maximizing over the worst-case distribution $P_\mu$ that is sub-Gaussian and worst-case  adversarial corruption of $\alpha$ fraction of data. Note that this lower bonus does not depend on $\varepsilon$ or $n$. For the choice of $\alpha=0.1$ in the rightmost figure, this lower bound is 0.126, and we will plot this as a baseline. Note that this is a lower bound on the minimax rate, where we take maximum over all possible adversarial corruptions. Since our simulated adversary is not the strongest one, it is possible that the error of PRIME (for large $\varepsilon$) can be smaller than 0.126.
>
> **Re: Line 212: Is this a general lemma for any S' or just for neighboring datasets?**
>
> Lemma 2.2 applies generally to any set $S’$. When we use this lemma for our proofs, we only apply it to neighboring datasets $S$ and $S’$. We will make it clear in the revision.
>
> **Re: U_t^{(s)} has a completely unclear definition.**
>
> We thank the reviewer for pointing this out. It was a typo. The correct definition is $U_t^{(s)} =\frac{\exp\Big( \alpha^{(s)} \sum_{r\in[t]} ({\rm Cov}(S_r^{(s)}) -{\mathbf I})\Big)}{{\rm Tr}\big(\exp( \alpha^{(s)} \sum_{r\in[t]} ({\rm Cov}(S_r^{(s)}) -{\mathbf I})) \big)}$.
> The role of the denominator is to normalize the resulting $U_t^{(s)}$ to have a Trace of one.
>
> **Section 2.3.2 needs a lot of work (at the expense of the discussion post Thm 6 + Sec. 5 which is mostly repetitive to the intro). What's the intuition for this \rho? Where did the 0.31 factor come from?**
>
> We will add more details and intuitions to Section 2.3.2. The $\rho$ is used as the threshold that filters samples with lower scores $\tau_i$ (line 24 in Algorithm 10 in the Supplementary Material). Specifically, at each filter step, each sample is filtered with a threshold $\rho\times Z$, where $Z\sim \text{Uniform}([0,1])$. If $\tau_i$ is larger than $\rho Z$ (and if $\tau_i$ is in the largest $2\alpha$ tail of the set \{$\tau_i$\}), then $x_i$ will be filtered. Furthermore, we need to make sure that $\rho$ is large enough, so we can filter more corrupted points than good points (Appendix E.2, Lemma E.1, Eq. (13) ). We also need to make sure $\rho$ is small enough so that every time we can filter enough points and reduce total score to make progress (Appendix E.2, Lemma E.1 Eq. (12)). Finally, we need to make sure $\rho$ is differentially private. The details on how we use this $\rho$ can be found at Appendix E.1 Algorithm 10. To privately get $\rho$, we applied DPthreshold (Appendix E.2 Algorithm 11). The Eq. (1) in the main paper makes sure that we have picked $\rho$ large enough to satisfy the progress condition. The details are in the proof of Lemma E.1 (Appendix E.3). In the progress condition of Lemma E.1, we need to show that the summation of the scores that are less than 0.75 times total scores. The constant 0.31 can be replaced by any constant that is between 1/4 and 1/3 to meet the requirement of this specific progress condition.
>
> We will move some of the technical conditions that $\rho$ needs to satisfy from Appendix to the main text, and clarify how we designed the DPthreshold algorithm.
>
> **Def 4.2: Don't get why S' has size (1-2\alpha) rather than (1-\alpha) and T should have size >= |S'|-\alpha*n, no? Some discussion as to the bounds in the resilience score would be much appreciated here.**
>
> We will add more discussions. In short, we wanted to write a single algorithm for both sub-Gaussian and heavy-tailed distributions. Sub-Gaussian can work fine with $S’$ of size $1-\alpha$, but covariance-bounded distributions require a smaller set $S’$ of size $(1-2\alpha)$. The reason is that unlike the sub-Gaussian counterpart, a dataset of size $n$ from a heavy-tailed distribution is not resilient as is. Instead you need to remove $\alpha n$ data points that have large norms and only then the remaining dataset of size $(1-\alpha)n$ is resilient. On top of this, an adversary can corrupt $\alpha n$ of those remaining points, and hence we get $(1-2\alpha)n$ data points that are well-behaved. This is the reason we need to work with a smaller set $S’$ that is resilient. Lemma I.1 in the Supplementary Material shows that we get a smaller resilient set for the covariance bounded case.
>
> **Typos:** All typos will be fixed in the revision, including line 351: Computing R(S): “exactly can take $d e^{\Theta(n)}$ operations”.

---

> > ### Comment · Reviewer_nCbz · 2021-08-26
> > **Authors' response acknowledged**
> >
> > Review remains unaltered.

---

### Official Review · Reviewer_reXf · 2021-07-21

**Rating:** 7
**Confidence:** 4

**Summary:**

New algorithms for estimating Gaussian means that are both robust and differentially private.

**Limitations And Societal Impact:**

See main review for technical limitations. The work is theoretical and doesn't directly address (or imply) social impact.

**Main Review:**

This paper provides two new results on estimating the mean of a distribution via algorithms that are both robust (to an $\alpha$ fraction of corruptions) and differentially private.

There are two new algorithms:

1. An exponential-time algorithm that samples from the "exponential mechanism" (Gibbs distribution) over a rough bounding box for the distribution. The score function for the exponential mechanism is the Tukey depth of a point in the data set. The authors observe that this empirical Tukey depth is close to the analogous population quantity on correctly distributed data. (A similar reasoning underlies the simplest nonprivate robust estimation algorithm, which returns a point of maximimum Tukey depth. However, analyzing the randomized version requires understanding how the volumes of the level sets of the Tukey depth change with the depth.)

2. A polynomial-time algorithm based on the (nonprivate) matrix-multiplicative-weights-based robust of Dong, Hopkins and Li. It was a little hard for me to understand what the real technical insight here was, but it seems like an important contribution, and one that provides a good starting point for future investigations.

The paper is clearly written and lays out its contributions succinctly. I think the authors could have done more to spell out dependencies on hidden parameters like R (the side-length of the bounding box from which the mean is drawn), and also to discuss how sensitive the results are to unknown co-variance structures. (For example, suppose we know a PSD matrix A such that the true covariance lies between $A$ and $cA$. Do the algorithm discussed here have sample complexity that scales polynomially with $c$? Logarithmically?) For what it's worth, I don't think it's fair to swallow dependencies on $R$ into the $\tilde O(\cdot)$ notation, since we don't know _a priori_ how $R$ relates to other quantities like $d$.

I view the contributions of the paper largely in formulating the problem and laying out initial solutions. I think it should be accepted based on those. That said, it would have been nice to understand what the technical roadblocks were to analyzing the noisy version of the MMW-based algorithm of Hopkins et al, and also to see some discussion of whether there are fundamental relationships between robust and private estimation in high dimensions—for example, is some automatic conversion between the two possible, along the lines of the work of Dwork and Lei (STOC 2009)?



**Time Spent Reviewing:**

3

---

> ### Author Response · Authors · 2021-08-10
> **Thank you for the constructive feedback.**
>
> We thank the reviewer for the constructive feedback.
>
> **Re: Dependency on R.** Reading the reviewers’ comments, we realized that the dependence on $\log(R)$ is much more significant than, say, dependence on $\log d$ or $\log(1/\zeta)$. For the efficient PRIME, the only dependence on $R$ is  $n =\Omega( \frac{\sqrt{d\log(1/\delta)}\min \big( \log R, \log (1/\delta)\big) }{\varepsilon})$ (from the analysis of $q_{\rm range}$ in Lemma D.2). Note that this dependence is mild as it is minimum between $\log R$ and $\log(1/\delta)$. We will make this dependence explicit in the revision. In particular, by replacing the minimum by $\log(1/\delta)$, we will get an extra term in the sample complexity of $n =\Omega( \frac{\sqrt{d\log(1/\delta)} \log (1/\delta)}{\varepsilon})$ and completely remove the dependence on $R$ in the sample complexity. In this case, the tilde notation will not be hiding any $R$ dependent factors.
> For the exponential time algorithm, Theorem 7 explicitly shows all the dependence on $\log R$, namely
>
> $$
> n=\widetilde\Omega\Big( \frac{d\log\frac{d R}{\alpha}}{\varepsilon\alpha} +
> \frac{\sqrt{d\log\frac1\delta}\min \big( \log \frac{dR}{\zeta},\log\frac{d}{\zeta\delta}\big) }{\varepsilon} +
> \frac{d+\log\frac{1}{\zeta}}{\alpha^2\log\frac1\alpha} +
> \frac{ d^{1/2}\log\frac{1}{\delta} + \log\frac{1}{\zeta} }{\varepsilon \alpha} \Big)
> $$
>
> The $\log R$ tem in the first factor is due to our loose bound on the support of the exponential mechanism. We can tighten this by using a hyper-cube of side length $\log(dn/\zeta)$ (which we know privately from $q_{\rm range}$) to bound the support of the exponential mechanism, in which case the $\log R$ dependence in the first term will be replaced by $\log(\log(dn/\zeta))$, which is very small. This is only a slight change in the algorithm.
>
> The second $\log R$ term again shows as the minimum $\min(\log R,\log(1/\delta))$, which can be replaced by $\log(1/\delta)$. These changes completely remove any dependence on $R$ in the sample complexity, and we will make these changes in the revision.
> Further, by slightly changing the $q_{\rm range}$ mechanism that uses private histogram to get a rough support of the data, we can completely remove the condition that $\mu$ needs to come from a bounded set, i.e. $R$ can be infinity. Here is the slightly modified $q_{\rm range}$ that satisfies our goal. Instead of binning (coordinatewise) the interval [-R,R] with a bin-size of 2, we bin the entire number line [-\infty,\infty] with a bin-size of 2. This can be done efficiently as we only need to bin those bins that are occupied by samples, requiring at most $n$ bins to be considered. This is possible as we follow the advanced private histogram scheme [56] that does not add any noise to unoccupied bins. This satisfies the same privacy and utility guarantee while removing any requirement on the boundedness of $\mu$.
>
>
> **Re: dependence on the covariance $A\preceq \Sigma \preceq cA$, . Do the algorithm discussed here have sample complexity that scales polynomially with c or Logarithmically?**
> For bounded covariance case, the error scales linearly in $c$ as the covariance scales linearly in $c$, i.e., $||\hat\mu-\mu|| = O(  \sqrt{ ||A ||} \sqrt{c \alpha } )$. The sample complexity stays the same.
>
> For sub-Gaussian case, the resulting error of PRIME is (ignoring logarithmic factors) $||\hat\mu-\mu|| = O ( \sqrt{||A||} ( \sqrt{(c-1) \alpha} + \alpha) )$. When $c=1$ we recover the previous error that scales as $O(\alpha)$. For larger $c$, the error increases up to $O(\sqrt{(c-1)\alpha})$. In terms of the resulting sample complexity, let $\beta$ denote the target error rate. Then, we require sample complexity of
> $$ n = \Omega \Big(\frac{d}{\beta^2} + \frac{d^{3/2}}{\varepsilon \min ( \beta,\beta^2/(c-1) ) } \Big)$$
> Which has a polynomial dependence on $c$ and recovers the previous sample complexity when $c$ is small enough.
>
> **Re: Technical roadblocks.**
> A direct adoption of [Hopkins et al.]’s algorithm by adding Gaussian noise to all the statistics does not provide DP, because sensitivity of that algorithm is prohibitively large. We need to make two technical innovations to achieve our goal.
> The first is the novel DPthreshold that achieves i) desired utility, ii) desired privacy, and most importantly iii) the sensitivity of the resulting set is preserved in the sense of Lemma 2.2. Such a mechanism did not exist previously, and it requires careful design and analysis.
>
> The second is the preservation of “sensitivity” over multiple iterations of filtering in Lemma 2.2. To the best of our knowledge such end-to-end sensitivity of an iterative algorithm is new. Here we are overloading the notion of sensitivity to refer to the set difference between two neighboring sets and how this difference evolves over iterations.
> These two technical components are critical in PRIME. Once we have made these two innovations, the utility analysis of the noisy filtering algorithm is straightforward. The technically most challenging part is the design and analysis of DPthreshold.
>
> **Re: fundamental relationships between robust and private estimation in high dimensions and connections to Dwork and Lei (STOC 2009).**
> Thank you for the reference. [Dwork,Lei] is a seminal work in connecting robustness and privacy and we will refer to it appropriately in the revision. We believe that there is a high-level connection between robustness and privacy, which our work provides some understanding of (along several existing results following up on [Dwork,Lei]). For instance our work suggests a certain computational gap might be there. We believe that our work opens several research directions at the intersection of robustness and privacy and that the fundamental conversion between the two is one such direction for future research.

---

### Decision · Program_Chairs · 2021-09-27

**Decision:**

Accept (Poster)

**Comment:**

The reviewers agreed that this is a strong and interesting technical work. The authors are advised to address dependence on the range parameter R, since this is often important in DP estimation tasks, as well as some related work mentioned by reviewers. Some weaknesses identified by reviewers include experiments (which is not the main contribution of the paper) and the suboptimal sample complexity of the efficient algorithm (which seems like a significant open problem), which can both be overlooked.